# Learning Equilibria in Adversarial Team Markov Games: A Nonconvex-Hidden-Concave Min-Max Optimization Problem

**Fivos Kalogiannis**[*]
University of California, Irvine
Archimedes/Athena RC, Greece

**Jingming Yan**[*]
University of California, Irvine

**Ioannis Panageas**
University of California, Irvine
Archimedes/Athena RC, Greece

## Abstract

We study the problem of learning a Nash equilibrium (NE) in Markov games which is a cornerstone in multi-agent reinforcement learning (MARL). In particular, we focus on infinite-horizon adversarial team Markov games (ATMGs) in which agents that share a common reward function compete against a single opponent, *the adversary*. These games unify two-player zero-sum Markov games and Markov potential games, resulting in a setting that encompasses both collaboration and competition. [65] provided an efficient equilibrium computation algorithm for ATMGs which presumes knowledge of the reward and transition functions and has no sample complexity guarantees. We contribute a learning algorithm that utilizes MARL policy gradient methods with iteration and sample complexity that is polynomial in the approximation error $\epsilon$ and the natural parameters of the ATMG, resolving the main caveats of the solution by [65]. It is worth noting that previously, the existence of learning algorithms for NE was known for Markov two-player zero-sum and potential games but not for ATMGs.

Seen through the lens of min-max optimization, computing a NE in these games consists a nonconvex–nonconcave saddle-point problem. Min-max optimization has received extensive study. Nevertheless, the case of nonconvex-nonconcave landscapes remains elusive: in full generality, finding saddle-points is computationally intractable [33]. We circumvent the aforementioned intractability by developing techniques that exploit the hidden structure of the objective function via a nonconvex–concave reformulation. However, this introduces the challenge of a feasibility set with coupled constraints. We tackle these challenges by establishing novel techniques for optimizing weakly-smooth nonconvex functions, extending the framework of [35].

## 1 Introduction

Multi-agent reinforcement learning (MARL) investigates behaviors of multiple interacting agents within a dynamic, shared environment where the actions of each agent not only impact their individual rewards but also the overall state of the system. MARL has introduced several practical techniques that have justifiably captured public interest in recent years, particularly in skill-intensive games like starcraft, go, chess, and poker [12, 97, 101, 79, 16, 15, 14, 86], where its empirical methods have

---

[*]Equal Contribution

38th Conference on Neural Information Processing Systems (NeurIPS 2024).

achieved super-human performance. More recently, MARL methods combined with large language models has excelled in the game of Diplomacy [6]. Despite these practical achievements, theoretical understanding of MARL has lagged behind its empirical successes.

Markov games (MGs) [95] is a rigorous and versatile mathematical structure that MARL employs to systematically formalize the strategic interactions in the dynamic settings [71]. These games extend Markov decision processes (MDPs) [88] to multiple agents, each making decisions and receiving rewards independently as the environment evolves. The joint decisions of the agents influence both individual rewards and the transition of the environment. MARL in general is occupied with leading the multi-agent system to a favorable outcome. Through the lens of game theory, the notion of a "favorable outcome" is formally defined through concepts like a Nash equilibrium and a (coarse) correlated equilibrium. Although computing Nash equilibria is generally computationally intractable—even in two-player games without states [28, 24]—it becomes tractable in fully cooperative settings like Markov potential games [114, 68] and is also tractable in competitive scenarios such as two-player zero-sum Markov games [27, 103, 17]. Recent advances [65] also show computational tractability in adversarial team Markov games (ATMGs)—a context that combines both cooperative and competitive dynamics among agents. More specifically, an infinite-horizon adversarial team Markov game (ATMG) is a Markov Game in which $n$ team players, compete against one adversary. Each of the team players receives the same reward and is equal to minus the reward of the adversary. ATMGs generalize both Markov zero-sum and potential games; the former can be viewed as ATMGs with $n = 1$, the latter by choosing the adversary to be dummy (having one action).

Nash equilibrium computation in ATMGs naturally leads to a min-max optimization problem. Min-max optimization has been deeply explored across game theory, optimization, and machine learning. The past decade it has witnessed a proliferation of min-max optimization applications, notably in areas like generative adversarial networks (GANs) [49], robust machine learning [73], and adversarial training [50]. In these applications, the optimization objectives often involve nonconvex–nonconcave functions which pose substantial challenges. Typically, the aim is to approximate saddle-points of $f(\boldsymbol{x}, \boldsymbol{y})$. In normal form games, these points correspond to Nash equilibria. This correspondence also holds true for MGs due to the gradient domination property [3]. Although we cannot aspire to cover the vast quantity of works in MARL and optimization, we select some representative works that we defer to Appendix A due to space constraints.

This paper aims to develop learning methods to approximate Nash equilibria in team Markov games by using only individual rewards and state observations as feedback, addressing the following question and answering one of the main caveats of the solutions provided in [65]:

> *Is it possible for agents to efficiently learn Nash equilibria in adversarial team Markov games, having only access to trajectory roll-out samples and (almost\*) no communication, i.e., independently?*    ($\star$)

## 1.1 Our Contributions

Let us provide some context before stating our main results. An infinite-horizon adversarial team Markov game (ATMG) is characterized by a finite state-space $\mathcal{S}$, $n$ team players, each equipped with a finite action-space $\mathcal{A}_i$, $i \in \{1, \ldots, n\}$, and one adversary with a finite action-space $\mathcal{B}$. Each of the team players receives the same reward which is equal to minus the reward of the adversary. The adversary's value function is defined as the discounted expected sum of their rewards, where the discount factor is $\gamma \in [0, 1)$. An approximate Nash equilibrium is a product distribution over policy space such that no agent can improve their value by unilaterally deviating. We propose a learning algorithm that has both iteration and sample complexity polynomial in the parameters of the Markov Game and returns approximate Nash equilibria.

**Theorem 1.1** (Informal Version of Theorem 3.3). *There is a learning algorithm (ISPNG) that uses bandit feedback and guarantees convergence to an $\epsilon$-approximate Nash equilibrium in adversarial*

---

\*We say "almost" as the agents need to take turns in updating their policies instead of making updates simultaneously. Nevertheless, the learning dynamics remain uncoupled.

*team Markov games, the sample and iteration complexities of which are*

$$\mathsf{poly}\left(\frac{1}{\epsilon}, |\mathcal{S}|, \sum_{k=1}^{n} |\mathcal{A}_i| + |\mathcal{B}|, \frac{1}{1-\gamma}\right).$$

We deem noteworthy that our algorithm manages to compute a Nash equilibrium in a Markov game, which combines opposing and shared agent interests, by only using a number of iterations and samples that is polynomial in the approximation error and the description of the game. Further, it manages to beat the *curse of multi-agents* [62]—*i.e.,* its iteration and sample complexity depends on $\sum_{k=1}^{n} |\mathcal{A}_i|$ instead of $\prod_{k=1}^{n} |\mathcal{A}_i|$.

In order to achieve the latter contribution, we acquired convergence guarantees for stochastic projected gradient descent in nonconvex functions when the gradient is Hölder-continuous—a notion of continuity weaker than that of Lipschitz. Finally, we contribute a general result that guarantees convergence to a saddle-point in functions that are nonconvex–hidden-strongly-concave.

## 1.2   Technical Overview

The problem of computing an approximate Nash equilibrium in an adversarial team Markov game boils down to computing an approximate saddle-point $(\boldsymbol{x}^*, \boldsymbol{y}^*)$ of the adversary's value function $V(\boldsymbol{x}, \boldsymbol{y})$; see Definition 2.3. The variables $\boldsymbol{x}$ denote the policies of the team, each member of which aims to individually minimize $V$. Moreover, $\boldsymbol{y}$ denotes the policy of the adversary who aims to maximize $V$. The equivalence between saddle-points and equilibria is due to (i) the game being *zero-sum* between the team and the adversary and (ii) the *gradient domination property* (see Lemma C.7) that holds per player, and has already been established in prior works [3, 68, 114]. In words, gradient domination in our setting implies that any approximate first-order stationary policy is also an approximate best response for that player.

The problem of computing an approximate saddle-point $(\boldsymbol{x}^*, \boldsymbol{y}^*)$ of the objective $V(\boldsymbol{x}, \boldsymbol{y})$ poses computational challenges due to its nonconvex–nonconcave nature. Previous work [65] showed that one can compute an approximate saddle-point $(\boldsymbol{x}^*, \boldsymbol{y}^*)$ of $V$, by first obtaining an approximate stationary point $\boldsymbol{x}^*$ of $\Phi(\boldsymbol{x}) = \max_{\boldsymbol{y}} V(\boldsymbol{x}, \boldsymbol{y})$ through a Moreau envelope argument and then extending it to $(\boldsymbol{x}^*, \boldsymbol{y}^*)$. The proof of extendibility uses involved arguments that utilize the Lagrange multipliers of a carefully chosen nonlinear program (for the stationary point $\boldsymbol{x}^*$), while the computation of $\boldsymbol{y}^*$ requires solving another linear program. It is worth noting that the aforementioned linear program presumes access to the full description of the reward function and the transition model of the underlying Markov game when the team plays policy $\boldsymbol{x}^*$. This fact prevents the possibility of casting this approach into a learning algorithm.

Our proposed (learning) algorithm bypasses the requirement for knowledge of the reward function and the transition model, and works under the bandit feedback framework. The first idea behind our algorithm is to consider the adversary's value function as a function $F$ of the *adversary's* state-action visitation measure $\boldsymbol{\lambda}$, $F(\boldsymbol{x}, \boldsymbol{\lambda}) \coloneqq V(\boldsymbol{x}, \boldsymbol{y})$, and the addition of a regularizing term $-\frac{\nu}{2} \|\boldsymbol{\lambda}\|^2$ ($\nu$ can be thought of as a small positive scalar). As a result, the max function of the regularized value function, $\Phi^\nu(\boldsymbol{x}) \coloneqq \max_{\boldsymbol{\lambda} \in \Lambda(\boldsymbol{x})} \left\{ F(\boldsymbol{x}, \boldsymbol{\lambda}) - \frac{\nu}{2} \|\boldsymbol{\lambda}\|^2 \right\}$, is differentiable, where $\Lambda(\boldsymbol{x}) \subseteq \Delta^{|\mathcal{S}||\mathcal{B}|}$ denotes the feasibility set of $\boldsymbol{\lambda}$ and depends on $\boldsymbol{x}$. Effectively, different policies, $\boldsymbol{x}$, for the team induce a different single agent Markov decision process for the adversary. The addition of the regularizer allows us to apply Danskin's theorem on a function with a unique maximizer circumventing the necessity of solving a linear program; one only needs to approach that unique solution. To the best of our knowledge, this is the first work introducing a function of $\boldsymbol{\lambda}$ as a regularizing term.

By reformulating the regularized value function using state-action visitation measure $\boldsymbol{\lambda}$, the problem boils down to learning an approximate saddle-point of a nonconvex–strongly-concave function with *coupled constraints*. Coupled constraints are a type of constraints that cannot be expressed as a Cartesian product (the main well-studied setting in min-max optimization [63]), *i.e.*, the feasibility set $\Lambda(\boldsymbol{x})$, depends on $\boldsymbol{x}$. The first challenge towards handling the coupled constraints is to argue that $\nabla \Phi^\nu$ is Hölder-continuous which is a notion of continuity weaker than Lipschitz continuity (see Definition 2.1). Specifically, in Theorem 3.2, we show that $\Phi^\nu(\boldsymbol{x})$ is weakly-smooth, or equivalently, $\nabla \Phi^\nu$ is Hölder-continuous. It seems unlikely that we could use Moreau envelope techniques to prove convergence of stochastic projected gradient descent on a weakly-smooth function. The next

step of our proof is to transfer the weakly-smooth nonconvex optimization problem into a smooth optimization problem with inexact gradient oracles, extending the techniques from [35] to nonconvex and constrained settings. Since we only allow each player to observe the reward they received and not the action chosen by the other players (including the adversary), one last challenge we have to deal with is the inability to estimate the state-action visitation measure $\boldsymbol{\lambda}$ of the adversary, making the gradient inexact when computing $\nabla \Phi^\nu(x)$ in both deterministic and stochastic settings.

## 2 Preliminaries

Starting, we will introduce the notation conventions we use and split the rest of the preliminaries into two subsections. Section 2.1 provides necessary definitions whereas Section 2.2 deals with the preliminaries of (adversarial team) Markov games and the notion of Nash equilibrium.

**Notation.** We denote $[n] := \{1, \ldots, n\}$. We use superscripts to denote the (discrete) time index, and subscripts to index the players. We use boldface for vectors and matrices; scalars will be denoted by lightface variables. We define $\|\cdot\|_2, \|\cdot\|_1, \|\cdot\|_\infty$ to be the $\ell_2$-norm, the $\ell_1$-norm and the $\ell_\infty$ norm respectively. The simplex of probability vectors supported on a finite set $\mathcal{A}$ is noted as $\Delta(\mathcal{A})$. Unless specified otherwise, we denote $\|\cdot\|_2$ by $\|\cdot\|$. $\mathrm{Diam}_{\mathcal{X}}$ denotes the diameter of a compact set $\mathcal{X}$ in $\ell_2$-distance. For simplicity in the exposition, we may sometimes use the $O(\cdot)$ notation to suppress dependencies that are polynomial in the natural parameters of the problem and $\tilde{O}(\cdot)$ to further hide logarithmic factors; precise statements are given in the Appendix. For the convenience of the reader, a comprehensive overview of our notation is given in Table 1.

### 2.1 Basic Definitions and Facts

We commence this subsection by introducing a number of concepts and statements of mathematical analysis and optimization. We define Hölder continuity and the notion of a stationary point in constrained minimization and min-max optimization.

The notion of Hölder continuity of the gradient is a weaker notion of Lipschitz gradient continuity.

**Definition 2.1** ($p$-Hölder continuous gradient)**.** *A function $\phi : \mathbb{R}^d \to \mathbb{R}$ is said to have a $(\ell_p, p)$-Hölder continuous gradient if for every $\boldsymbol{z}, \boldsymbol{z}' \in \mathbb{R}^d$, it holds that:*

$$\|\nabla\phi(\boldsymbol{z}) - \nabla\phi(\boldsymbol{z}')\|_2 \leq \ell_p \|\boldsymbol{z} - \boldsymbol{z}'\|_2^p.$$

*When $p = 1$, we retrieve the definition of an $\ell$-smooth function.*

Throughout, following standard conventions, we will refer to functions for which the gradient is $p$-Hölder continuous with a $p < 1$ as *weakly-smooth*. We state the notions of first-order stationarity relevant to our work.

**Definition 2.2** ($\epsilon$-FOSP)**.** *In the context of the constrained minimization problem $\min_{\boldsymbol{z} \in \mathcal{Z}} \phi(\boldsymbol{z})$, a point $\boldsymbol{z} \in \mathcal{Z}$ is said to be an $\epsilon$-approximate stationary point if,*

$$\langle -\nabla_{\boldsymbol{z}}\phi(\boldsymbol{z}), \boldsymbol{z}' - \boldsymbol{z}\rangle \leq \epsilon, \quad \forall \boldsymbol{z}' \in \mathcal{Z}.$$

Similarly, we will define an $\epsilon$-approximate saddle-point for the constrained min-max optimization problem $\min_{\mathcal{X}} \max_{\mathcal{Y}} f(\boldsymbol{x}, \boldsymbol{y})$.

**Definition 2.3** ($\epsilon$-SP)**.** *Let a function $f : \mathcal{X} \times \mathcal{Y} \to \mathbb{R}$. A point $(\boldsymbol{x}, \boldsymbol{y}) \in \mathcal{X} \times \mathcal{Y}$ is said to be an $\epsilon$-approximate saddle-point (or $\epsilon$-FOSP for the min-max problem) if,*

$$-\nabla_{\boldsymbol{x}} f(\boldsymbol{x}, \boldsymbol{y})^\top (\boldsymbol{x}' - \boldsymbol{x}) \leq \epsilon, \forall \boldsymbol{x}' \in \mathcal{X};$$
$$\nabla_{\boldsymbol{y}} f(\boldsymbol{x}, \boldsymbol{y})^\top (\boldsymbol{y}' - \boldsymbol{y}) \leq \epsilon, \forall \boldsymbol{y}' \in \mathcal{Y}.$$

### 2.2 Adversarial Team Markov Games

An adversarial team Markov game is the Markov game extension of normal-form adversarial team games [98]. The game takes place in an infinite-horizon discounted setting where a team of identically-interested players compete against one adversarial player, the *adversary*. We can formally define an adversarial team Markov game as a tuple $\Gamma(\mathcal{S}, [n+1], \mathcal{A}, \mathcal{B}, r, \mathbb{P}, \gamma, \boldsymbol{\rho})$, where:

- $\mathcal{S}$ is the finite set of states, or *state-space*, with cardinality $S := |\mathcal{S}|$;
- $[n+1]$ is the set of players, with the first $n$ players belonging to the team and the last one being the adversary;
- $\mathcal{A} = \bigtimes_{i=1}^{n} \mathcal{A}_i$ is the finite set of the team's joint actions (or, team's *action-space*), while $\mathcal{A}_i$ is the $i$-th player's *action-space*; respectively $\mathcal{B}$ is the adversary's action-space; further, $A := \max_{i \in [n]} |\mathcal{A}_i|$ and $B := |\mathcal{B}|$;
- $r : \mathcal{S} \times \mathcal{A} \times \mathcal{B} \to [0, 1]$ is the adversary's reward function;
- $\mathbb{P} : \mathcal{S} \times \mathcal{A} \times \mathcal{B} \to \Delta(\mathcal{S})$ is transition probability function;
- $\gamma \in [0, 1)$ is the discount factor;
- $\boldsymbol{\rho} \in \Delta(\mathcal{S})$ is the initial state distribution. We assume that $\boldsymbol{\rho}$ is of full-support, $\rho(s) > 0, \forall s \in \mathcal{S}$.

Every team player $i \in [n]$ gets the same reward and the sum of team players' rewards are equal to the adversary's loss, *i.e.*, $\sum_{i=1}^{n} r_i(s, \boldsymbol{a}, b) = -r(s, \boldsymbol{a}, b)$.

### 2.2.1 Policies, Value Function, and Visitation Measures

In this part, we describe policy classes, the value function, and the state-action visitation measures. All of these notions are indispensable for our analysis.

**Policy Definitions.** For any agent $i$, a *stationary* policy $\boldsymbol{\pi}_i$ is defined as a mapping from any given state to a probability distribution over possible actions, where $\boldsymbol{\pi}_i : \mathcal{S} \ni s \mapsto \boldsymbol{\pi}_i(\cdot|s) \in \Delta(\mathcal{A}_i)$. A policy $\boldsymbol{\pi}_i$ is described as *deterministic* when, for any state, it selects a particular action with probability of 1. To simplify, we denote the policy spaces for the team and the adversary as $\Pi_{\text{team}} : \mathcal{S} \to \Delta(\mathcal{A})$ and $\Pi_{\text{adv}} : \mathcal{S} \to \Delta(\mathcal{B})$, respectively. Additionally, the combined policy space for all participants can be represented as $\Pi : \mathcal{S} \to \Delta(\mathcal{A}) \times \Delta(\mathcal{B})$.

**Direct Policy Parametrization.** In the context of our work, we assume the strategy of *direct policy representation* for all players. Specifically, for each player $i$ within the set $[n]$, the policy space $\mathcal{X}_i$ is defined as $\Delta(\mathcal{A}_i)^S$, with $\boldsymbol{\pi}_i = \boldsymbol{x}_i$, such that the probability of choosing action $a$ in state $s$, $x_{i,s,a}$, equals $\pi_i(a|s)$. By the usual game-theoretic convention, $\boldsymbol{\pi}_{-i}$ denotes the policy of all agents apart from $i$. For the adversary, $\mathcal{Y}$ is set as $\Delta(\mathcal{B})^S$, with $\boldsymbol{\pi}_{\text{adv}} = \boldsymbol{y}$, so that $y_{s,a} = \boldsymbol{\pi}_{\text{adv}}(a|s)$.

Having defined policies, we can introduce some standard shortcut notations such as $r(s, \boldsymbol{x}, \boldsymbol{y}) := \mathbb{E}_{(\boldsymbol{a}\, b) \sim (\boldsymbol{x}, \boldsymbol{y})}[r(s, \boldsymbol{a}, b)]$, and the vectors $\boldsymbol{r}(\boldsymbol{x}) \in \mathbb{R}^{|\mathcal{S}| \times |\mathcal{B}|}, \boldsymbol{r}(\boldsymbol{x}, \boldsymbol{y}) \in \mathbb{R}^{|\mathcal{S}|}$ with $\boldsymbol{r}(\boldsymbol{x}) := [\mathbb{E}_{\boldsymbol{a} \sim \boldsymbol{x}}[r(s, \boldsymbol{a}, b)]]_{s,b}$ and $\boldsymbol{r}(\boldsymbol{x}, \boldsymbol{y}) := [\mathbb{E}_{(\boldsymbol{a}, b) \sim \boldsymbol{x}}[r(s, \boldsymbol{a}, b)]]_s$. Further, we define $\mathbb{P}(s'|s, \boldsymbol{x}, \boldsymbol{y})$ as $\mathbb{P}(s'|s, \boldsymbol{x}, \boldsymbol{y}) := \mathbb{E}_{(\boldsymbol{a}, b) \sim (\boldsymbol{x}, \boldsymbol{y})}[\mathbb{P}(s'|s, \boldsymbol{a}, b)]$ and the vector $\mathbb{P}(s, \boldsymbol{x}, b) \in \Delta(\mathcal{S})$ with $\mathbb{P}(s, \boldsymbol{x}, \boldsymbol{y}) := [\mathbb{E}_{(\boldsymbol{a}, b) \sim (\boldsymbol{x}, \boldsymbol{y})}[\mathbb{P}(s'|s, \boldsymbol{a}, b)]]_{s'}$.

**The Value Function.** The *value function* $V_s$, for a given state $s \in \mathcal{S}$, is defined as the adversary's expected total discounted reward over time under a combined policy $(\boldsymbol{\pi}_{\text{team}}, \boldsymbol{\pi}_{\text{adv}})$ from the policy space $\Pi$, with $\boldsymbol{x} = \boldsymbol{\pi}_{\text{team}}$ being the aggregation of policies $(\boldsymbol{\pi}_1, \dots, \boldsymbol{\pi}_n)$. Formally, this is represented as

$$V_s(\boldsymbol{x}, \boldsymbol{y}) := \mathbb{E}_{\boldsymbol{x}, \boldsymbol{y}} \left[ \sum_{h=0}^{\infty} \gamma^h r(s_h, \boldsymbol{a}_h, b_h) \,\middle|\, s_0 = s \right],$$

where the expected value is calculated over the distribution of trajectories generated by the policies $\boldsymbol{x}$ and $\boldsymbol{y}$. If the initial state is instead sampled from a distribution $\boldsymbol{\rho}$, the value function is expressed as $V_{\boldsymbol{\rho}}(\boldsymbol{x}, \boldsymbol{y}) = \mathbb{E}_{s \sim \boldsymbol{\rho}}[V_s(\boldsymbol{x}, \boldsymbol{y})]$.

**Visitation Measures.** The important quantity of state-action visitation measures, or the expected discounted sum of visitations of a state-action pair.

**Definition 2.4** (State-Action Visit. Measure). *For any initial distribution $\boldsymbol{\rho} \in \Delta(\mathcal{S})$, transition matrix $\mathbb{P}$, a team policy $\boldsymbol{x}$, and a policy $\boldsymbol{y} \in \mathcal{Y}$, we define the station-action visitation measure of the adversary $\boldsymbol{\lambda}(\boldsymbol{y}; \boldsymbol{x})$ as follows:*

$$\lambda_{s,b}(\boldsymbol{y}; \boldsymbol{x}) := \sum_{h=0}^{\infty} \gamma^h \, \mathbb{P}(s_h = s, b_h = b | \boldsymbol{x}, \boldsymbol{y}, s_0 \sim \boldsymbol{\rho}).$$

*Where $\lambda_{s,b}(\boldsymbol{y}; \boldsymbol{x})$ denotes the $(s,b)^{th}$ entry of $\boldsymbol{\lambda}(\boldsymbol{y}; \boldsymbol{x})$.*

As we will further discuss in the appendix (Appendix C.1), the correspondence between $\boldsymbol{y}$ and $\boldsymbol{\lambda}$ is "1–1" for a fixed team policy $\boldsymbol{x}$. This property is crucial for our contributions.

**Reformulation of the Value Function.** A key property of the value function $V_{\boldsymbol{\rho}}$ is that it can be rewritten as a concave function of the state-action visitation measure:

$$V_{\boldsymbol{\rho}}(\boldsymbol{x}, \boldsymbol{y}) = \boldsymbol{r}(\boldsymbol{x})^{\top} \boldsymbol{\lambda}(\boldsymbol{y}; \boldsymbol{x}).$$

**Definition 2.5** ($\epsilon$-NE). *A product policy $(\boldsymbol{x}^*, \boldsymbol{y}^*) \in \mathcal{X} \times \mathcal{Y}$ is called an $\epsilon$-approximate Nash equilibrium for an $\epsilon \geq 0$, when*

$$V_{\boldsymbol{\rho}}(\boldsymbol{x}^*, \boldsymbol{y}^*) \leq V_{\boldsymbol{\rho}}((\boldsymbol{x}_i', \boldsymbol{x}_{-i}^*), \boldsymbol{y}^*) + \epsilon, \ \forall \boldsymbol{x}_i' \in \mathcal{X}_i, \ \forall i \in [n];$$

$$and$$

$$V_{\boldsymbol{\rho}}(\boldsymbol{x}^*, \boldsymbol{y}^*) \geq V_{\boldsymbol{\rho}}(\boldsymbol{x}^*, \boldsymbol{y}') - \epsilon, \qquad \forall \boldsymbol{y}' \in \mathcal{Y}.$$

### 2.2.2 The Gradient and Visitation Measure Estimators.

An essential element that led to the development of policy gradient methods is the policy gradient theorem [104]. Notably, it has enabled the design of finite-sample gradient estimators. This technique fits well into the *MARL independent learning protocol* [27]. After all agents have proposed their policy, the MDP is run to acquire batches of trajectories from which all agents will observe the chain's state and their individual reward. These samples are utilized to estimate gradients.

The team agents implement a batch version of the REINFORCE estimator whose definition is deferred to the Appendix C.6.1. As for the estimators that the adversary utilizes, we define the state-action visitation measure estimator and their gradient estimator closely following [113].

**Definition 2.6** (State-Action Visitation Measure Estimator). *Let $\boldsymbol{e}_{s,b}$ be the standard basis for the $(s,b)^{th}$ entry. Let $\tau = (s_0, b_0, s_1, b_1, \cdots, s_{H-1}, b_{H-1})$ denote a trajectory with length $H$ sampled under initial distribution $\boldsymbol{\rho}$ and policy $\boldsymbol{y}$ We define the estimator for $\boldsymbol{\lambda}(\boldsymbol{y}; \boldsymbol{x})$ with the trajectory $\tau$ as the following*

$$\tilde{\boldsymbol{\lambda}}(\tau | \boldsymbol{y}) := \sum_{h=0}^{H-1} \gamma^h \cdot \boldsymbol{e}_{s_h, b_h}.$$

By applying policy gradient theorem [104] along with the chain-rule, the gradient estimator for a value-function that is nonlinear in $\boldsymbol{\lambda}(\boldsymbol{y}; \boldsymbol{x})$, is computed by the following estimator [113].

**Definition 2.7** (Gradient Estimator). *Let $\tau = (s_0, b_0, s_1, b_1, \cdots, s_{H-1}, b_{H-1})$ denote a trajectory with length $H$ sampled under initial distribution $\boldsymbol{\rho}$ and policy $\boldsymbol{y}$. Let $F(\boldsymbol{\lambda}(\boldsymbol{y}))$ be the value function of the MDP w.r.t. $\boldsymbol{\lambda}(\boldsymbol{y})$ and $\boldsymbol{u} := \nabla_{\lambda} F(\boldsymbol{\lambda}(\boldsymbol{y}))$. The estimator for gradient $\nabla_{\boldsymbol{y}} F(\boldsymbol{\lambda}(\boldsymbol{y}))$ using the sampled trajectory $\tau$ is defined as*

$$\tilde{\boldsymbol{g}}(\tau | \boldsymbol{y}; \boldsymbol{u}) := \sum_{h=0}^{H-1} \gamma^h \cdot \boldsymbol{u}(s_h, b_h) \cdot \left( \sum_{h'=0}^{h} \nabla_{\boldsymbol{y}} \log \boldsymbol{y}(b_{h'} | s_{h'}) \right).$$

**Sufficient Exploration.** A standard, while rather naive, technique of bounding the variance of the REINFORCE gradient estimator is using $\zeta$-greedy policy parametrization. Effectively, every action in a player's dispose is played with a probability of at least $\zeta$. For our convenience, we ensure sufficient exploration by a $\zeta$-*truncated simplex* approach. Moreover, for a given feasibility set $\mathcal{X}$, we denote $\mathcal{X}^{\zeta}$ to be the $\zeta$-truncated feasibility set.

## 3 Main Results

We present our main results in two different subsections. In Section 3.1 we manage to attain guarantees for convergence to an approximate stationary-point to constrained nonconvex optimization with an stochastic inexact gradient oracle— we do so by extending previous results of [35]. While in Section 3.2, we apply the latter results along with RL techniques in order to design the first learning algorithm that computes a Nash equilibrium in ATMGs.

## 3.1 Stochastic Weakly-Smooth Nonconvex Optimization with Inexact Gradients

In this subsection we prove that projected gradient descent with a stohcastic inexact gradient oracle converges to an $\epsilon$-FOSP in nonconvex functions with Hölder continuous gradients. We will use this key result in subsequent sections. We begin by defining the inexact gradient oracle and its stochastic version.

**Definition 3.1** (Inexact Gradient Oracle). *Let a differentiable function $\phi(z)$ and its gradient $\nabla\phi(z)$. We call the vector-valued function $g(z)$ a $\vartheta$-inexact gradient oracle if,*

$$\|g(z) - \nabla\phi(z)\| \le \vartheta, \quad \forall z.$$

Further, given a random variable $\xi$ in some sample space $\Xi$, we define a stochastic inexact gradient oracle $G : \mathcal{Z} \times \Xi \to \mathbb{R}^d$. We assume that the expected value of this oracle will be equal to a $\vartheta$-inexact gradient oracle $g(z)$. Additionally to being unbiased (with respect to a $\vartheta$-inexact gradient oracle), we assume its variance to be bounded.

**Assumption 3.1** (Unbiased and Bounded Variance). For a variance parameter $\sigma^2 > 0$, the gradient oracle $G$, satisfies

$$\mathbb{E}_\xi[G(z,\xi)] = g(z) \quad \text{and} \quad \mathbb{E}_\xi\left[\|G(z,\xi) - g(z)\|^2\right] \le \sigma^2.$$

Following, we consider the simple update rule of *Mini-Batch Inexact Stochastic Projected Gradient Descent*, with a batch size $M > 0$ and $\hat{g}^t = \frac{1}{M}\sum_{j=1}^M G\left(z^t, \xi_j^t\right)$,

$$z^{t+1} = \text{Proj}_{\mathcal{Z}}\left(z^t - \eta\hat{g}^t\right). \hspace{3cm} \text{(Inexact Stoch-PGD)}$$

We can now state our convergence Theorem for (Inexact Stoch-PGD) whose proof we defer to the appendix.

**Theorem 3.1** (Convergence to $\epsilon$-FOSP; Formally in Theorem B.1). *Let $\phi : \mathcal{Z} \to \mathbb{R}$ be a Lipschitz continuous function with $(\ell_p, p)$-Hölder continuous gradient and a desired accuracy $\epsilon$. Also, let a stochastic inexact first-order oracle $G$ satisfying Assumption 3.1. The update rule (Inexact Stoch-PGD), with a step-size $\eta = \mathcal{O}\left(\epsilon^{\frac{1-p}{p}}\right)$, computes an $\epsilon$-approximate stationary point after $T = O\left(\epsilon^{-\frac{1+p}{p}}\right)$ iterations.*

## 3.2 Learning Nash Equilibria in Adversarial Team Markov Games

In this subsection we state our contributed Algorithm 1, or ISPNG, which converges to an $\epsilon$-NE for any ATMG, $\Gamma$, with an iteration and sample complexity that scales polynomially with $1/\epsilon$ and the parameters of $\Gamma$. To simply describe the algorithm, the team players initialize their policies and then the following two steps are repeated for $T$ iterations:

1. the adversary approximately maximizes a *regularized version* of their value function, $V_\rho^\nu(x,y) := r(x)^\top\lambda(y;x) - \frac{\nu}{2}\|\lambda(y;x)\|^2$, using Algorithm 2, and then

2. every agent independently performs a gradient descent step on the value function.

During this process, all agents use only bandit feedback information in order to estimate the gradients of the value function. We remark that the learning dynamics remain uncoupled. The only instance of communication between agents is the fact that the team and the adversary take turns when updating their policies. During their turn, the adversary approximately best-responds.

Of particular interest is the sub-routine of Algorithm 2, VIS-REG-PG. It is effectively a directly parameterized policy gradient method for an objective function that is concave in the state-action visitation measure $\lambda(y;x) \in \mathbb{R}^{|\mathcal{S}||\mathcal{B}|}$. The objective function is merely the original value function plus a quadratic term, $-\frac{\nu}{2}\|\lambda(y;x)\|^2$. We remind the reader that due to the existence of this introduced regularizer, the utility of the adversary $u = \nabla_{\lambda(y;x)}F_\rho^\nu(x,y) = r(x) - \nu\lambda(y;x)$. In order to estimate a gradient, the adversary needs to collect a number of trajectories, $\tau = (s_0, b_0, s_1, \ldots, s_{H-1}, b_{H-1}, s_H)$, each of length $H$. Notably, the adversary only uses the empirical state-action visitation measure for the purpose of gradient estimation of the regularized function.

---

**Algorithm 1** Independent Stochastic Policy-Nested-Gradient (ISPNG)

---

**Input:** Accuracy $\epsilon > 0$

1: Based on $\epsilon$, set stepsize $\eta_x$, $T_x$ iterations, batch size $M$, truncation parameter $\zeta_x$, and inner-loop accuracy $\epsilon_y > 0$.                    ▷ see Theorem C.3
2: $x_i^{(0)}(s,a) = \frac{1}{|\mathcal{A}_i|}$, $\forall (s,a) \in \mathcal{S} \times \mathcal{A}_i$.                    ▷ for all agents $i \in [n]$
3: **for** $t \leftarrow 1, 2, \ldots, T_x$ **do**
4:     $\boldsymbol{y}^{(t)} \leftarrow \text{VIS-REG-PG}(\boldsymbol{x}^{(t-1)}, \epsilon_y)$                    ▷ see Algorithm 2
5:     $\hat{\boldsymbol{g}}_i^{(t)} \leftarrow \text{REINFORCE}\left(\boldsymbol{x}^{(t-1)}, \boldsymbol{y}^{(t)}; M\right)$                    ▷ for all agents $i \in [n]$
6:     $\boldsymbol{x}_i^{(t)} \leftarrow \text{Proj}_{\mathcal{X}_i^{\zeta_x}}\left(\boldsymbol{x}_i^{(t-1)} - \eta_x \hat{\boldsymbol{g}}_i^{(t)}\right)$                    ▷ for all agents $i \in [n]$
7: **end for**
8: $\boldsymbol{y}^{(T_x+1)} \leftarrow \text{VIS-REG-PG}(\boldsymbol{x}^{T_x}, \epsilon_y)$
9: $\boldsymbol{x}^* \leftarrow \boldsymbol{x}^{(t^\star)}$                    ▷ pick the best iterate
10: $\boldsymbol{y}^* \leftarrow \boldsymbol{y}^{(t^\star+1)}$

---

**Algorithm 2** Visitation-Regularized Policy Gradient Algorithm (VIS-REG-PG)

---

**Input:** An MDP, a joint strategy of the team $\boldsymbol{x}$, and a desired accuracy $\epsilon > 0$.

1: Based on $\epsilon$, set batch size $K$, sample traj. length $H$, stepsize $\eta_y$, truncation parameter $\zeta_y$ and regularization coeff. $\nu$.                    ▷ see Theorem C.3
2: $y^{(0)}(s,b) \leftarrow \frac{1}{|\mathcal{B}|}$, $\forall (s,b) \in \mathcal{S} \times \mathcal{B}$.
3: **for** Epoch $t \leftarrow 0, 1, \ldots, T_y$ **do**
4:     Independently sample $K$ trajectories, $\mathcal{K}^{(t)}$, of length $H$ under policy $\boldsymbol{y}^{(t)}$.
5:     $\hat{\boldsymbol{\lambda}}^{(t)} \leftarrow \frac{1}{K} \sum_{\tau \in \mathcal{K}^{(t)}} \tilde{\boldsymbol{\lambda}}(\tau | \boldsymbol{y}^{(t)})$,
6:     $\boldsymbol{u} \leftarrow \boldsymbol{r}(\boldsymbol{x}) - \nu \hat{\boldsymbol{\lambda}}^{(t)}$.
7:     $\hat{\boldsymbol{g}}_{\boldsymbol{y}}^{(t)} \leftarrow \frac{1}{K} \sum_{\tau \in \mathcal{K}^{(t)}} \tilde{\boldsymbol{g}}(\tau | \boldsymbol{y}^{(t)}; \boldsymbol{u})$.                    ▷ $\tilde{\boldsymbol{g}}$ as in Definition 2.7.
8:     $\boldsymbol{y}^{(t+1)} \leftarrow \text{Proj}_{\mathcal{Y}^{\zeta_y}}(\boldsymbol{y}^{(t)} + \eta_y \hat{\boldsymbol{g}}_{\boldsymbol{y}}^{(t)})$.
9: **end for**

---

### 3.3 Analyzing Independent Stochastic Policy-Nested-Gradient

Algorithm 1, or ISPNG, is an instance of a nested-loop algorithm. As we have already informally stated, ISPNG runs gradient descent on the regularized max function $\Phi^\nu(\boldsymbol{x}) = \max_{\boldsymbol{\lambda} \in \Lambda(\boldsymbol{x})} \left\{ \boldsymbol{r}(\boldsymbol{x})^\top \boldsymbol{\lambda} - \frac{\nu}{2} \|\boldsymbol{\lambda}\|^2 \right\}$ for some parameter $\nu$. This function has Hölder-continuous gradient and, as such, the convergence proof is underpinned by Theorem 3.1. Formally we state that:

**Theorem 3.2** (Grad. Contunuity of Reg-Max Function). *Let function $\Phi^\nu(\boldsymbol{x})$ be the maximum function of the regularized value function of an ATMG, with regularization coefficient $\nu > 0$. It is the case that, (i) $\Phi^\nu$ is differentiable, (ii) $\nabla_{\boldsymbol{x}} \Phi^\nu$ is $(1/2, \ell_{1/2})$-Hölder continuous, i.e,*

$$\|\nabla_{\boldsymbol{x}} \Phi^\nu(\boldsymbol{x}) - \nabla_{\boldsymbol{x}} \Phi^\nu(\overline{\boldsymbol{x}})\| \leq \ell_{1/2} \|\boldsymbol{x} - \boldsymbol{x}'\|^{\frac{1}{2}}$$

*with $\ell_{1/2} := \frac{30n^{\frac{1}{4}} |\mathcal{S}|^{\frac{5}{4}} \left(\sum_i |\mathcal{A}_i| + |\mathcal{B}|\right)^2}{\nu \min_s \rho(s)(1-\gamma)^{\frac{13}{2}}}$.*

ISPNG manages to run gradient descent on function $\Phi^\nu$ though the agents can never observe the exact gradient of $\Phi^\nu$. This is not only due to the randomness of gradient estimators but mainly because they cannot observe the adversary's actions and thus do not know the gradient w.r.t the regularizing term. Fortunately, the regularization coefficient plays a second role in bounding the inexactness error of the gradient estimates. For that reason, parameter $\nu$ admits a careful tuning.

Finally, the differentiability of $\Phi^\nu$ and the per-player gradient domination property of the $V_{\boldsymbol{\rho}}$ implies that an $\epsilon$-FOSP $\boldsymbol{x}^*$ and the corresponding best-response for the regularized value function, $\boldsymbol{y}^*$, constitute an $\epsilon$-NE, leading to the main Theorem of this subsection:

**Theorem 3.3** (Main Result; Formally in Theorem C.3). *Given a desired accuracy $\epsilon > 0$, Algorithm 1 outputs a joint policy $(\boldsymbol{x}^*, \boldsymbol{y}^*)$ for which it holds that,*

$$\mathbb{E}\left[V_{\boldsymbol{\rho}}(\boldsymbol{x}^*, \boldsymbol{y}^*) - \min_{\boldsymbol{x}_i' \in \mathcal{X}_i} V_{\boldsymbol{\rho}}(\boldsymbol{x}_i', \boldsymbol{x}_{-i}^*, \boldsymbol{y}^*)\right] \leq \epsilon, \quad \forall i \in [n];$$

*and*

$$\mathbb{E}\left[\max_{\boldsymbol{y}' \in \mathcal{Y}} V_{\boldsymbol{\rho}}(\boldsymbol{x}^*, \boldsymbol{y}') - V_{\boldsymbol{\rho}}(\boldsymbol{x}^*, \boldsymbol{y}^*)\right] \leq \epsilon,$$

*with a number of iterations and a number of samples that are* $\mathsf{poly}\left(\frac{1}{\epsilon}, n, |\mathcal{S}|, \sum_i |\mathcal{A}_i| + |\mathcal{B}|, D_{\mathrm{m}}, \frac{1}{1-\gamma}, \frac{1}{\min_s \rho(s)}\right)$. *By $D_{\mathrm{m}}$ we denote the mismatch coefficient $D_{\mathrm{m}} := \left\|\frac{\boldsymbol{d}_{\boldsymbol{\rho}}^{\boldsymbol{x}, \boldsymbol{y}}}{\boldsymbol{\rho}}\right\|_{\infty}$ (Definition C.2).*

## 4 Minimax in Nonconvex–Hidden-Strongly-Concave Functions

Finally, we would state a more general result compared to that of Theorem 3.3. We consider the general min-max nonconvex–nonconcave optimization problem, $\min_{\boldsymbol{x} \in \mathcal{X}} \max_{\boldsymbol{y} \in \mathcal{Y}} f(\boldsymbol{x}, \boldsymbol{y})$, when an additional structural assumption holds, *i.e.*, when $f$ is nonconvex–hidden-strongly-concave. In particular, function $f$ admits a reformulation of the form,

$$H(\boldsymbol{x}, \boldsymbol{u}) := f\left(\boldsymbol{x}, c^{-1}(\boldsymbol{u}; \boldsymbol{x})\right),$$

where function $H$ is a nonconvex–strongly-concave function defined on $\mathcal{X} \times \mathcal{U}$. The sets $\mathcal{X}$ and $\mathcal{U}$ are closed and convex, while $c(\cdot; \boldsymbol{x}) : \mathcal{Y} \to \mathcal{U}$ is an invertible mapping parametrized by $\boldsymbol{x}$. Moreover, we will denote $\mathcal{U}(\boldsymbol{x}) := \{\boldsymbol{u} | \boldsymbol{u} = c(\boldsymbol{y}; \boldsymbol{x}), \forall \boldsymbol{y} \in \mathcal{Y}\}$. We further assume that the mapping $c$ and its inverse are Lipschitz-continuous. Specifically,

**Assumption 4.1.** For the mapping $c$ and its inverse, $c^{-1}$, it holds that

$$\|c(\boldsymbol{y}; \boldsymbol{x}) - c(\boldsymbol{y}'; \boldsymbol{x}')\| \leq L_c(\|\boldsymbol{x} - \boldsymbol{x}'\| + \|\boldsymbol{y} - \boldsymbol{y}'\|), \quad \forall \boldsymbol{x}, \boldsymbol{x}' \in \mathcal{X}; \boldsymbol{y}, \boldsymbol{y}' \in \mathcal{Y}$$
$$\|c^{-1}(\boldsymbol{u}; \boldsymbol{x}) - c^{-1}(\boldsymbol{u}'; \boldsymbol{x}')\| \leq L_{c^{-1}}(\|\boldsymbol{x} - \boldsymbol{x}'\| + \|\boldsymbol{u} - \boldsymbol{u}'\|), \forall \boldsymbol{x}, \boldsymbol{x}' \in \mathcal{X}; \boldsymbol{u}, \boldsymbol{u}' \in \mathcal{U}.$$

If this is the case, the maximizer $\boldsymbol{u}^\star(\boldsymbol{x}) := \operatorname{argmax}_{\boldsymbol{u} \in \mathcal{U}(\boldsymbol{x})} H(\boldsymbol{x}, \boldsymbol{u})$, is Hölder continuous w.r.t. $\boldsymbol{x}$ as stated by the following Theorem.

**Theorem 4.1** (Formally in Theorem D.2). *Let function $f(\boldsymbol{x}, \boldsymbol{y})$ be nonconvex–hidden-strongly-concave with a modulus of $\nu > 0$. Let also function $H$ be a $L_H$-Lipschitz continuous and $\ell_H$-smooth nonconvex–strongly-concave reformulation of $f$ with an invertible mapping $c$ for which Assumption 4.1 holds. Then,*

$$\|\boldsymbol{u}^\star(\boldsymbol{x}) - \boldsymbol{u}^\star(\boldsymbol{x}')\| \leq L_\star \|\boldsymbol{x} - \boldsymbol{x}'\|^{\frac{1}{2}}, \quad \forall \boldsymbol{x}, \boldsymbol{x}' \in \mathcal{X}.$$

*where, $L_\star = O\left(\frac{1}{\nu}\right)$.*

**Theorem 4.2** (Convergence to an $\epsilon$-SP; Formally in Theorem D.3). *Let $f$ be a nonconvex–hidden-strongly-concave function obeying to the same assumptions as $f$ in Theorem 4.1 and $\epsilon > 0$. Further assume a maximization oracle with $O(\nu\epsilon^2)$-accuracy. There exists an algorithm that computes an $\epsilon$-approximate saddle-point $(\boldsymbol{x}^*, \boldsymbol{y}^*)$ by making $T = O\left(\frac{1}{\nu^2\epsilon^3}\right)$ calls to the maximization oracle. Also, the maximization oracle can be implemented by stochastic gradient ascent with iteration complexity $T' = \tilde{O}\left(\frac{1}{\nu^3\epsilon^2}\right)$, and stepsize $\eta_y = O(\nu^2\epsilon^2)$.*

## 5 Conclusion, Future Work, and Limitations

**Conclusions** We expanded stochastic gradient techniques to be able to compute a stationary point in constrained optimization of nonconvex with weakly-smooth functions. We applied that result to design the first learning algorithm that computes an $\epsilon$-approximate Nash equilibrium in adversarial team Markov games using a finite number of samples and iterations that scale polynomially with $1/\epsilon$ and the natural parameters of the game.

**Future Work**  We believe that some questions that require further investigation are the following: (i) Is it possible to extend the techniques of [34] to establish convergence guarantees of stochastic gradient descent on nonconvex functions with Hölder-continuous gradient without batch-sampling of the gradient? (ii) Can we design a two-timescale gradient descent-ascent scheme for ATMGs that converges to a Nash equilibrium with best-iterate guarantees? (iii) Can we utilize some variance-reduction techniques to achieve a better sample complexity for learning an $\epsilon$-NE in ATMGs?

**Limitations**  The main limitations of our work are (i) the notion of independent learning as presented is weaker than the one presented in [27] – *i.e.*, our algorithm has an "inner loop", (ii) the fact that we did not present an example for which the function $\Phi^\nu$ fails to be smooth; hence, it is unclear if we can prove the smoothness of this function and achieve tighter analysis. The first item can be addressed in future work by developing a two-timescale algorithmic approach. As for the second item, we remark that even if it is the case that $\Phi^\nu$ is smooth for ATMGs, our provided convergence rates would be straightforwardly improved without any qualitative modification of the algorithm. Also, we would like to highlight that this discussion is related to Remark 2.

## Acknowledgements

This work has been partially supported by project MIS 5154714 of the National Recovery and Resilience Plan Greece 2.0 funded by the European Union under the NextGenerationEU Program. FK carried over part of the research during an Archimedes Research Internship. IP would like to acknowledge an ICS research award and a startup grant from UCI.

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

# Appendix

## Table of Contents

## A   Further Related Work

We accommodate this section to mention a brief collection of related literature in the fields of team games, reinforcement learning, and optimization. The literature is vast and we can only manage to mention some representative works.

### A.1   Team Games

Research on team games has been a major focus in economic and group decision theory for decades [75, 53, 89, 59]. A key modern reference is [98], which introduced the *team-maxmin equilibrium (TME)* for normal-form games, where the team's strategy maximizes their minimal expected payoff against any adversary response. Despite their optimality, TMEs are computationally intractable even for 3-player team games [54, 11]. Recently, practical algorithms have been developed for multiplayer games [117, 116, 7]. Team equilibria are also relevant to two-player zero-sum games with imperfect recall [87].

Due to TME's intractability, *TMECor*, a relaxed equilibrium concept involving a *correlation device*, has been studied [42, 21, 7, 117, 111, 110, 20]. TMECor permits correlated strategies but can be impractical in certain scenarios [98]. TMECor is also NP-hard for *imperfect-information* extensive-form games (EFGs) [26], although fixed-parameter-tractable (FPT) algorithms have been developed for specific EFG classes [111, 110].

The computational aspects of standard Nash equilibrium (NE) in adversarial team games are not well-understood, even in normal-form games. Von Neumann's *minimax theorem* [102] does not apply to team games, rendering traditional methods ineffective. [94] characterized the *duality gap* between teams, while in [66] it was shown that standard no-regret learning dynamics, such as gradient descent and optimistic Hedge, may fail to converge to mixed NE in binary-action adversarial team games.

## A.2 Reinforcement Learning

**Multiagent RL** Nash equilibrium computation has been central in multiagent RL. Notable algorithms, such as Nash-Q [60, 61], guarantee convergence to Nash equilibria only under strict game conditions. The behavior of independent policy gradient methods [93] remains poorly understood. The impossibility result by the authors of [56] precludes universal convergence to Nash equilibria even in normal-form games, aligning with the computational intractability (PPAD-completeness) of Nash equilibria in two-player general-sum games [28, 24]. Surprisingly, recent work shows similar hardness in turn-based stochastic games, making (stationary) CCEs intractable [29, 64].

Thus, research has focused on specific game classes, like Markov potential games [68, 37, 114, 23, 74, 46] or two-player zero-sum Markov games [27, 103, 91, 22, 92]. As noted, adversarial Markov team games can unify and extend these settings. Identifying multi-agent settings where Nash equilibria are efficiently computable is a key open problem (see, *e.g.*, [27]). Recent guarantees for convergence to Nash equilibria have been found in symmetric games, including symmetric team games [39]. Additionally, weaker solution concepts, relaxing either Markovian or stationarity properties, have gained attention [29, 62].

**Convex RL** Maximizing a value function regularized by a term that is strongly-concave with respect to the state-action visitation measure is an instance of a convex RL problem. In that sense, our work is also related to that strain of literature. Convex RL [108, 112] is a framework that generalizes standard MDP problems by considering the optimization of an objective function that is convex (or concave) in the state (or state-action) visitation measures that the agent's policies induce. The value function of standard RL has an objective function linear to that measure. Common well-known problems that are unified below the lens of convex are (i) "pure-exploration" RL [57], where the agent maximizes the entropy of the state visitation measure, (ii) imitation learning [1], where an agent minimizes the distance of the state visitation measure their policy induces and the one induced by an expert, (iii) risk-averse RL [47] where the agent optimizes an objective function that is sensitive to the tail behavior of the agent and not merely their expected behavior [100, 99, 25, 9, 115, 80], (iv) constrained RL [4], where an agent optimizes their value function while making sure to satisfy a number of constraints that are dependent on their state-action visitation measure [5, 107, 13, 2], (v) diverse skills discovery, where the goal is to drive learning agents to acquire a diverse set of emergent skills [19, 41, 52, 55, 58, 72, 96, 109].

## A.3 Optimization

**Min-max Optimization** Min-max optimization studies problems of the form $\min_{\boldsymbol{x} \in \mathcal{X}} \max_{\boldsymbol{y} \in \mathcal{Y}} f(\boldsymbol{x}, \boldsymbol{y})$. When the objective function $f$ is convex in $\boldsymbol{x}$ and concave in $\boldsymbol{y}$, the corresponding variational inequality (VI) is monotone, and a wide range of algorithms have been proposed for computing an approximate saddle-point — see, *e.g.*, [81, 69].

It is also known that standard Gradient Descent/Ascent (GDA) exhibits time-averaged convergence while the actual trajectory of iterates might cycle [30, 31]. Methods like Extra Gradient or techniques such as optimism are used to ensure convergence [30, 31, 32, 76, 78, 18, 51].

For more general objectives, we know how to compute approximate saddle-points when the weak Minty Property is satisfied [36] and for functions where one (or both) side satisfies the PŁcondition [83, 44, 105]. On the negative side, we know that the problem in its full generality (nonconvex–nonconcave landscape with coupled linear constraints) is computationally intractable [33].

**Hidden-Convex Optimization** This nascent field of optimization [43, 48] considers nonconvex objectives that can be reformulated, through a change of variables, into a convex objective. Further, in the context of game theory, the notion of hidden-monotonicity has made its appearance in [45] and the subsequent works [77, 90].

**Weakly-Smooth Optimization** The majority of references that we encounter for weakly-smooth minimization assume convexity and concern the unconstrained setting. We mention the important references of [35, 82] while also more recent works [106, 85, 84].

# B Nonconvex Weakly-Smooth Constrained Optimization

In this section we prove that stochastic projected gradient descent with an stochastic inexact oracle converges to an $\epsilon$-FOSP in functions with Hölder continuous gradient. We complement this section with the proof of folklore lemmas of constrained optimization that show that the "gradient mapping" (Definition B.1) is an appropriate surrogate of stationarity also for the family of functions we consider.

**Definition B.1** (Gradient Mapping). *We define the gradient mapping and stochastic gradient mapping, $r_\eta$ and $\hat{r}_\eta$, to be:*

- $r_\eta(z) := \frac{1}{\eta}(z - \operatorname{Proj}_{\mathcal{Z}}(z - \eta g(z)))$, *with a shorthand notation,* $r_\eta^t := r_\eta(z)$;

- $\hat{r}_\eta(z) := \frac{1}{\eta}(z - \operatorname{Proj}_{\mathcal{Z}}(z - \eta \hat{g}(z)))$, *similarly,* $\hat{r}_\eta^t := \hat{r}_\eta(z)$ .

## B.1 Auxiliary Lemmas

In general, demonstrating that the gradient mapping is an adequate surrogate of stationarity in differentiable constraint optimization relies on the Lipschitz continuity of the function. We make sure that this is the case when the gradient is only Hölder continuous with $p < 1$.

**Lemma B.1** (Inexact-Gradient Mapping as a Stationarity Surrogate). If $\|r_\eta(z)\| \leq \epsilon$ for some $z \in \mathcal{Z}$, it holds that:

$$\max_{z' \in \mathcal{Z}, \|z' - z^+\| \leq 1} \langle -\nabla\phi(z^+), z' - z^+ \rangle \leq \vartheta + \eta^2 \epsilon + \ell_p \eta^p \epsilon^p,$$

where $z^+ := \operatorname{Proj}_{\mathcal{Z}}(z - \eta g(z))$.

**Proof.** In (Inexact Stoch-PGD), $\|g(z) - \nabla\phi(z)\| \leq \vartheta$, $\forall z \in \mathcal{Z}$. Since $z^+ := \operatorname{Proj}_{\mathcal{Z}}(z - \eta g(z))$, it holds that

$$z^+ = \operatorname*{argmin}_{z' \in \mathcal{Z}} \left\{ \|z' - (z - \eta g(z))\|^2 \right\}.$$

Due to the optimality condition, we have

$$-\left(z^+ - z + \frac{1}{\eta}g(z)\right) \in N_{\mathcal{Z}}(z^+),$$

where $N_{\mathcal{Z}}(z)$ is the normal cone of $\mathcal{Z}$ at $z$, $N_{\mathcal{Z}}(z) := \{v | \langle v, z' - z \rangle \leq 0, \forall z' \in \mathcal{Z}\}$. From the latter, we can conclude that

$$-\left(z^+ - z + \frac{1}{\eta}\nabla\phi(z)\right) - \frac{1}{\eta}(-\nabla\phi(z) + g(z)) \in N_{\mathcal{Z}}(z^+)$$

$$-\left(z^+ - z + \frac{1}{\eta}\nabla\phi(z)\right) \in N_{\mathcal{Z}}(z^+) + B\left(\frac{\vartheta}{\eta}\right)$$

$$-\frac{1}{\eta}\nabla\phi(z^+) - \left(z^+ - z + \frac{1}{\eta}\nabla\phi(z) - \frac{1}{\eta}\nabla\phi(z^+)\right) \in N_{\mathcal{Z}}(z^+) + B\left(\frac{\vartheta}{\eta}\right).$$

Now, we bound $\left\|z^+ - z + \frac{1}{\eta}\nabla\phi(z) - \frac{1}{\eta}\nabla\phi(z^+)\right\|$,

$$\left\|z^+ - z + \frac{1}{\eta}\nabla\phi(z) - \frac{1}{\eta}\nabla\phi(z^+)\right\| \leq \|z^+ - z\| + \frac{1}{\eta}\|\nabla\phi(z) - \nabla\phi(z^+)\|$$

$$\leq \|z^+ - z\| + \frac{\ell_p}{\eta}\|z^+ - z\|^p$$

$$\leq \eta\epsilon + \frac{\ell_p}{\eta^{1-p}}\epsilon^p.$$

Therefore we have

$$-\nabla\phi(z^+) \in N_{\mathcal{Z}}(z^+) + B\left(\vartheta + \eta^2\epsilon + \ell_p\eta^p\epsilon^p\right).$$

The latter display implies the statement of the lemma. $\qquad\square$

We immediately have the following corollary,

**Corollary B.1.** For any $z \in \mathcal{Z}$, denote $z^+ := \operatorname{Proj}_{\mathcal{Z}} (z - \eta g(z))$. $\mathbb{E}\left[\|r_\eta(z)\|\right] \leq \epsilon$ implies that

$$
\mathbb{E}\left[\max_{z' \in \mathcal{Z}, \|z'-z^+\| \leq 1} \langle -\nabla \phi(z^+), z' - z^+ \rangle\right] \leq \vartheta + \eta^2 \epsilon + \ell_p \eta^p \epsilon^p.
$$

## B.2 Stochastic PGD with Inexact Gradients

The folklore proof of gradient descent for nonconvex functions relies on the Lipschitz continuity of the gradient to prove convergence to a first-order stationary point. When the gradient are not Lipschitz continuous but continuous in the weaker notion of Hölder continuity implies the following fact that we will eventually use to prove a "descent lemma".

**Fact B.1.** Let a function $\phi : \mathcal{Z} \to \mathbb{R}$ with $(p, \ell_p)$-Hölder continuous gradient. Then, it is the case that for all $z, z'$,

$$
|\phi(z') - \phi(z) + \langle \nabla f(z), z' - z \rangle| \leq \frac{\ell_p}{1+p} \|z' - z\|^{1+p}.
$$

Following [35], we discuss functions with Hölder-continuous gradient (see Definition 2.1) through the framework of inexact oracle. We show that the answer $(\phi(z), \nabla \phi(z))$ of an exact oracle for a nonconvex function satisfying Hölder gradient continuity can be translated into some "inexact" information for a smooth function. Parameters $\delta, \ell'$ in Proposition B.1 can be treated as "inexactness" parameters and will be chosen as appropriate parameters of the exponent $p$ of Hölder continuity.

**Proposition B.1.** For given $\delta, \ell_p$, and a tuning of $\ell' := \frac{\ell_p^{\frac{2}{1+p}}}{\delta^{\frac{1-p}{1+p}}}$, it holds that for any $x, x'$,

$$
\frac{\ell_p}{1+p} \|x - x'\|^{1+p} \leq \frac{\ell'}{2} \|x - x'\|^2 + \delta.
$$

**Proof.** We let $\chi := \|x - x'\|$. By choosing the optimal $\ell'$ we can verify that

$$
2 \max_{\chi \geq 0} \left\{ \frac{\ell_p}{1+p} \chi^{-1+p} - \delta \chi^{-2} \right\} = \ell_p \left( \frac{\ell_p}{2\delta} \cdot \frac{1-p}{1+p} \right)^{\frac{1-p}{1+p}} \leq \frac{\ell_p^{\frac{2}{1+p}}}{\delta^{\frac{1-p}{1+p}}}.
$$

Where in the inequality we use the fact that $\left( \frac{1-p}{2(1+p)} \right)^{\frac{1-p}{1+p}} \leq 1$ for $0 \leq p \leq 1$. Setting $\ell' = \frac{\ell_p^{\frac{2}{1+p}}}{\delta^{\frac{1-p}{1+p}}}$ yields the desired inequality. $\square$

Now, we can use Proposition B.1 as in place of the "descent-lemma" to Theorem C.3 to prove convergence to an $\epsilon$-FOSP.

**Theorem B.1.** *Let $\phi : \mathcal{Z} \to \mathbb{R}$ be a $(p, \ell_p)$-weakly smooth nonconvex function. Further, assume a stochastic inexact gradient oracle $\hat{g}$. I.e., it holds that $\mathbb{E}\left[\hat{g}(z) - g(z)\right] = 0$ and $\mathbb{E}\left[\|\hat{g}(z) - g(z)\|^2\right] \leq \frac{\sigma^2}{M}$ for some $g : \mathcal{Z} \to \mathcal{Z}^*$ where $\|g(z) - \nabla \phi(z)\| \leq \vartheta, \forall z \in \mathcal{Z}$. Implementing $T$ updates of the form* (Inexact Stoch-PGD) *using $\hat{g}$ and a stepsize $\eta = \frac{1}{2\ell'}$ guarantees that:*

$$
\frac{1}{T} \sum_{t=0}^{T-1} \mathbb{E}\left[\|\hat{r}_\eta^t\|^2\right] \leq \frac{8\ell_p^{\frac{2}{1+p}} \left(\mathbb{E}\left[\phi\left(z^0\right)\right] - \phi^*\right)}{\delta^{\frac{1-p}{1+p}} T} + \frac{8\sigma^2}{M} + 8\ell_p^{\frac{2}{1+p}} \delta^{\frac{2p}{1+p}} + 4\vartheta^2.
$$

We postpone the proof to state a corollary that might help the reader gain some intuition on how the iteration complexity scales with $p$.

**Corollary B.2.** Let $\phi$, $\hat{g}$, the update rule of (Inexact Stoch-PGD) as in Theorem B.1, and stepsize $\eta = (\frac{\epsilon^{1-p}}{2^{3-2p} \cdot \ell_p})^{\frac{1}{p}}$. For $t^*$ drawn uniformly at random from $[1, \ldots, T]$, it holds that:

$$\mathbb{E}\left[\|r_\eta^*\|^2\right] \leq \frac{8\ell_p^{\frac{2}{1+p}} \left(\mathbb{E}\left[\phi\left(z^0\right)\right] - \phi^*\right)}{\delta^{\frac{1-p}{1+p}} T} + 16\ell_p^{\frac{2}{1+p}} \delta^{\frac{2p}{1+p}} + 8\vartheta^2 + \frac{18\sigma^2}{M},$$

where $r_\eta^* := r_\eta^{t^*}$. Furthermore, by setting the parameters as $T \geq \frac{8^{\frac{1+p}{p}} \ell_p^{\frac{1}{p}} \left(\mathbb{E}[\phi(z^0)] - \phi^*\right)}{\epsilon^{\frac{1+p}{p}}}$, $\delta \leq \frac{\left(\frac{\epsilon}{8}\right)^{\frac{1+p}{p}}}{\ell_p^{\frac{1}{p}}}$, $\vartheta \leq \frac{\epsilon}{8}$, and $M \geq \frac{9\sigma^2}{2\epsilon^2}$, it is guaranteed that there will exist a $t^\star \in \{0, \ldots, T-1\}$ such that $\mathbb{E}[r_\eta(z^{t^\star})] \leq \epsilon$.

**Proof.** For the first claim,

$$\mathbb{E}\left[\|r_\eta^*\|^2\right] = \mathbb{E}\left[\left\|\frac{1}{\eta}\left(z^{t^*} - \mathrm{Proj}_{\mathcal{Z}}\left(z^{t^*} - \eta g(z^{t^*})\right)\right)\right\|^2\right]$$

$$\leq 2\mathbb{E}\left[\left\|\frac{1}{\eta}\left(z^{t^*} - \mathrm{Proj}_{\mathcal{Z}}\left(z^{t^*} - \eta\hat{g}(z^{t^*})\right)\right)\right\|^2\right]$$

$$+ 2\mathbb{E}\left[\left\|\frac{1}{\eta}\left(\mathrm{Proj}_{\mathcal{Z}}\left(z^{t^*} - \eta\hat{g}(z^{t^*})\right) - \mathrm{Proj}_{\mathcal{Z}}\left(z^{t^*} - \eta g(z^{t^*})\right)\right)\right\|^2\right]$$

$$\leq 2\mathbb{E}\left[\|\hat{r}_\eta^*\|^2\right] + 2\mathbb{E}\left[\left\|\frac{1}{\eta}\left(z^{t^*} - \eta\hat{g}(z^{t^*}) - z^{t^*} - \eta g(z^{t^*})\right)\right\|^2\right]$$

$$= 2\mathbb{E}\left[\|\hat{r}_\eta^*\|^2\right] + 2\mathbb{E}\left[\left\|\hat{g}(z^{t^*}) - g(z^{t^*})\right\|^2\right]$$

$$\leq \frac{8\ell_p^{\frac{2}{1+p}} \left(\mathbb{E}\left[\phi\left(z^0\right)\right] - \phi^*\right)}{\delta^{\frac{1-p}{1+p}} T} + 16\ell_p^{\frac{2}{1+p}} \delta^{\frac{2p}{1+p}} + 8\vartheta^2 + \frac{18\sigma^2}{M}.$$

Where the last inequality follows from Theorem B.1 and the fact that $\mathbb{E}\left[\|\hat{g}(z) - g(z)\|^2\right] \leq \frac{\sigma^2}{M}$. By setting the parameters as in the corollary, we have $\mathbb{E}[r_\eta(z^{t^\star})] \leq \epsilon$. $\qquad\square$

**Remark 1.** *With the same parameters we choose in Corollary B.2, Lemma B.1 guarantees that for any $p \in (0, 1]$, $r_\eta(z)$ is a sufficient surrogate for stationarity. In particular, $\|r_\eta(z)\| \leq \epsilon$ implies that*

$$-\nabla\phi(z^+) \in N_{\mathcal{Z}}(z^+) + B\left(\left(\left(\frac{8^{1-p}}{\ell_p}\right)^{\frac{2}{p}} + 9\right)\epsilon\right).$$

Finally, we state one more auxiliary claim before proceeding to the proof of Theorem C.3.

**Claim B.1.** Consider an iterate of (Inexact Stoch-PGD), $z^t$. Also, define $z^+ = \mathrm{Proj}_{\mathcal{Z}}\left(z^t - \eta g(z)\right)$, where $g$ is the inexact-gradient oracle. It is the case that,

$$\|z^{t+1} - z^+\| \leq \eta^2 \frac{\sigma^2}{M}.$$

**Proof.** The proof follows easily from arguments we have already used,

$$\mathbb{E}\left[\|z^{t+1} - z^+\|^2\right] = \mathbb{E}\left[\|\mathrm{Proj}_{\mathcal{Z}}\left(z^t - \eta\hat{g}^t\right) - \mathrm{Proj}_{\mathcal{Z}}\left(z^t - \eta g^t\right)\|^2\right]$$

$$\leq \mathbb{E}\left[\|\eta\hat{g}^t - \eta g^t\|^2\right]$$

$$= \eta^2 \mathbb{E}\left[\|\hat{g}^t - g^t\|^2\right]$$

$$\leq \eta^2 \frac{\sigma^2}{M}.$$

$\qquad\square$

**Proof of Theorem B.1**

**Proof.** Since $\|g(z) - \nabla\phi(z)\| \leq \vartheta$, from the weakly-smooth condition, we have

$$\phi\left(z^{t+1}\right) \leq \phi(z^t) + \left\langle \nabla\phi\left(z^t\right), z^{t+1} - z^t \right\rangle + \frac{\ell_p}{1+p}\left\|z^{t+1} - z^t\right\|^{1+p}$$

$$\leq \phi(z^t) + \left\langle \nabla\phi\left(z^t\right), z^{t+1} - z^t \right\rangle + \frac{\ell'}{2}\left\|z^{t+1} - z^t\right\|^2 + \delta \tag{1}$$

$$= \phi(z^t) + \left\langle g\left(z^t\right), z^{t+1} - z^t \right\rangle + \left\langle \nabla\phi(z^t) - g(z^t), z^{t+1} - z^t \right\rangle + \frac{\ell'\eta^2}{2}\left\|\hat{r}_\eta^t\right\|^2 + \delta$$

$$= \phi(z^t) - \eta\left\langle g\left(z^t\right), \hat{r}_\eta^t \right\rangle + \eta\left\langle \nabla\phi(z^t) - g(z^t), \hat{r}_\eta^t \right\rangle + \frac{\ell'\eta^2}{2}\left\|\hat{r}_\eta^t\right\|^2 + \delta \tag{2}$$

$$= \phi(z^t) - \eta\left\langle \hat{g}\left(z^t\right), \hat{r}_\eta^t \right\rangle + \eta\left\langle \hat{g}\left(z^t\right) - g\left(z^t\right), \hat{r}_\eta^t \right\rangle + \eta\left\langle \nabla\phi(z^t) - g(z^t), \hat{r}_\eta^t \right\rangle$$
$$+ \frac{\ell'\eta^2}{2}\left\|\hat{r}_\eta^t\right\|^2 + \delta$$

$$\leq \phi(z^t) - \eta\left\|\hat{r}_\eta^t\right\|^2 + \eta\left\langle \hat{g}\left(z^t\right) - g\left(z^t\right), \hat{r}_\eta^t \right\rangle + \eta\left\langle \nabla\phi(z^t) - g(z^t), \hat{r}_\eta^t \right\rangle$$
$$+ \frac{\ell'\eta^2}{2}\left\|\hat{r}_\eta^t\right\|^2 + \delta \tag{3}$$

$$\leq \phi(z^t) - \eta\left\|\hat{r}_\eta^t\right\|^2 + \eta\left\langle \hat{g}\left(z^t\right) - g\left(z^t\right), \hat{r}_\eta^t \right\rangle + \frac{\eta}{2}\left\|\nabla\phi(z^t) - g(z^t)\right\|^2$$
$$+ \frac{\eta}{2}\left\|\hat{r}_\eta^t\right\|^2 + \frac{\ell'\eta^2}{2}\left\|\hat{r}_\eta^t\right\|^2 + \delta \tag{4}$$

$$\leq \phi(z^t) - \left(\frac{\eta}{2} - \frac{\ell'\eta^2}{2}\right)\left\|\hat{r}_\eta^t\right\|^2 + \eta\left\langle \hat{g}\left(z^t\right) - g\left(z^t\right), \hat{r}_\eta^t \right\rangle + \frac{\eta}{2}\vartheta^2 + \delta \tag{5}$$

$$\leq \phi(z^t) - \left(\frac{\eta}{2} - \frac{\ell'\eta^2}{2}\right)\left\|\hat{r}_\eta^t\right\|^2$$
$$+ \eta\left\langle \hat{g}\left(z^t\right) - g\left(z^t\right), r_\eta^t \right\rangle + \eta\left\langle \hat{g}\left(z^t\right) - g\left(z^t\right), \hat{r}_\eta^t - r_\eta^t \right\rangle + \frac{\eta}{2}\vartheta^2 + \delta.$$

Where

- (1) is because of Proposition B.1;
- in (2), we plug-in the definition of $\hat{r}_\eta^t$;
- (3) uses the fact that $-\left\langle \hat{g}(z^t), \hat{r}_\eta^t \right\rangle \leq -\frac{1}{\eta}\left\|\hat{r}_\eta^t\right\|^2$;
- (4) is due to Young's inequality;
- in (5), we plug-in the error bound on the inexact-gradient oracle $\|\nabla\phi(z) - g(z)\|^2 \leq \vartheta$.

Continuing we have

$$\phi\left(z^{t+1}\right)$$
$$\leq \phi(z^t) - \left(\frac{\eta}{2} - \frac{\ell'\eta^2}{2}\right)\left\|\hat{r}_\eta^t\right\|^2 + \eta\left\langle \hat{g}\left(z^t\right) - g\left(z^t\right), r_\eta^t \right\rangle + \eta\left\langle \hat{g}\left(z^t\right) - g\left(z^t\right), \hat{r}_\eta^t - r_\eta^t \right\rangle$$
$$+ \frac{\eta}{2}\vartheta^2 + \delta$$
$$\leq \phi(z^t) - \left(\frac{\eta}{2} - \frac{\ell'\eta^2}{2}\right)\left\|\hat{r}_\eta^t\right\|^2 + \eta\left\langle \hat{g}\left(z^t\right) - g\left(z^t\right), r_\eta^t \right\rangle + \eta\left\|\hat{g}\left(z^t\right) - g\left(z^t\right)\right\|^2 + \frac{\eta}{2}\vartheta^2 + \delta.$$

Summing for $t = 0, \ldots, T - 1$,

$$\sum_{t=0}^{T-1} \phi\left(\boldsymbol{z}^{t+1}\right)$$

$$\leq \sum_{t=0}^{T-1} \left( \phi\left(\boldsymbol{z}^t\right) - \left(\frac{\eta}{2} - \frac{\ell'\eta^2}{2}\right) \left\|\hat{\boldsymbol{r}}_\eta^t\right\|^2 + \eta \left\langle \hat{\boldsymbol{g}}\left(\boldsymbol{z}^t\right) - \boldsymbol{g}\left(\boldsymbol{z}^t\right), \boldsymbol{r}_\eta^t \right\rangle + \eta \left\|\hat{\boldsymbol{g}}\left(\boldsymbol{z}^t\right) - \boldsymbol{g}\left(\boldsymbol{z}^t\right)\right\|^2 \right)$$

$$+ \frac{\eta}{2}\vartheta^2 T + \delta T.$$

This is equivalent to

$$\sum_{t=0}^{T-1} \left(\frac{\eta}{2} - \frac{\ell'\eta^2}{2}\right) \left\|\hat{\boldsymbol{r}}_\eta^t\right\|^2$$

$$\leq \phi\left(\boldsymbol{z}^0\right) - \phi\left(\boldsymbol{z}^T\right) + \sum_{t=0}^{T-1} \left( \eta \left\langle \hat{\boldsymbol{g}}\left(\boldsymbol{z}^t\right) - \boldsymbol{g}\left(\boldsymbol{z}^t\right), \boldsymbol{r}_\eta^t \right\rangle + \eta \left\|\hat{\boldsymbol{g}}\left(\boldsymbol{z}^t\right) - \boldsymbol{g}\left(\boldsymbol{z}^t\right)\right\|^2 \right)$$

$$+ \frac{\eta}{2}\vartheta^2 T + \delta T$$

$$\leq \phi\left(\boldsymbol{z}^0\right) - \phi^* + \sum_{t=0}^{T-1} \left( \eta \left\langle \hat{\boldsymbol{g}}\left(\boldsymbol{z}^t\right) - \boldsymbol{g}\left(\boldsymbol{z}^t\right), \boldsymbol{r}_\eta^t \right\rangle + \eta \left\|\hat{\boldsymbol{g}}\left(\boldsymbol{z}^t\right) - \boldsymbol{g}\left(\boldsymbol{z}^t\right)\right\|^2 \right) + \frac{\eta}{2}\vartheta^2 T + \delta T.$$

Taking expectations, we have

$$\left(\frac{\eta}{2} - \frac{\ell'\eta^2}{2}\right) \sum_{t=0}^{T-1} \mathbb{E}\left[\left\|\hat{\boldsymbol{r}}_\eta^t\right\|^2\right]$$

$$\leq \mathbb{E}\left[\phi\left(\boldsymbol{z}^0\right)\right] - \phi^* + \sum_{t=0}^{T-1} \left( \eta \mathbb{E}\left[\left\langle \hat{\boldsymbol{g}}\left(\boldsymbol{z}^t\right) - \boldsymbol{g}\left(\boldsymbol{z}^t\right), \boldsymbol{r}_\eta^t \right\rangle\right] + \eta \mathbb{E}\left[\left\|\hat{\boldsymbol{g}}\left(\boldsymbol{z}^t\right) - \boldsymbol{g}\left(\boldsymbol{z}^t\right)\right\|^2\right] \right)$$

$$+ \delta T + \frac{\eta}{2}\vartheta^2 T$$

$$\leq \mathbb{E}\left[\phi\left(\boldsymbol{z}^0\right)\right] - \phi^* + \eta\frac{\sigma^2}{M}T + \delta T + \frac{\eta}{2}\vartheta^2 T.$$

By setting $\eta \leftarrow \frac{1}{2\ell'}$, it holds that

$$\frac{1}{T} \sum_{t=0}^{T-1} \mathbb{E}\left[\left\|\hat{\boldsymbol{r}}_\eta^t\right\|^2\right] \leq \frac{8\ell'\left(\mathbb{E}\left[\phi\left(\boldsymbol{z}^0\right)\right] - \phi^*\right)}{T} + \frac{8\sigma^2}{M} + 8\ell'\delta + 4\vartheta^2$$

$$= \frac{8\ell_p^{\frac{2}{1+p}}\left(\mathbb{E}\left[\phi\left(\boldsymbol{z}^0\right)\right] - \phi^*\right)}{\delta^{\frac{1-p}{1+p}}T} + \frac{8\sigma^2}{M} + 8\ell_p^{\frac{2}{1+p}}\delta^{\frac{2p}{1+p}} + 4\vartheta^2.$$

This completes the proof.

$\square$

# C Adversarial Team Markov Games

In this section we the formal proofs of our claims regarding ATMGs. Before proceeding, let us provide a roadmap of the current section:

- Beginning, in Table 1 we offer a concise summary of our ATMG-related notation.
- We proceed to present a number of crucial facts regarding MDPs in Appendix C.1. In particular, facts regarding the state-action visitation measure.
- In Appendix C.3, we demonstrate that the regularized value function has a unique maximizer that changes in Hölder-continuous way w.r.t. to team policies $x$. This leads to the Hölder-continuity of the gradient of the regularized maximum function $\Phi^\nu$ (see Theorem C.2).
- Having established the latter, in Section 3.2 we invoke the results on gradient descent for nonconvex functions with Hölder continuous gradient to get our main theorem regarding $\epsilon$-NE learning in ATMGs (Theorem C.3).
- The tuning of the parameters of Theorem C.3 is supported by (i) Appendix C.5, where we get precise guarantees for maximizing the regularizing value function w.r.t. the adversary's policy $y$ (Theorem C.4) (ii) Appendix C.6, where we define and analyze the gradient estimators used by the agents of the MG.

Table 1: Notation

**Parameters of the model:**

| | |
|---|---|
| $\mathcal{S}$ | State space |
| $\mathcal{N}$ | Set of players |
| $r$ | Reward function of the adversary |
| $n$ | Number of players in the team |
| $\mathcal{A}_i$ | Action space of player $i$ of the team |
| $\mathcal{A}$ | Team's joint action space |
| $\mathcal{B}$ | Action space of the adversary |
| $A_i$ | Number of actions available to player $i$ of the team |
| $B$ | Number of actions available to the adversary |
| $\mathcal{X}_i$ | The set of feasible directly parameterized policies of player $i$: $\mathcal{X}_i := \Delta(\mathcal{A}_i)^S$ |
| $\mathcal{X}$ | The set of feasible directly parameterized policies of the team: $\mathcal{X} := \bigtimes_{i=1}^n \mathcal{X}_i$ |
| $\mathcal{Y}$ | The set of feasible directly parameterized policies of the adversary player: $\mathcal{Y} := (\mathcal{B})^S$ |
| $\mathbb{P}(s'\|s,\boldsymbol{a},b)$ | Probability of transitioning from state $s$ to $s'$ under the action profile $(\boldsymbol{a}, b)$ |
| $\mathbb{P}(\boldsymbol{x},\boldsymbol{y})$ | The (row-stochastic) transition matrix of the Markov chain induced by $(\boldsymbol{x}, \boldsymbol{y})$ |
| $\gamma$ | Discount factor |
| $d_{s_0}^{\boldsymbol{x},\boldsymbol{y}}(s)$ | The (un-normalized) state visitation measure for policy $(\boldsymbol{x}, \boldsymbol{y})$ |
| $\boldsymbol{\lambda}(\boldsymbol{y};\boldsymbol{x})$ | The state-action visitation measure of the adversary when the team is playing policy $\boldsymbol{x}$ |
| $\boldsymbol{V}(\boldsymbol{x},\boldsymbol{y}), V_{\boldsymbol{\rho}}(\boldsymbol{x},\boldsymbol{y})$ | The value vector per-state, the expected value under initial distribution $\boldsymbol{\rho}$ |
| $\boldsymbol{V}^{\nu}(\boldsymbol{x},\boldsymbol{y}), V_{\boldsymbol{\rho}}^{\nu}(\boldsymbol{x},\boldsymbol{y})$ | The *regularized* value vector per-state, the expected value under initial distribution $\boldsymbol{\rho}$ |
| $\tau$ | A trajectory of states and joint actions of the Markov game |

**Estimators:**

| | |
|---|---|
| $\tilde{\boldsymbol{\lambda}}$ | A single estimate of the state-action visitation measure |
| $\hat{\boldsymbol{\lambda}}$ | The estimator of the state-action visitation measure |
| $\tilde{\boldsymbol{g}}$ | A single estimate of the gradient |
| $\hat{\boldsymbol{g}}_{\boldsymbol{y}}$ | The estimator of the gradient |

**Parameters:**

| | |
|---|---|
| $L_V$ | Lipschitz constant of the value function $V_{\boldsymbol{\rho}}(\cdot,\cdot)$ |
| $\ell_V$ | Smoothness constant of the value function $V_{\boldsymbol{\rho}}(\cdot,\cdot)$ |
| $D_{\mathrm{m}}$ | Distribution mismatch coefficient |

**Additional notation:**

| | |
|---|---|
| $\Phi(\boldsymbol{x})$ | Maximum of the value function given $\boldsymbol{x}$: $\Phi(\boldsymbol{x}) = \max_{\boldsymbol{y} \in \mathcal{Y}} V_{\boldsymbol{\rho}}(\boldsymbol{x},\boldsymbol{y})$ |
| $\Phi^{\nu}(\boldsymbol{x})$ | Maximum of the *regularized* value function given $\boldsymbol{x}$: $\Phi^{\nu}(\boldsymbol{x}) = \max_{\boldsymbol{y} \in \mathcal{Y}} V_{\boldsymbol{\rho}}^{\nu}(\boldsymbol{x},\boldsymbol{y})$ |

### C.1 Further Background on Markov Decision Processes

We need additional preliminaries on Markov decision processes (MDPs). Specifically, we will discuss the properties of the *(discounted) state and state-action visitation measure*. These measures represent the "discounted" expected amount of time the Markov chain—induced by the players' fixed policies—spends at state $s$ (respectively, at a state action pair $(s, b)$) starting from initial state $s'$. Each visit is weighted by a discount factor $\gamma^h$, where $h$ is the visit time. Notably, in [3] it is defined as a probability measure, meaning that for an initial state distribution $\rho$, the discounted state visitation distribution sums to 1. For convenience, we will use the unnormalized definition from [88, Chapter 6.10], which sums to $\frac{1}{1-\gamma}$. This is why we refer to it as a *measure* instead of a *distribution*.

**Definition C.1** (State Visit. Measure). *Given an initial state distribution $\rho \in \Delta(\mathcal{S})$ and a stationary joint policy $\boldsymbol{\pi} \in \Pi$, we define the state visitation frequency $d_{\overline{s}}^{\boldsymbol{\pi}}$ as follows:*

$$d_{\overline{s}}^{\boldsymbol{\pi}}(s) = \sum_{h=0}^{\infty} \gamma^h \, \mathbb{P}(s_h = s | \boldsymbol{\pi}, s_0 = \overline{s}).$$

*Additionally, expanding the definition, we define $d_{\boldsymbol{\rho}}^{\boldsymbol{\pi}}(s) = \mathbb{E}_{\overline{s} \sim \boldsymbol{\rho}} \left[ d_{\overline{s}}^{\boldsymbol{\pi}}(s) \right].$*

For convenience, the expression $d_{\boldsymbol{\rho}}^{\boldsymbol{x}, \boldsymbol{y}}(s)$ is utilized to represent the state visitation measure resulting from the policies $(\boldsymbol{x}, \boldsymbol{y}) \in \mathcal{X} \times \mathcal{Y}$.

**Fact C.1.** Let MDP, $\mathcal{M}(\mathcal{S}, \mathcal{B}, \mathbb{P}, r, \gamma)$. Let a policy $\boldsymbol{\pi} \in \Delta(\mathcal{B})^{|\mathcal{S}|}$. For the corresponding state-action visitation measure $\boldsymbol{\lambda} \in \mathbb{R}^{S \times B}$ and the state visitation measure $\boldsymbol{d}_{\boldsymbol{\rho}}^{\boldsymbol{\pi}} \in \mathbb{R}^S$, it holds that,

$$\lambda_{s,b}(\boldsymbol{\pi}) = d_{\boldsymbol{\rho}}^{\boldsymbol{\pi}}(s) \pi(s, b), \quad \forall s \in \mathcal{S}, \forall b \in \mathcal{B}.$$

A quantity that is important in contemporary RL literature is that of the mismatch coefficient which we formally define here.

**Definition C.2** (Distribution Mismatch Coefficient). *Let $\rho \in \Delta(\mathcal{S})$ be a full-support distribution over states, and $\Pi$ be the joint set of policies. We define the distribution mismatch coefficient $D$ as*

$$D_{\mathrm{m}} := \sup_{\boldsymbol{\pi} \in \Pi} \left\| \frac{\boldsymbol{d}_{\boldsymbol{\rho}}^{\boldsymbol{\pi}}}{\boldsymbol{\rho}} \right\|_{\infty},$$

*where $\frac{\boldsymbol{d}_{\boldsymbol{\rho}}^{\boldsymbol{\pi}}}{\boldsymbol{\rho}}$ denotes element-wise division.*

The following theorem that relates policies and visitation measures is essential to our analysis.

**Theorem C.1** ([88, Theorem 6.9.1]). *Consider an adversarial Markov game $\Gamma$ and a fixed team policy $\boldsymbol{x}$,*

*(i) Any $\boldsymbol{y} \in \mathcal{Y}$ defines a feasible state-action visitation measure $\boldsymbol{\lambda} \in \mathbb{R}^{|\mathcal{S}| \times |\mathcal{B}|}$; namely,*

$$\lambda_{s,b}(\boldsymbol{y}; \boldsymbol{x}) := \sum_{\overline{s} \in \mathcal{S}} \rho(\overline{s}) \cdot \mathbb{E}_{\boldsymbol{y}} \left[ \gamma^t \, \mathbb{P}(s^{(t)} = s, b^{(t)} = b \mid \boldsymbol{x}, s^{(0)} = \overline{s}) \right].$$

*(ii) Any feasible state-action visitation measure $\boldsymbol{\lambda}$ defines a feasible $\boldsymbol{y} \in \mathcal{Y}$; namely,*

$$y_{s,b} := \frac{\lambda(s, b)}{\sum_{b' \in \mathcal{B}} \lambda(s, b')}, \, \forall (s, b) \in \mathcal{S} \times \mathcal{B}.$$

*Further, for any such $\boldsymbol{y} \in \mathcal{Y}$ it holds that $\lambda_{s,b}(\boldsymbol{y}; \boldsymbol{x}) = \lambda(s, b), \, \forall (s, b) \in \mathcal{S} \times \mathcal{B}$, where $\boldsymbol{\lambda}(\boldsymbol{y}; \boldsymbol{x})$ is the induced discounted state-action measure.*

An implication of the latter theorem is the fact that $\boldsymbol{\lambda}(\cdot; \boldsymbol{x})$ is a "1–1" mapping between policies and visitation measures. Following, we see that this mapping is also Lipschitz-continuous and smooth (see Lemmas C.1 to C.3).

**Lemma C.1.** For any initial distribution $\rho \in \Delta(\mathcal{S})$, function $V_{\boldsymbol{\rho}}$ is $L$-Lipschitz and $\ell$-smooth with $L := \frac{\sqrt{\sum_{i=1}^{n} |\mathcal{A}_i| + |\mathcal{B}|}}{(1-\gamma)^2}$ and $\ell := \frac{2\gamma \left( \sum_{i=1}^{n} |\mathcal{A}_i| + |\mathcal{B}| \right)}{(1-\gamma)^3}$, in other words

$$|V_{\boldsymbol{\rho}}(\boldsymbol{x}, \boldsymbol{y}) - V_{\boldsymbol{\rho}}(\boldsymbol{x}', \boldsymbol{y}')| \leq \frac{\sqrt{\sum_{i=1}^{n} |\mathcal{A}_i| + |\mathcal{B}|}}{(1-\gamma)^2} \|(\boldsymbol{x}, \boldsymbol{y}) - (\boldsymbol{x}', \boldsymbol{y}')\|;$$

$$\|\nabla V_{\boldsymbol{\rho}}(\boldsymbol{x}, \boldsymbol{y}) - \nabla V_{\boldsymbol{\rho}}(\boldsymbol{x}', \boldsymbol{y}')\| \leq \frac{2\gamma \left( \sum_{i=1}^{n} |\mathcal{A}_i| + |\mathcal{B}| \right)}{(1-\gamma)^3} \|(\boldsymbol{x}, \boldsymbol{y}) - (\boldsymbol{x}', \boldsymbol{y}')\|.$$

**Proof.** The proof follows from Lemma 4.4 in [68]. □

**Lemma C.2.** Let $\boldsymbol{\lambda} \in \mathbb{R}^{|\mathcal{S}||\mathcal{B}|}$ be the state-action visitation measure for the adversary, then $\boldsymbol{\lambda}$ is $L_\lambda$-Lipschitz continuous and $\ell_\lambda$-smooth w.r.t to policy $(\boldsymbol{x}, \boldsymbol{y})$. Specifically, we have

$$\|\boldsymbol{\lambda}(\boldsymbol{y};\boldsymbol{x}) - \boldsymbol{\lambda}(\boldsymbol{y}';\boldsymbol{x}')\| \leq \frac{|\mathcal{S}|^{\frac{1}{2}}\left(\sum_i |\mathcal{A}_i| + |\mathcal{B}|\right)}{(1-\gamma)^2}\left(\|\boldsymbol{x} - \boldsymbol{x}'\| + \|\boldsymbol{y} - \boldsymbol{y}'\|\right),$$

and

$$\|\nabla\boldsymbol{\lambda}(\boldsymbol{y};\boldsymbol{x}) - \nabla\boldsymbol{\lambda}(\boldsymbol{y}';\boldsymbol{x}')\| \leq \frac{2|\mathcal{S}|^{\frac{1}{2}}\left(\sum_i |\mathcal{A}_i| + |\mathcal{B}|\right)^{\frac{3}{2}}}{(1-\gamma)^3}\left(\|\boldsymbol{x} - \boldsymbol{x}'\| + \|\boldsymbol{y} - \boldsymbol{y}'\|\right).$$

**Proof.** Each $\lambda_{s,b}$ can be considered as a value function for the given state $s$ and the reward function is $r(a', b') = \mathbb{1}(b = b')$. Then by applying Lemma C.1, we have

$$|\lambda_{s,b}(\boldsymbol{y};\boldsymbol{x}) - \lambda_{s,b}(\boldsymbol{y}';\boldsymbol{x}')| \leq \frac{\sqrt{\sum_i |\mathcal{A}_i| + |\mathcal{B}|}}{(1-\gamma)^2} \cdot \left(\|\boldsymbol{x} - \boldsymbol{x}'\| + \|\boldsymbol{y} - \boldsymbol{y}'\|\right), \tag{6}$$

$$\|\nabla\lambda_{s,b}(\boldsymbol{y};\boldsymbol{x}) - \nabla\lambda_{s,b}(\boldsymbol{y}';\boldsymbol{x}')\| \leq \frac{2\gamma\left(\sum_i |\mathcal{A}_i| + |\mathcal{B}|\right)}{(1-\gamma)^3} \cdot \left(\|\boldsymbol{x} - \boldsymbol{x}'\| + \|\boldsymbol{y} - \boldsymbol{y}'\|\right). \tag{7}$$

From Equation (6) we get

$$\|\boldsymbol{\lambda}(\boldsymbol{y};\boldsymbol{x}) - \boldsymbol{\lambda}(\boldsymbol{y}';\boldsymbol{x}')\|_\infty \leq \frac{\sqrt{\sum_i |\mathcal{A}_i| + |\mathcal{B}|}}{(1-\gamma)^2} \cdot \left(\|\boldsymbol{x} - \boldsymbol{x}'\| + \|\boldsymbol{y} - \boldsymbol{y}'\|\right).$$

This implies

$$\begin{aligned}
\|\boldsymbol{\lambda}(\boldsymbol{y};\boldsymbol{x}) - \boldsymbol{\lambda}(\boldsymbol{y}';\boldsymbol{x}')\| &\leq \sqrt{|\mathcal{S}||\mathcal{B}|}\,\|\boldsymbol{\lambda}(\boldsymbol{y};\boldsymbol{x}) - \boldsymbol{\lambda}(\boldsymbol{y}';\boldsymbol{x}')\|_\infty \\
&\leq \frac{\sqrt{|\mathcal{S}||\mathcal{B}|}\sqrt{\sum_i |\mathcal{A}_i| + |\mathcal{B}|}}{(1-\gamma)^2} \cdot \left(\|\boldsymbol{x} - \boldsymbol{x}'\| + \|\boldsymbol{y} - \boldsymbol{y}'\|\right) \\
&\leq \frac{\sqrt{|\mathcal{S}|}\left(\sum_i |\mathcal{A}_i| + |\mathcal{B}|\right)}{(1-\gamma)^2} \cdot \left(\|\boldsymbol{x} - \boldsymbol{x}'\| + \|\boldsymbol{y} - \boldsymbol{y}'\|\right).
\end{aligned}$$

Similarly, from Equation (7), we have

$$\|\nabla\lambda_{s,b}(\boldsymbol{y};\boldsymbol{x}) - \nabla\lambda_{s,b}(\boldsymbol{y}';\boldsymbol{x}')\| \leq \frac{2\gamma\left(\sum_i |\mathcal{A}_i| + |\mathcal{B}|\right)}{(1-\gamma)^3} \cdot \left(\|\boldsymbol{x} - \boldsymbol{x}'\| + \|\boldsymbol{y} - \boldsymbol{y}'\|\right).$$

Thus

$$\begin{aligned}
\|\nabla\boldsymbol{\lambda}(\boldsymbol{y};\boldsymbol{x}) - \nabla\boldsymbol{\lambda}(\boldsymbol{y}';\boldsymbol{x}')\|_F &\leq \sqrt{|\mathcal{S}||\mathcal{B}|} \cdot \max_{s\in\mathcal{S}, b\in\mathcal{B}} \|\nabla\lambda_{s,b}(\boldsymbol{y};\boldsymbol{x}) - \nabla\lambda_{s,b}(\boldsymbol{y}';\boldsymbol{x}')\| \\
&\leq \frac{2\gamma\sqrt{|\mathcal{S}||\mathcal{B}|}\left(\sum_i |\mathcal{A}_i| + |\mathcal{B}|\right)}{(1-\gamma)^3} \cdot \left(\|\boldsymbol{x} - \boldsymbol{x}'\| + \|\boldsymbol{y} - \boldsymbol{y}'\|\right).
\end{aligned}$$

Where $\|\cdot\|_F$ denotes the Frobenius norm of the matrix. Finally, we have

$$\begin{aligned}
\|\nabla\boldsymbol{\lambda}(\boldsymbol{y};\boldsymbol{x}) - \nabla\boldsymbol{\lambda}(\boldsymbol{y}';\boldsymbol{x}')\| &\leq \|\nabla\boldsymbol{\lambda}(\boldsymbol{y};\boldsymbol{x}) - \nabla\boldsymbol{\lambda}(\boldsymbol{y}';\boldsymbol{x}')\|_F \\
&\leq \frac{2\gamma\sqrt{|\mathcal{S}||\mathcal{B}|}\left(\sum_i |\mathcal{A}_i| + |\mathcal{B}|\right)}{(1-\gamma)^3} \cdot \left(\|\boldsymbol{x} - \boldsymbol{x}'\| + \|\boldsymbol{y} - \boldsymbol{y}'\|\right) \\
&\leq \frac{2\gamma\sqrt{|S|\left(\sum_i |\mathcal{A}_i| + |\mathcal{B}|\right)^3}}{(1-\gamma)^3} \cdot \left(\|\boldsymbol{x} - \boldsymbol{x}'\| + \|\boldsymbol{y} - \boldsymbol{y}'\|\right) \\
&\leq \frac{2\sqrt{|S|\left(\sum_i |\mathcal{A}_i| + |\mathcal{B}|\right)^3}}{(1-\gamma)^3} \cdot \left(\|\boldsymbol{x} - \boldsymbol{x}'\| + \|\boldsymbol{y} - \boldsymbol{y}'\|\right).
\end{aligned}$$

□

**Lemma C.3.** Consider $\lambda_{\text{inv}}(\boldsymbol{\lambda}(\boldsymbol{y};\boldsymbol{x})) := \frac{\lambda_{s,b}(\boldsymbol{y};\boldsymbol{x})}{\sum_{b'}\lambda_{s,b'}(\boldsymbol{y};\boldsymbol{x})}$, which is a function that maps the adversary's state-action visitation measure $\boldsymbol{\lambda}(\boldsymbol{y};\boldsymbol{x}) \in \Lambda(\boldsymbol{x})$ to the adversary's policy $\boldsymbol{y} \in \mathcal{Y}$. For any fixed team policy $\boldsymbol{x}$, $\lambda_{\text{inv}}$ is $L_{\lambda_{\text{inv}}}$-Lipschitz continuous with respect $\boldsymbol{\lambda}$ where $L_{\lambda_{\text{inv}}} = \max_{s\in\mathcal{S}}\frac{2}{\rho(s)(1-\gamma)}$. Specifically, it holds that

$$\|\boldsymbol{y} - \boldsymbol{y}'\| \leq L_{\lambda_{\text{inv}}}\|\boldsymbol{\lambda}(\boldsymbol{y};\boldsymbol{x}) - \boldsymbol{\lambda}(\boldsymbol{y}';\boldsymbol{x})\|.$$

**Proof.** Take the partial derivative of $\lambda_{\text{inv}}(\boldsymbol{\lambda})$, we have

$$\left|\frac{\partial}{\partial\lambda_{s,b}}\frac{\lambda_{s,b}}{\sum_{b'}\lambda_{s,b'}}\right| = \left|\frac{1}{\sum_{b'}\lambda_{s,b'}} - \frac{\lambda_{s,b}}{(\sum_{b'}\lambda_{s,b'})^2}\right|$$

$$\leq \left|\frac{1}{\sum_{b'}\lambda_{s,b'}}\right| + \left|\frac{\lambda_{s,b}}{(\sum_{b'}\lambda_{s,b'})^2}\right|$$

$$\leq \max_{s\in\mathcal{S}}\left\{\left|\frac{1}{\rho(s)}\right| + \frac{1}{\rho(s)(1-\gamma)}\right\}$$

$$\leq \max_{s\in\mathcal{S}}\frac{2}{\rho(s)(1-\gamma)}.$$

This implies the Lipschitz continuity. $\qquad\square$

**The Regularized Value Function.**

**Lemma C.4.** Function $V_\rho^\nu(\boldsymbol{x},\boldsymbol{y}) := V_\rho^\nu(\boldsymbol{x},\boldsymbol{y}) - \frac{\nu}{2}\|\boldsymbol{\lambda}(\boldsymbol{y};\boldsymbol{x})\|^2$ is $L_\nu$-Lipschitz continuous and $\ell_\nu$-smooth, where $L_\nu := L + \frac{\nu L_\lambda}{2(1-\gamma)}$ and $\ell_\nu := \ell + \frac{\nu\ell_\lambda}{2(1-\gamma)} + \frac{\nu L_\lambda^2}{2}$.

**Proof.** For Lipschitz continuity, we have

$$\left|V_\rho^\nu(\boldsymbol{x},\boldsymbol{y}) - V_\rho^\nu(\boldsymbol{x}',\boldsymbol{y}')\right|$$

$$\leq |V_\rho(\boldsymbol{x},\boldsymbol{y}) - V_\rho(\boldsymbol{x}',\boldsymbol{y}')| + \frac{\nu}{2}\left|\|\boldsymbol{\lambda}(\boldsymbol{y};\boldsymbol{x})\|^2 - \|\boldsymbol{\lambda}(\boldsymbol{y}';\boldsymbol{x}')\|^2\right|$$

$$\leq |V_\rho(\boldsymbol{x},\boldsymbol{y}) - V_\rho(\boldsymbol{x}',\boldsymbol{y}')| + \frac{\nu}{2}\max_{\boldsymbol{x},\boldsymbol{y}}\|\boldsymbol{\lambda}(\boldsymbol{y};\boldsymbol{x})\|\cdot\left|\|\boldsymbol{\lambda}(\boldsymbol{y};\boldsymbol{x})\| - \|\boldsymbol{\lambda}(\boldsymbol{y}';\boldsymbol{x}')\|\right|$$

$$\leq |V_\rho(\boldsymbol{x},\boldsymbol{y}) - V_\rho(\boldsymbol{x}',\boldsymbol{y}')| + \frac{\nu}{2}\cdot\frac{1}{1-\gamma}\|\boldsymbol{\lambda}(\boldsymbol{y};\boldsymbol{x}) - \boldsymbol{\lambda}(\boldsymbol{y}';\boldsymbol{x}')\|$$

$$\leq \left(L + \frac{\nu L_\lambda}{2(1-\gamma)}\right)\cdot(\|\boldsymbol{x} - \boldsymbol{x}'\| + \|\boldsymbol{y} - \boldsymbol{y}'\|).$$

For smoothness, denote the Jacobian matrix of $\boldsymbol{\lambda}(\boldsymbol{y};\boldsymbol{x})$ w.r.t to $(\boldsymbol{x},\boldsymbol{y})$ by $\mathbf{J}_\lambda(\boldsymbol{x},\boldsymbol{y})$, it holds that

$$\left\|\nabla_{\boldsymbol{x}}\|\boldsymbol{\lambda}(\boldsymbol{y};\boldsymbol{x})\|^2 - \nabla_{\boldsymbol{x}}\|\boldsymbol{\lambda}(\boldsymbol{y}';\boldsymbol{x}')\|^2\right\|$$

$$= \left\|\boldsymbol{\lambda}(\boldsymbol{y};\boldsymbol{x})^\top\mathbf{J}_\lambda(\boldsymbol{x},\boldsymbol{y}) - \boldsymbol{\lambda}(\boldsymbol{y}';\boldsymbol{x}')^\top\mathbf{J}_\lambda(\boldsymbol{x}',\boldsymbol{y}')\right\|$$

$$\leq \left\|\boldsymbol{\lambda}(\boldsymbol{y};\boldsymbol{x})^\top\mathbf{J}_\lambda(\boldsymbol{x},\boldsymbol{y}) - \boldsymbol{\lambda}(\boldsymbol{y};\boldsymbol{x})^\top\mathbf{J}_\lambda(\boldsymbol{x}',\boldsymbol{y}')\right\|$$

$$\quad + \left\|\boldsymbol{\lambda}(\boldsymbol{y};\boldsymbol{x})^\top\mathbf{J}_\lambda(\boldsymbol{x}',\boldsymbol{y}') - \boldsymbol{\lambda}(\boldsymbol{y}';\boldsymbol{x}')^\top\mathbf{J}_\lambda(\boldsymbol{x}',\boldsymbol{y}')\right\|$$

$$\leq \|\boldsymbol{\lambda}(\boldsymbol{y};\boldsymbol{x})\|\|\mathbf{J}_\lambda(\boldsymbol{x},\boldsymbol{y}) - \mathbf{J}_\lambda(\boldsymbol{x}',\boldsymbol{y}')\| + \|\boldsymbol{\lambda}(\boldsymbol{y};\boldsymbol{x}) - \boldsymbol{\lambda}(\boldsymbol{y}';\boldsymbol{x}')\|\|\mathbf{J}_\lambda(\boldsymbol{x}',\boldsymbol{y}')\|$$

$$\leq \frac{1}{1-\gamma}\|\mathbf{J}_\lambda(\boldsymbol{x},\boldsymbol{y}) - \mathbf{J}_\lambda(\boldsymbol{x}',\boldsymbol{y}')\| + \|\boldsymbol{\lambda}(\boldsymbol{y};\boldsymbol{x}) - \boldsymbol{\lambda}(\boldsymbol{y}';\boldsymbol{x}')\|\|\mathbf{J}_\lambda(\boldsymbol{x}',\boldsymbol{y}')\| \qquad(8)$$

$$\leq \left(\frac{\ell_\lambda}{1-\gamma} + L_\lambda\cdot L_\lambda\right)\cdot(\|\boldsymbol{x} - \boldsymbol{x}'\| + \|\boldsymbol{y} - \boldsymbol{y}'\|).$$

Where in (8) we used the fact that $\|\mathbf{J}_{\boldsymbol{\lambda}}(\boldsymbol{x}', \boldsymbol{y}')\| \le L_\lambda$. We conclude that

$$
\begin{aligned}
&\left\| \nabla V_{\boldsymbol{\rho}}^{\nu}(\boldsymbol{x}, \boldsymbol{y}) - \nabla V_{\boldsymbol{\rho}}^{\nu}(\boldsymbol{x}', \boldsymbol{y}') \right\| \\
&= \left\| \nabla V_{\boldsymbol{\rho}}(\boldsymbol{x}, \boldsymbol{y}) - \nabla V_{\boldsymbol{\rho}}(\boldsymbol{x}', \boldsymbol{y}') + \frac{\nu}{2} \left( \nabla_{\boldsymbol{x}} \|\boldsymbol{\lambda}(\boldsymbol{y}; \boldsymbol{x})\|^2 - \nabla_{\boldsymbol{x}} \|\boldsymbol{\lambda}(\boldsymbol{y}'; \boldsymbol{x}')\|^2 \right) \right\| \\
&\le \left\| \nabla V_{\boldsymbol{\rho}}(\boldsymbol{x}, \boldsymbol{y}) - \nabla V_{\boldsymbol{\rho}}(\boldsymbol{x}', \boldsymbol{y}') \right\| + \frac{\nu}{2} \left\| \nabla_{\boldsymbol{x}} \|\boldsymbol{\lambda}(\boldsymbol{y}; \boldsymbol{x})\|^2 - \nabla_{\boldsymbol{x}} \|\boldsymbol{\lambda}(\boldsymbol{y}'; \boldsymbol{x}')\|^2 \right\| \\
&\le \left( \ell + \frac{\nu \ell_\lambda}{2(1-\gamma)} + \frac{\nu L_\lambda^2}{2} \right) \cdot (\|\boldsymbol{x} - \boldsymbol{x}'\| + \|\boldsymbol{y} - \boldsymbol{y}'\|).
\end{aligned}
$$

$\square$

Finally, we compute the Lipschitz continuity parameter of the reward vector that we already used in our previous claims.

**Lemma C.5.** Let $\boldsymbol{r}(\boldsymbol{x})$ be the reward function for the adversary when the team is playing policy $\boldsymbol{x}$. Then it holds that

$$
\|\boldsymbol{r}(\boldsymbol{x}) - \boldsymbol{r}(\boldsymbol{x}')\| \le L_r \|\boldsymbol{x} - \boldsymbol{x}'\|,
$$

where $L_r = \sqrt{S} \left( \sum_{i=1}^n |\mathcal{A}_i| + B \right)$.

**Proof.**

$$
\begin{aligned}
\|\boldsymbol{r}(\boldsymbol{x}) - \boldsymbol{r}(\boldsymbol{x}')\| &\le \sqrt{|\mathcal{S}||\mathcal{B}|} \|r(\boldsymbol{x}) - r(\boldsymbol{x}')\|_\infty \\
&= \sqrt{|\mathcal{S}||\mathcal{B}|} \max_{s,b} |r(s, \boldsymbol{x}, b) - r(s, \boldsymbol{x}', b)| \\
&= \sqrt{|\mathcal{S}||\mathcal{B}| \sum_{i=1}^n |\mathcal{A}_i|} \|\boldsymbol{x} - \boldsymbol{x}'\| \qquad (9)\\
&\le \sqrt{|\mathcal{S}|} \left( \sum_{i=1}^n |\mathcal{A}_i| + B \right) \|\boldsymbol{x} - \boldsymbol{x}'\|.
\end{aligned}
$$

Where (9) follows from Claim D.9. in [65]. $\square$

## C.2 Auxiliary Lemmas

**Bounding the stationarity error on the truncated simplex.** The $\zeta$-truncated simplex, $\Delta^{m,\zeta}$ is defined a the set of all probability vectors with no entry smaller than $\zeta > 0$. More formally, for a given dimension $m$ and a $0 < \zeta \le \frac{1}{m}$, the $\zeta$-truncated simplex is defined to be

$$
\Delta^{m,\zeta} = \left\{ \boldsymbol{x} \ \middle| \ x_i \ge \zeta, \sum_{i=1}^m x_i = 1 \right\}.
$$

**Lemma C.6.** [40, Lemma 15] Let $\Delta^{m,\zeta}$ be the $\zeta$-truncated $m$-simplex. If $0 \le \zeta \le \frac{1}{2m}$, then for all $\boldsymbol{x} \in \Delta^m$, there exists a $\boldsymbol{x}_\zeta \in \Delta^{m,\zeta}$ such that $\|\boldsymbol{x} - \boldsymbol{x}_\zeta\| \le 2\zeta m$.

**Proposition C.1** (Stationarity on the trunc. simplex). Let an $L_f$-Lipschitz continuous differentiable function $f : \Delta^m \to \mathbb{R}$. Also, let an $\epsilon$-approximate stationary point $\boldsymbol{x}_\zeta$ when the feasibility set is the truncated simplex $\Delta^{m,\zeta}$ such that

$$
\left\langle -\nabla f(\boldsymbol{x}_\zeta), \boldsymbol{x}'_\zeta - \boldsymbol{x}_\zeta \right\rangle \le \epsilon, \quad \forall \boldsymbol{x}'_\zeta \in \Delta^{m,\zeta}.
$$

Then, $\boldsymbol{x}_\zeta$ is an $(\epsilon + 2\zeta m L_f)$-stationary point when the entire simplex is considered, *i.e*,

$$
\left\langle -\nabla f(\boldsymbol{x}_\zeta), \boldsymbol{x} - \boldsymbol{x}_\zeta \right\rangle \le \epsilon + 2\zeta m L_f, \quad \forall \boldsymbol{x} \in \Delta^m.
$$

**Proof.** Consider $\boldsymbol{x}'_\zeta \in \Delta^{m,\zeta}$ such that $\left\| \boldsymbol{x} - \boldsymbol{x}'_\zeta \right\| \le 2\zeta m$, such point exists due to Lemma C.6, we have

$$
\begin{aligned}
\left\langle -\nabla f(\boldsymbol{x}_\zeta), \boldsymbol{x} - \boldsymbol{x}_\zeta \right\rangle &= \left\langle -\nabla f(\boldsymbol{x}_\zeta), \boldsymbol{x}'_\zeta - \boldsymbol{x}_\zeta \right\rangle + \left\langle -\nabla f(\boldsymbol{x}_\zeta), \boldsymbol{x} - \boldsymbol{x}'_\zeta \right\rangle \\
&\le \epsilon + 2\zeta m L_f.
\end{aligned}
$$

Where in the last inequality we use the fact that for all $\boldsymbol{x}'_\zeta \in \Delta^{m,\zeta}$, we have $\left\langle -\nabla f(\boldsymbol{x}_\zeta), \boldsymbol{x}'_\zeta - \boldsymbol{x}_\zeta \right\rangle \leq \epsilon$ and $\|\nabla f(\boldsymbol{x}_\zeta)\| \leq L_f$. $\qquad\square$

**From stationarity to optimality.**

**Lemma C.7** (Gradient Domination). Let a single-agent MDP with action-space $\mathcal{A}$ and directly-parametrized policy $\boldsymbol{x} \in \Delta^\zeta (\mathcal{A})^{|\mathcal{S}|}$. Then it holds that

$$\max_{\boldsymbol{x}^\star \in \Delta(\mathcal{A})^{|\mathcal{S}|}} V_{\boldsymbol{\rho}}(\boldsymbol{x}^\star) - V_{\boldsymbol{\rho}}(\boldsymbol{x}) \leq \frac{1}{1-\gamma} D_{\mathrm{m}} \max_{\boldsymbol{x}' \in \Delta^\zeta(\mathcal{A})^{|\mathcal{S}|}} (\boldsymbol{x}' - \boldsymbol{x})^\top \nabla_{\boldsymbol{x}} V_{\boldsymbol{\rho}}(\boldsymbol{x}) + \frac{2D_{\mathrm{m}}\zeta|\mathcal{S}||\mathcal{A}|L}{1-\gamma}.$$

**Proof.** The proof follows easily from Proposition C.1 and the gradient domination property [3, Lemma 4.1]. $\qquad\square$

### C.3 Continuity of the maximizers

We begin this section by firstly introducing a proposition which we will leverage in Lemma C.8.

**Proposition C.2.** Consider the inequality of the form $\alpha\lambda^2 \leq \beta\lambda\chi + \gamma\chi$ where $\lambda$ and $\chi$ are variables and $\alpha, \beta, \gamma$ are coefficients. Under the constraints that $\alpha, \beta, \gamma, \lambda, \chi \geq 0$, there is no solution of the form $\lambda \leq c\chi$ for any finite constant $c \geq 0$.

**Proof.** Solving the quadratic inequality for $\lambda$ gives:

$$0 \leq \lambda \leq \frac{\beta\chi + \sqrt{\chi(4\alpha\gamma + \beta^2\chi)}}{2\alpha}.$$

We search for a positive constant $c$ such that $\lambda \leq \frac{\beta\chi + \sqrt{\chi(4\alpha\gamma + \beta^2\chi)}}{2\alpha} \leq c\chi$, or equivalently,

$$\frac{1}{2\alpha}\left(\beta + \sqrt{\frac{4\alpha\gamma}{\chi} + \beta^2}\right) \leq c.$$

By observing that when $\alpha, \beta, \gamma$ are all positive constants and as $\chi \to 0$, $c \to \infty$, we conclude that no such finite constant $c$ exists. $\qquad\square$

We first define the maximizer for the regularized function $\boldsymbol{r}(\boldsymbol{x})^\top \boldsymbol{\lambda} - \frac{\nu}{2}\|\boldsymbol{\lambda}\|^2$. Since the function is strongly-concave w.r.t. $\boldsymbol{\lambda}$, the maximizing $\boldsymbol{\lambda}$ is unique. Specifically we denote

$$\boldsymbol{\lambda}^\star(\boldsymbol{x}) := \operatorname*{argmax}_{\boldsymbol{\lambda} \in \Lambda(\boldsymbol{x})} \left\{\boldsymbol{r}(\boldsymbol{x})^\top \boldsymbol{\lambda} - \frac{\nu}{2}\|\boldsymbol{\lambda}\|^2\right\}. \tag{10}$$

Now we are ready to show an important lemma regarding to $\boldsymbol{\lambda}^\star(\boldsymbol{x})$.

**Lemma C.8** (Continuity of the max. of reg. functions). For any adversarial Markov game $\Gamma$, $\boldsymbol{\lambda}^\star(\boldsymbol{x})$ defined in (10) is $(1/2, L_\star)$-Hölder continuous, specifically

$$\|\boldsymbol{\lambda}^*(\boldsymbol{x}_1) - \boldsymbol{\lambda}^*(\boldsymbol{x}_2)\| \leq L_\star \|\boldsymbol{x}_1 - \boldsymbol{x}_2\|^{1/2},$$

where $L_\star := \frac{2(n)^{1/4}}{\nu(1-\gamma)^{3/2}}|\mathcal{S}|^{1/2}\left(\sum_{k=1}^n |\mathcal{A}_k| + |\mathcal{B}|\right)^{\frac{3}{4}}$.

**Proof.** Consider team policies, $\boldsymbol{x}_1, \boldsymbol{x}_2$. It holds true for the unique maximizers $\boldsymbol{\lambda}^\star(\boldsymbol{x}_1), \boldsymbol{\lambda}^\star(\boldsymbol{x}_2)$ of the adversary's regularized value function that,

$$\left(\boldsymbol{r}(\boldsymbol{x}_1) - \nu\boldsymbol{\lambda}^\star(\boldsymbol{x}_1)\right)^\top (\boldsymbol{\lambda}_1 - \boldsymbol{\lambda}^\star(\boldsymbol{x}_1)) \leq 0, \quad \forall\boldsymbol{\lambda}_1 \in \Lambda(\boldsymbol{x}_1); \tag{11}$$

$$\left(\boldsymbol{r}(\boldsymbol{x}_2) - \nu\boldsymbol{\lambda}^\star(\boldsymbol{x}_2)\right)^\top (\boldsymbol{\lambda}_2 - \boldsymbol{\lambda}^\star(\boldsymbol{x}_2)) \leq 0, \quad \forall\boldsymbol{\lambda}_2 \in \Lambda(\boldsymbol{x}_2),$$

where $\Lambda(\boldsymbol{x})$ is the set of feasible state-action visitation measures of the adversary given team's policy $\boldsymbol{x}$. To bound the distance between the two vectors $\boldsymbol{\lambda}^\star(\boldsymbol{x}_1), \boldsymbol{\lambda}^\star(\boldsymbol{x}_2)$, we observe that for any measure $\overline{\boldsymbol{\lambda}} \in \Lambda(\boldsymbol{x}_1) \cup \Lambda(\boldsymbol{x}_2)$, there exist a measure $\overline{\boldsymbol{\lambda}}_1 \in \Lambda(\boldsymbol{x}_1)$ that shares the same adversary's policy $\boldsymbol{y}$ as

in $\overline{\boldsymbol{\lambda}}$. It then follows from Lemma C.2 that $\left\|\overline{\boldsymbol{\lambda}}_1 - \overline{\boldsymbol{\lambda}}\right\| \leq L_\lambda \left\|\boldsymbol{x}_1 - \boldsymbol{x}_2\right\|$. Therefore we have for all $\overline{\boldsymbol{\lambda}} \in \Lambda(\boldsymbol{x}_1)$,

$$
\begin{aligned}
\left(\boldsymbol{r}(\boldsymbol{x}_1) - \nu\boldsymbol{\lambda}^\star(\boldsymbol{x}_1)\right)^\top \left(\overline{\boldsymbol{\lambda}} - \boldsymbol{\lambda}^\star(\boldsymbol{x}_1)\right) &= \left(\boldsymbol{r}(\boldsymbol{x}_1) - \nu\boldsymbol{\lambda}^\star(\boldsymbol{x}_1)\right)^\top \left(\overline{\boldsymbol{\lambda}} - \overline{\boldsymbol{\lambda}}_1\right) \\
&\quad + \left(\boldsymbol{r}(\boldsymbol{x}_1) - \nu\boldsymbol{\lambda}^\star(\boldsymbol{x}_1)\right)^\top \left(\overline{\boldsymbol{\lambda}}_1 - \boldsymbol{\lambda}^\star(\boldsymbol{x}_1)\right) \\
&\leq L_\lambda \sqrt{|\mathcal{S}||\mathcal{B}|} \left(1 + \frac{\nu}{1-\gamma}\right) \left\|\boldsymbol{x}_1 - \boldsymbol{x}_2\right\|. \qquad (12)
\end{aligned}
$$

Where the last inequality follows from (11) and the fact that $\left\|\boldsymbol{r}(\boldsymbol{x}) - \nu\boldsymbol{\lambda}^\star(\boldsymbol{x})\right\| \leq \sqrt{|\mathcal{S}||\mathcal{B}|} \left(1 + \frac{\nu}{1-\gamma}\right)$ for any $\boldsymbol{x} \in \mathcal{X}$. Similarly, it also holds that for all $\overline{\boldsymbol{\lambda}} \in \Lambda(\boldsymbol{x}_1)$,

$$
\left(\boldsymbol{r}(\boldsymbol{x}_2) - \nu\boldsymbol{\lambda}^\star(\boldsymbol{x}_2)\right)^\top \left(\overline{\boldsymbol{\lambda}} - \boldsymbol{\lambda}^\star(\boldsymbol{x}_2)\right) \leq L_\lambda \sqrt{|\mathcal{S}||\mathcal{B}|} \left(1 + \frac{\nu}{1-\gamma}\right) \left\|\boldsymbol{x}_1 - \boldsymbol{x}_2\right\|. \qquad (13)
$$

Plugging in $\overline{\boldsymbol{\lambda}} \leftarrow \boldsymbol{\lambda}^\star(\boldsymbol{x}_2)$ and $\overline{\boldsymbol{\lambda}} \leftarrow \boldsymbol{\lambda}^\star(\boldsymbol{x}_1)$ into (12) and (13) respectively

$$
\left(\boldsymbol{r}(\boldsymbol{x}_1) - \nu\boldsymbol{\lambda}^\star(\boldsymbol{x}_1)\right)^\top \left(\boldsymbol{\lambda}^\star(\boldsymbol{x}_2) - \boldsymbol{\lambda}^\star(\boldsymbol{x}_1)\right) \leq L_\lambda \sqrt{|\mathcal{S}||\mathcal{B}|} \left(1 + \frac{\nu}{1-\gamma}\right) \left\|\boldsymbol{x}_1 - \boldsymbol{x}_2\right\|,
$$

$$
\left(\boldsymbol{r}(\boldsymbol{x}_2) - \nu\boldsymbol{\lambda}^\star(\boldsymbol{x}_2)\right)^\top \left(\boldsymbol{\lambda}^\star(\boldsymbol{x}_1) - \boldsymbol{\lambda}^\star(\boldsymbol{x}_2)\right) \leq L_\lambda \sqrt{|\mathcal{S}||\mathcal{B}|} \left(1 + \frac{\nu}{1-\gamma}\right) \left\|\boldsymbol{x}_1 - \boldsymbol{x}_2\right\|.
$$

Adding the two inequalities results in

$$
\left(\left(\boldsymbol{r}(\boldsymbol{x}_1) - \nu\boldsymbol{\lambda}^\star(\boldsymbol{x}_1)\right) - \left(\boldsymbol{r}(\boldsymbol{x}_2) - \nu\boldsymbol{\lambda}^\star(\boldsymbol{x}_2)\right)\right)^\top \left(\boldsymbol{\lambda}^\star(\boldsymbol{x}_1) - \boldsymbol{\lambda}^\star(\boldsymbol{x}_2)\right)
$$

$$
\leq 2L_\lambda \sqrt{|\mathcal{S}||\mathcal{B}|} \left(1 + \frac{\nu}{1-\gamma}\right) \left\|\boldsymbol{x}_1 - \boldsymbol{x}_2\right\|. \qquad (14)
$$

On the other hand, since $\boldsymbol{r}(\boldsymbol{x})^\top \boldsymbol{\lambda} - \frac{\nu}{2}\|\boldsymbol{\lambda}\|^2$ is $\nu$-strongly concave in $\boldsymbol{\lambda}$, we have for all $\boldsymbol{\lambda}_1 \in \Lambda(\boldsymbol{x}_1)$.

$$
\left(\boldsymbol{\lambda}_1 - \boldsymbol{\lambda}^\star(\boldsymbol{x}_1)\right)^\top \left(\left(\boldsymbol{r}(\boldsymbol{x}_1) - \nu\boldsymbol{\lambda}_1\right) - \left(\boldsymbol{r}(\boldsymbol{x}_1) - \nu\boldsymbol{\lambda}^\star(\boldsymbol{x}_1)\right)\right) + \nu\|\boldsymbol{\lambda}_1 - \boldsymbol{\lambda}^\star(\boldsymbol{x}_1)\|^2 \leq 0.
$$

We again use the fact that for every $\overline{\boldsymbol{\lambda}} \in \overline{\Lambda}$, it holds that there exists $\overline{\boldsymbol{\lambda}}_1 \in \Lambda(\boldsymbol{x}_1)$ s.t. $\|\overline{\boldsymbol{\lambda}} - \overline{\boldsymbol{\lambda}}_1\| \leq L_\lambda\|\boldsymbol{x}_1 - \boldsymbol{x}_2\|$. Therefore it holds that for any $\overline{\boldsymbol{\lambda}} \in \overline{\Lambda}$,

$$
\begin{aligned}
0 &\geq \left(\overline{\boldsymbol{\lambda}}_1 + (\overline{\boldsymbol{\lambda}} - \overline{\boldsymbol{\lambda}}) - \boldsymbol{\lambda}^\star(\boldsymbol{x}_1)\right)^\top \left(\left(\boldsymbol{r}(\boldsymbol{x}_1) - \nu\overline{\boldsymbol{\lambda}}_1 + (\overline{\boldsymbol{\lambda}} - \overline{\boldsymbol{\lambda}})\right) - \left(\boldsymbol{r}(\boldsymbol{x}_1) - \nu\boldsymbol{\lambda}^\star(\boldsymbol{x}_1)\right)\right) \\
&\quad + \nu\|\overline{\boldsymbol{\lambda}}_1 + (\overline{\boldsymbol{\lambda}} - \overline{\boldsymbol{\lambda}}) - \boldsymbol{\lambda}^\star(\boldsymbol{x}_1)\|^2 \\
&= \left((\overline{\boldsymbol{\lambda}} - \boldsymbol{\lambda}^\star(\boldsymbol{x}_1)) + (\overline{\boldsymbol{\lambda}}_1 - \overline{\boldsymbol{\lambda}})\right)^\top \left(\left(\boldsymbol{r}(\boldsymbol{x}_1) - \nu\overline{\boldsymbol{\lambda}}_1 + (\overline{\boldsymbol{\lambda}} - \overline{\boldsymbol{\lambda}})\right) - \left(\boldsymbol{r}(\boldsymbol{x}_1) - \nu\boldsymbol{\lambda}^\star(\boldsymbol{x}_1)\right)\right) \\
&\quad + \nu\|\overline{\boldsymbol{\lambda}}_1 - \overline{\boldsymbol{\lambda}}\|^2 + \nu\|\overline{\boldsymbol{\lambda}} - \boldsymbol{\lambda}^\star(\boldsymbol{x}_1)\|^2 + 2\nu\left\langle\overline{\boldsymbol{\lambda}}_1 - \overline{\boldsymbol{\lambda}}, \overline{\boldsymbol{\lambda}} - \boldsymbol{\lambda}^\star(\boldsymbol{x}_1)\right\rangle \\
&= \left(\overline{\boldsymbol{\lambda}} - \boldsymbol{\lambda}^\star(\boldsymbol{x}_1)\right)^\top \left(\left(\boldsymbol{r}(\boldsymbol{x}_1) - \nu\overline{\boldsymbol{\lambda}}_1 + (\overline{\boldsymbol{\lambda}} - \overline{\boldsymbol{\lambda}})\right) - \left(\boldsymbol{r}(\boldsymbol{x}_1) - \nu\boldsymbol{\lambda}^\star(\boldsymbol{x}_1)\right)\right) \\
&\quad + \nu\|\overline{\boldsymbol{\lambda}}_1 - \overline{\boldsymbol{\lambda}}\|^2 + \nu\|\overline{\boldsymbol{\lambda}} - \boldsymbol{\lambda}^\star(\boldsymbol{x}_1)\|^2 + 2\nu\left\langle\overline{\boldsymbol{\lambda}}_1 - \overline{\boldsymbol{\lambda}}, \overline{\boldsymbol{\lambda}} - \boldsymbol{\lambda}^\star(\boldsymbol{x}_1)\right\rangle \\
&\quad + \left(\overline{\boldsymbol{\lambda}}_1 - \overline{\boldsymbol{\lambda}}\right)^\top \left(\left(\boldsymbol{r}(\boldsymbol{x}_1) - \nu\overline{\boldsymbol{\lambda}}_1 + (\overline{\boldsymbol{\lambda}} - \overline{\boldsymbol{\lambda}})\right) - \left(\boldsymbol{r}(\boldsymbol{x}_1) - \nu\boldsymbol{\lambda}^\star(\boldsymbol{x}_1)\right)\right).
\end{aligned}
$$

Rearranging, we have

$$
\begin{aligned}
&\left(\overline{\boldsymbol{\lambda}} - \boldsymbol{\lambda}^\star(\boldsymbol{x}_1)\right)^\top \left(\left(\boldsymbol{r}(\boldsymbol{x}_1) - \nu\overline{\boldsymbol{\lambda}}\right) - \left(\boldsymbol{r}(\boldsymbol{x}_1) - \nu\boldsymbol{\lambda}^\star(\boldsymbol{x}_1)\right)\right) + \nu\|\overline{\boldsymbol{\lambda}} - \boldsymbol{\lambda}^\star(\boldsymbol{x}_1)\|^2 \\
&\leq \underbrace{-\nu\left(\overline{\boldsymbol{\lambda}} - \boldsymbol{\lambda}^\star(\boldsymbol{x}_1)\right)^\top \left(\overline{\boldsymbol{\lambda}} - \overline{\boldsymbol{\lambda}}_1\right)}_{\Omega_1} \\
&\quad \underbrace{-\nu\|\overline{\boldsymbol{\lambda}}_1 - \overline{\boldsymbol{\lambda}}\|^2 - 2\nu\left\langle\overline{\boldsymbol{\lambda}}_1 - \overline{\boldsymbol{\lambda}}, \overline{\boldsymbol{\lambda}} - \boldsymbol{\lambda}^\star(\boldsymbol{x}_1)\right\rangle}_{\Omega_2} \\
&\quad \underbrace{-\left(\overline{\boldsymbol{\lambda}}_1 - \overline{\boldsymbol{\lambda}}\right)^\top \left(\left(\boldsymbol{r}(\boldsymbol{x}_1) - \nu\overline{\boldsymbol{\lambda}}_1 + \nu(\overline{\boldsymbol{\lambda}} - \overline{\boldsymbol{\lambda}})\right) - \left(\boldsymbol{r}(\boldsymbol{x}_1) - \nu\boldsymbol{\lambda}^\star(\boldsymbol{x}_1)\right)\right)}_{\Omega_3}.
\end{aligned}
$$

We bound $\Omega_1, \Omega_2$, and $\Omega_3$ separately.

- For $\Omega_1$, since $\|\overline{\boldsymbol{\lambda}} - \overline{\boldsymbol{\lambda}}_1\| \leq L_\lambda \|\boldsymbol{x}_1 - \boldsymbol{x}_2\|$, we have

$$\Omega_1 \leq \left| \nu \left( \overline{\boldsymbol{\lambda}} - \boldsymbol{\lambda}^\star(\boldsymbol{x}_1) \right)^\top \left( \overline{\boldsymbol{\lambda}} - \overline{\boldsymbol{\lambda}}_1 \right) \right| \leq \nu \left\| \overline{\boldsymbol{\lambda}} - \boldsymbol{\lambda}^\star(\boldsymbol{x}_1) \right\| \left\| \overline{\boldsymbol{\lambda}} - \overline{\boldsymbol{\lambda}}_1 \right\|$$

$$\leq \frac{2\nu}{1 - \gamma} \sqrt{|\mathcal{S}||\mathcal{B}|} \cdot L_\lambda \|\boldsymbol{x}_1 - \boldsymbol{x}_2\|.$$

Where we use the fact that $\left\| \overline{\boldsymbol{\lambda}} - \boldsymbol{\lambda}^\star(\boldsymbol{x}_1) \right\| \leq \left\| \overline{\boldsymbol{\lambda}} \right\| + \left\| \boldsymbol{\lambda}^\star(\boldsymbol{x}_1) \right\|$ and $\|\boldsymbol{\lambda}\| \leq \|\boldsymbol{\lambda}\|_1 = \frac{1}{1-\gamma}$.

- For $\Omega_2$, only the second term is possibly non-negative, it holds that

$$\left| \left\langle \overline{\boldsymbol{\lambda}}_1 - \overline{\boldsymbol{\lambda}}, \overline{\boldsymbol{\lambda}} - \boldsymbol{\lambda}^\star(\boldsymbol{x}_1) \right\rangle \right| \leq L_\lambda \|\boldsymbol{x}_1 - \boldsymbol{x}_2\| \frac{2}{1 - \gamma} \sqrt{|\mathcal{S}||\mathcal{B}|}.$$

Resulting in

$$\Omega_2 \leq \frac{4\nu}{1 - \gamma} \sqrt{|\mathcal{S}||\mathcal{B}|} L_\lambda \|\boldsymbol{x}_1 - \boldsymbol{x}_2\|.$$

- For $\Omega_3$:

$$\Omega_3 \leq \left| \left( \overline{\boldsymbol{\lambda}}_1 - \overline{\boldsymbol{\lambda}} \right)^\top \left( -\nu \overline{\boldsymbol{\lambda}}_1 + \nu \boldsymbol{\lambda}^\star(\boldsymbol{x}_1) \right) \right| \leq \frac{2\nu}{1 - \gamma} \sqrt{|\mathcal{S}||\mathcal{B}|} L_\lambda \|\boldsymbol{x}_1 - \boldsymbol{x}_2\|.$$

Finally, by putting the bounds of $\Omega_1, \Omega_2, \Omega_3$ and setting $L' := \frac{8\nu}{1-\gamma} \sqrt{|\mathcal{S}||\mathcal{B}|} L_\lambda$, we have for all $\overline{\boldsymbol{\lambda}} \in \overline{\Lambda}$,

$$\left( \overline{\boldsymbol{\lambda}} - \boldsymbol{\lambda}^\star(\boldsymbol{x}_1) \right)^\top \left( \left( \boldsymbol{r}(\boldsymbol{x}_1) - \nu \overline{\boldsymbol{\lambda}} \right) - \left( \boldsymbol{r}(\boldsymbol{x}_1) - \nu \boldsymbol{\lambda}^\star(\boldsymbol{x}_1) \right) \right) + \nu \|\overline{\boldsymbol{\lambda}} - \boldsymbol{\lambda}^\star(\boldsymbol{x}_1)\|^2 \leq L' \|\boldsymbol{x}_1 - \boldsymbol{x}_2\|, \tag{15}$$

Concluding, we plug $\overline{\boldsymbol{\lambda}} \leftarrow \boldsymbol{\lambda}^\star(\boldsymbol{x}_2)$ in (15), resulting

$$\left( \boldsymbol{\lambda}^\star(\boldsymbol{x}_2) - \boldsymbol{\lambda}^\star(\boldsymbol{x}_1) \right)^\top \left( \left( \boldsymbol{r}(\boldsymbol{x}_1) - \nu \boldsymbol{\lambda}^\star(\boldsymbol{x}_2) \right) - \left( \boldsymbol{r}(\boldsymbol{x}_1) - \nu \boldsymbol{\lambda}^\star(\boldsymbol{x}_1) \right) \right) + \nu \|\boldsymbol{\lambda}^\star(\boldsymbol{x}_2) - \boldsymbol{\lambda}^\star(\boldsymbol{x}_1)\|^2$$
$$\leq L' \|\boldsymbol{x}_1 - \boldsymbol{x}_2\|. \tag{16}$$

Adding (14) and (16), we have

$$2 L_\lambda \sqrt{|\mathcal{S}||\mathcal{B}|} \left( 1 + \frac{\nu}{1 - \gamma} \right) \|\boldsymbol{x}_1 - \boldsymbol{x}_2\| + L' \|\boldsymbol{x}_1 - \boldsymbol{x}_2\|$$

$$\geq \left( \boldsymbol{\lambda}^\star(\boldsymbol{x}_2) - \boldsymbol{\lambda}^\star(\boldsymbol{x}_1) \right)^\top \left( \left( \boldsymbol{r}(\boldsymbol{x}_1) - \nu \boldsymbol{\lambda}^\star(\boldsymbol{x}_1) \right) - \left( \boldsymbol{r}(\boldsymbol{x}_2) - \nu \boldsymbol{\lambda}^\star(\boldsymbol{x}_2) \right) \right)$$

$$+ \left( \boldsymbol{\lambda}^\star(\boldsymbol{x}_2) - \boldsymbol{\lambda}^\star(\boldsymbol{x}_1) \right)^\top \left( \left( \boldsymbol{r}(\boldsymbol{x}_1) - \nu \boldsymbol{\lambda}^\star(\boldsymbol{x}_2) \right) - \left( \boldsymbol{r}(\boldsymbol{x}_1) - \nu \boldsymbol{\lambda}^\star(\boldsymbol{x}_1) \right) \right) + \nu \|\boldsymbol{\lambda}^\star(\boldsymbol{x}_2) - \boldsymbol{\lambda}^\star(\boldsymbol{x}_1)\|^2$$

$$= \left( \boldsymbol{\lambda}^\star(\boldsymbol{x}_2) - \boldsymbol{\lambda}^\star(\boldsymbol{x}_1) \right)^\top \left( \left( \boldsymbol{r}(\boldsymbol{x}_1) - \nu \boldsymbol{\lambda}^\star(\boldsymbol{x}_2) \right) - \left( \boldsymbol{r}(\boldsymbol{x}_2) - \nu \boldsymbol{\lambda}^\star(\boldsymbol{x}_2) \right) \right) + \nu \|\boldsymbol{\lambda}^\star(\boldsymbol{x}_2) - \boldsymbol{\lambda}^\star(\boldsymbol{x}_1)\|^2.$$

Rearranging we get

$$\nu \|\boldsymbol{\lambda}^\star(\boldsymbol{x}_2) - \boldsymbol{\lambda}^\star(\boldsymbol{x}_1)\|^2 \leq \left( \boldsymbol{\lambda}^\star(\boldsymbol{x}_2) - \boldsymbol{\lambda}^\star(\boldsymbol{x}_1) \right)^\top \left( \left( \boldsymbol{r}(\boldsymbol{x}_2) - \nu \boldsymbol{\lambda}^\star(\boldsymbol{x}_2) \right) - \left( \boldsymbol{r}(\boldsymbol{x}_1) - \nu \boldsymbol{\lambda}^\star(\boldsymbol{x}_2) \right) \right)$$
$$+ L'' \|\boldsymbol{x}_1 - \boldsymbol{x}_2\|$$
$$\leq L_r \|\boldsymbol{\lambda}^\star(\boldsymbol{x}_2) - \boldsymbol{\lambda}^\star(\boldsymbol{x}_1)\| \|\boldsymbol{x}_1 - \boldsymbol{x}_2\| + L'' \|\boldsymbol{x}_1 - \boldsymbol{x}_2\|,$$

where $L'' := 2 \left( 1 + \frac{\nu}{1-\gamma} \right) \sqrt{|\mathcal{S}||\mathcal{B}|} L_\lambda + L' = \left( 2 + \frac{10\nu}{1-\gamma} \right) \sqrt{|\mathcal{S}||\mathcal{B}|} L_\lambda$.

By setting $\lambda = \|\boldsymbol{\lambda}^\star(\boldsymbol{x}_2) - \boldsymbol{\lambda}^\star(\boldsymbol{x}_1)\|$ and $\chi = \|\boldsymbol{x}_1 - \boldsymbol{x}_2\|$, we consider the inequality $\nu \lambda^2 \leq L_r \lambda \chi + L'' \chi$ with coefficients $\nu, L_r, L'' \geq 0$ and variables $\lambda$ and $\chi$. Choosing $\alpha \leftarrow \nu$, $\beta \leftarrow L_r$, and $\gamma \leftarrow L''$, then Proposition C.2 implies that $\boldsymbol{\lambda}^\star(\boldsymbol{x})$ is not Lipschitz-continuous w.r.t the team policy $\boldsymbol{x}$. Hence, we consider a solution of the form $0 \leq \lambda \leq \frac{\beta \chi + \sqrt{\chi(4\alpha\gamma + \beta^2 \chi)}}{2\alpha} \leq c \chi^p$, where $\frac{1}{2} - p \geq 0$. We choose $p = \frac{1}{2}$ since it yields the best convergence rate from Theorem B.1. Solving the above inequality with $p = \frac{1}{2}$ gives that

- if $\chi = 0$, the inequality holds trivially;

- if $\chi > 0$, we have

$$c \geq \frac{\beta\chi + \sqrt{\chi(4\alpha\gamma + \beta^2\chi)}}{2\alpha\sqrt{\chi}} = \frac{\beta\sqrt{\chi}}{2\alpha} + \frac{\sqrt{4\alpha\gamma + \beta^2\chi}}{2\alpha}.$$

Since $\chi = \|\boldsymbol{x}_1 - \boldsymbol{x}_2\| \leq \sqrt{n|\mathcal{S}|}\mathrm{Diam}_{\mathcal{X}_i} = \sqrt{2n|\mathcal{S}|}$, by plugging in the coefficients, we have

$$c = \frac{1}{\nu}\sqrt{2L_r^2 X + 4\nu L''} \leq \frac{2(n)^{1/4}}{\nu(1-\gamma)^{3/2}}|\mathcal{S}|^{1/2}\left(\sum_{k=1}^{n}|\mathcal{A}_k| + |\mathcal{B}|\right)^{\frac{3}{4}}.$$

By setting $L_\star = c$, we conclude that

$$\|\boldsymbol{\lambda}^*(\boldsymbol{x}_1) - \boldsymbol{\lambda}^*(\boldsymbol{x}_2)\| \leq L_\star\|\boldsymbol{x}_1 - \boldsymbol{x}_2\|^{1/2}.$$

$\square$

We are now ready to show that $\Phi^\nu(\boldsymbol{x})$ is weakly-smooth.

**Theorem C.2** (Hölder Continuous Max Value Func.). *Let function $\Phi^\nu(\boldsymbol{x})$ be the maximum function of the regularized value function, $\Phi^\nu(\boldsymbol{x}) := \max_{\boldsymbol{y}\in\mathcal{Y}}V_{\boldsymbol{\rho}}^\nu(\boldsymbol{x},\boldsymbol{y})$. It is the case that,*

- $\Phi^\nu$ *is differentiable,*

- $\nabla_{\boldsymbol{x}}\Phi^\nu$ *is $(1/2, \ell_{1/2})$-Hölder continuous, i.e,*

$$\|\nabla_{\boldsymbol{x}}\Phi^\nu(\boldsymbol{x}) - \nabla_{\boldsymbol{x}}\Phi^\nu(\boldsymbol{x}')\| \leq \ell_{1/2}\|\boldsymbol{x} - \boldsymbol{x}'\|^{1/2},$$

*with $\ell_{1/2} := \frac{30n^{\frac{1}{4}}|\mathcal{S}|^{\frac{5}{4}}\left(\sum_i|\mathcal{A}_i| + |\mathcal{B}|\right)^2}{\nu\min_s\rho(s)(1-\gamma)^{\frac{13}{2}}}.$*

**Proof.** Since $\Phi^\nu(\boldsymbol{x})$ has a unique maximizer $\boldsymbol{\lambda} \in \Lambda(\boldsymbol{x})$, by applying Danskin's Theorem [8] and the "1–1" correspondence between $\boldsymbol{\lambda}$ and $\boldsymbol{y}$ (Theorem C.1), we have

$$\|\nabla_{\boldsymbol{x}}\Phi^\nu(\boldsymbol{x}) - \nabla_{\boldsymbol{x}}\Phi^\nu(\boldsymbol{x}')\| = \left\|\nabla_{\boldsymbol{x}}V_{\boldsymbol{\rho}}^\nu(\boldsymbol{x},\boldsymbol{y}(\boldsymbol{\lambda}^\star(\boldsymbol{x}))) - \nabla_{\boldsymbol{x}}V_{\boldsymbol{\rho}}^\nu(\boldsymbol{x}',\boldsymbol{y}(\boldsymbol{\lambda}^\star(\boldsymbol{x}')))\right\|$$

$$\leq \ell_\nu\left(\|\boldsymbol{x} - \boldsymbol{x}'\| + \|\boldsymbol{y}(\boldsymbol{\lambda}^\star(\boldsymbol{x})) - \boldsymbol{y}(\boldsymbol{\lambda}^\star(\boldsymbol{x}'))\|\right)$$

$$\leq \ell_\nu(\|\boldsymbol{x} - \boldsymbol{x}'\| + L_{\lambda_{\mathrm{inv}}}\|\boldsymbol{\lambda}^\star(\boldsymbol{x}) - \boldsymbol{\lambda}^\star(\boldsymbol{x}')\|)$$

$$\leq \ell_\nu\left((2n|\mathcal{S}|)^{\frac{1}{4}} + L_{\lambda_{\mathrm{inv}}}L_\star\right)\cdot\|\boldsymbol{x} - \boldsymbol{x}'\|^{\frac{1}{2}}.$$

Where in the last inequality we used Lemma C.8 and the fact that $\|\boldsymbol{x} - \boldsymbol{x}'\| \leq \sqrt{2n|\mathcal{S}|}$. Plugging in the coefficients in Lemma C.8, Lemma C.3, and Lemma C.4, it yields that

$$\|\nabla_{\boldsymbol{x}}\Phi^\nu(\boldsymbol{x}) - \nabla_{\boldsymbol{x}}\Phi^\nu(\boldsymbol{x}')\| \leq \frac{30n^{\frac{1}{4}}|\mathcal{S}|^{\frac{5}{4}}\left(\sum_i|\mathcal{A}_i| + |\mathcal{B}|\right)^2}{\nu\min_s\boldsymbol{\rho}(s)(1-\gamma)^{\frac{13}{2}}}\cdot\|\boldsymbol{x} - \boldsymbol{x}'\|^{\frac{1}{2}}.$$

$\square$

## C.4 Analysis of ISPNG: Proof of Theorem 3.3

In this part we show that Algorithm 1 converges to an $\epsilon$-NE. Essentially, Algorithm 1 implements projected gradient descent on the regularized maximum function $\Phi^\nu : \mathcal{X} \to \mathbb{R}$ with a stochastic $\vartheta$-inexact gradient oracle. Function $\Phi^\nu$ is Hölder-continuous (see Theorem C.2) and as such we can invoke Theorem C.3 to prove convergence to an $\epsilon$-FOSP.

The inexactness of the gradient oracle, $\vartheta$, is the sum of two error sources:

1. the fact that the adversary can only approximately maximize the regularized value function $V_{\boldsymbol{\rho}}^\nu(\boldsymbol{x},\boldsymbol{y})$ — the iteration and sample complexity of maximizing $V_{\boldsymbol{\rho}}^\nu(\boldsymbol{x},\cdot)$ is provided in Theorem C.4;

2. the exact estimation of $\nabla \Phi^\nu$ requires estimation of the adversary's policy $\boldsymbol{y}$ — as we assume that the agents do not observe each other's actions this is impossible. Nevertheless, in Lemma C.9 it is proven that the inexactness error is bounded and controlled through the regularizer's coefficient $\nu$.

After quantifying $\vartheta$ as the function of the latter two terms, the optimality gap of Algorithm 2 and a term that scales with $O(\nu)$, we can tune the rest of the parameters accordingly.

The resulting $\epsilon$-FOSP, thanks to the gradient domination property (Lemma C.7), corresponds to an $\epsilon$-NE.

**Bounding the error of the inexact gradient.** Following, we prove that the inexactness error of the gradient oracle is bounded by a function of the controllable parameter $\nu$.

**Lemma C.9** (Inexact gradient). *Let* $V_{\boldsymbol{\rho}}^\nu(\boldsymbol{x}, \boldsymbol{y}) := V_{\boldsymbol{\rho}}(\boldsymbol{x}, \boldsymbol{y}) - \frac{\nu}{2} \sum_s \|d^{\boldsymbol{x}, \boldsymbol{y}}(s)\boldsymbol{y}(s)\|^2$, $\boldsymbol{y}^\nu(\boldsymbol{x}) := \mathrm{argmax}_{\boldsymbol{y}}\{V_{\boldsymbol{\rho}}^\nu(\boldsymbol{x}, \boldsymbol{y})\}$, and $\boldsymbol{g}(\boldsymbol{x}) := \nabla_{\boldsymbol{x}} V(\boldsymbol{x}, \boldsymbol{y}_\nu(\boldsymbol{x}))$. Then, it holds that*

$$\|\boldsymbol{g}(\boldsymbol{x}) - \nabla_{\boldsymbol{x}} V_{\boldsymbol{\rho}}^\nu(\boldsymbol{x}, \boldsymbol{y})\|_2 \leq \frac{\nu|\mathcal{S}|^{\frac{1}{2}}\left(\sum_{i=1}^n |\mathcal{A}_i| + |\mathcal{B}|\right)}{(1-\gamma)^3}.$$

**Proof.** We observe that

$$\|\boldsymbol{g}(\boldsymbol{x}) - \nabla_{\boldsymbol{x}} V_{\boldsymbol{\rho}}^\nu(\boldsymbol{x}, \boldsymbol{y})\| = \left\|\nabla_{\boldsymbol{x}}\left(-\frac{\nu}{2}\|\boldsymbol{\lambda}(\boldsymbol{y}; \boldsymbol{x})\|^2\right)\right\|$$
$$= \nu \|\boldsymbol{\lambda}(\boldsymbol{y}; \boldsymbol{x})\| \|\nabla_{\boldsymbol{x}}\boldsymbol{\lambda}(\boldsymbol{y}; \boldsymbol{x})\|$$
$$\leq \frac{\nu}{1-\gamma} L_\lambda.$$

Where in the last inequality we use the fact that $\|\boldsymbol{\lambda}\| \leq \frac{1}{1-\gamma}$ and $\nabla_{\boldsymbol{x}}\boldsymbol{\lambda}(\boldsymbol{y}; \boldsymbol{x}) \leq L_\lambda$.

$\square$

**Learning an $\epsilon$-NE.** We can now compile the intermediate statements to guarantee that ISPNG computes an $\epsilon$-NE for any desired accuracy $\epsilon > 0$ within a finite number of iterations and samples.

**Theorem C.3.** *Consider an adversarial team Markov game $\Gamma$ and Algorithm 1, ISPNG, with an outer-loop parameter tuning of:*

- $T = \frac{1061683200 D_{\mathrm{m}}^5 n^{\frac{1}{2}} |\mathcal{S}|^{\frac{9}{2}}\left(\sum_{i=1}^n |\mathcal{A}_i| + |\mathcal{B}|\right)^6}{(1-\gamma)^{24}(\min_s \rho(s))^2 \epsilon^5}$;

- $\eta_x = \frac{(\min_s \rho(s))^2(1-\gamma)^{22}\epsilon^3}{33177600 D_{\mathrm{m}}^3 n^{\frac{1}{2}} |\mathcal{S}|^{\frac{9}{2}}\left(\sum_{i=1}^n |\mathcal{A}_i| + |\mathcal{B}|\right)^6}$;

- $\zeta_x = \frac{(1-\gamma)^3 \epsilon}{6 D_{\mathrm{m}} |\mathcal{S}|\left(\sum_{i=1}^n |\mathcal{A}_i| + |\mathcal{B}|\right)^{\frac{3}{2}}}$;

- $M = \frac{2034 D_{\mathrm{m}}^3 |\mathcal{S}|\left(\sum_{i=1}^n |\mathcal{A}_i| + |\mathcal{B}|\right)^{\frac{7}{2}}}{(1-\gamma)^{10}(\min_s \rho(s))^4 \epsilon^3} \max\left\{\frac{(1-\gamma)^4(\min_s \rho(s))^4\left(\sum_{i=1}^n |\mathcal{A}_i| + |\mathcal{B}|\right)}{|\mathcal{S}|}, \frac{9}{2}\right\}.$

*Also, let the tuning of the inner-loop subroutine Algorithm 2 (VIS-REG-PG) be:*

- $\nu = \frac{(1-\gamma)^4 \epsilon}{48 D_{\mathrm{m}} |\mathcal{S}|\left(\sum_{i=1}^n |\mathcal{A}_i| + |\mathcal{B}|\right)}$;

- $T_y = \tilde{O}\left(\frac{D_{\mathrm{m}}^5 |\mathcal{S}|^6\left(\sum_{i=1}^n |\mathcal{A}_i| + |\mathcal{B}|\right)^9}{(1-\gamma)^{21}(\min_s \rho(s))^4 \epsilon^5}\right)$;

- $\eta_y = \frac{(1-\gamma)^{28}(\min_s \rho(s))^4 \epsilon^4}{978447237120 D_{\mathrm{m}}^4 |\mathcal{S}|^5\left(\sum_{i=1}^n |\mathcal{A}_i| + |\mathcal{B}|\right)^8}$;

- $\zeta_y = \frac{(1-\gamma)^{15}(\min_s \rho(s))^2 \epsilon^3}{18432 D_{\mathrm{m}}^2 |\mathcal{S}|^{\frac{7}{2}}\left(\sum_{i=1}^n |\mathcal{A}_i| + |\mathcal{B}|\right)^6}$;

- $K = \frac{19365101568 D_{\mathrm{m}}^4 |\mathcal{S}|^7\left(\sum_{i=1}^n |\mathcal{A}_i| + |\mathcal{B}|\right)^{12}}{(1-\gamma)^{36}(\min_s \rho(s))^4 \epsilon^6}$;

- $H = \frac{2}{1-\gamma} \log \left( \frac{2293235712 D_{\mathrm{m}}^4 |\mathcal{S}|^4 (\sum_{i=1}^{n} |\mathcal{A}_i| + |\mathcal{B}|)^6}{(1-\gamma)^{22} (\min_s \rho(s))^2 \epsilon^4} \right).$

*It is the case that the output of the algorithm, $(\boldsymbol{x}^*, \boldsymbol{y}^*)$, will be an $\epsilon$-NE in expectation. Specifically, we have*

$$\mathbb{E} \left[ V_{\boldsymbol{\rho}}(\boldsymbol{x}^*, \boldsymbol{y}^*) - \min_{\boldsymbol{x}_i' \in \mathcal{X}_i} V_{\boldsymbol{\rho}}(\boldsymbol{x}_i', \boldsymbol{x}_{-i}^*, \boldsymbol{y}^*) \right] \le \epsilon, \quad \forall i \in [n]$$

*and*

$$\mathbb{E} \left[ \max_{\boldsymbol{y}' \in \mathcal{Y}} V_{\boldsymbol{\rho}}(\boldsymbol{x}^*, \boldsymbol{y}') - V_{\boldsymbol{\rho}}(\boldsymbol{x}^*, \boldsymbol{y}^*) \right] \le \epsilon.$$

**Proof.** Let $\boldsymbol{x}^*, \boldsymbol{y}^*$ be the final output of the algorithm, from Lemma C.7, we have

$$\mathbb{E} \left[ V_{\boldsymbol{\rho}}(\boldsymbol{x}^*, \boldsymbol{y}^*) - \min_{\boldsymbol{x}_i' \in \mathcal{X}_i} V_{\boldsymbol{\rho}}(\boldsymbol{x}_i', \boldsymbol{x}_{-i}^*, \boldsymbol{y}^*) \right]$$

$$\le \frac{1}{1-\gamma} D_{\mathrm{m}} \mathbb{E} \left[ \max_{\boldsymbol{x}_i'} \left( -\nabla V_{\boldsymbol{\rho}}(\boldsymbol{x}^*, \boldsymbol{y}^*) \right)^\top (\boldsymbol{x}_i' - \boldsymbol{x}_i^*) \right] + \frac{2 D_{\mathrm{m}} \zeta |\mathcal{S}| \left( \sum_{i=1}^{n} |\mathcal{A}_i| + |\mathcal{B}| \right) L}{1-\gamma}$$

$$\le \frac{1}{1-\gamma} D_{\mathrm{m}} \mathbb{E} \left[ \max_{\boldsymbol{x}_i'} \left( -\nabla V_{\boldsymbol{\rho}}^{\nu}(\boldsymbol{x}^*, \boldsymbol{y}^*) \right)^\top (\boldsymbol{x}_i' - \boldsymbol{x}_i^*) \right] + \frac{\nu}{1-\gamma} L_\lambda \mathrm{Diam}_{\mathcal{X}_i}$$

$$+ \frac{2 D_{\mathrm{m}} \zeta |\mathcal{S}| \left( \sum_{i=1}^{n} |\mathcal{A}_i| + |\mathcal{B}| \right) L}{1-\gamma}. \tag{17}$$

Where (17) follows from Lemma C.9. As for the first term of (17), let us define $\boldsymbol{y}^\star(\boldsymbol{x}) = \mathrm{argmax}_{\boldsymbol{y} \in \mathcal{Y}} V_{\boldsymbol{\rho}}^{\nu}(\boldsymbol{x}, \boldsymbol{y})$ and $\boldsymbol{x}^+ = \mathrm{Proj}_{\mathcal{X}} \left( \boldsymbol{x}^{t^*-1} - \eta_x \nabla V_{\boldsymbol{\rho}}(\boldsymbol{x}^{t^*-1}, \boldsymbol{y}^\star(\boldsymbol{x}^{t^*-1})) \right).$ Then it holds that,

$$\mathbb{E} \left[ \max_{\boldsymbol{x}_i'} \left\{ \left( -\nabla V_{\boldsymbol{\rho}}^{\nu}(\boldsymbol{x}^*, \boldsymbol{y}^*) \right)^\top (\boldsymbol{x}_i' - \boldsymbol{x}_i^*) \right\} \right]$$

$$= \mathbb{E} \left[ \max_{\boldsymbol{x}_i'} \left\{ \left( -\nabla V_{\boldsymbol{\rho}}^{\nu}(\boldsymbol{x}_i^+, \boldsymbol{x}_{-i}^*, \boldsymbol{y}^\star(\boldsymbol{x}^+)) \right)^\top (\boldsymbol{x}_i' - \boldsymbol{x}_i^*) \right. \right.$$

$$\left. \left. + \left( \nabla V_{\boldsymbol{\rho}}^{\nu}(\boldsymbol{x}_i^+, \boldsymbol{x}_{-i}^* \boldsymbol{y}^\star(\boldsymbol{x}^+)) - \nabla V_{\boldsymbol{\rho}}^{\nu}(\boldsymbol{x}^*, \boldsymbol{y}^*) \right)^\top (\boldsymbol{x}_i' - \boldsymbol{x}_i^*) \right\} \right]$$

$$\le \mathbb{E} \left[ \max_{\boldsymbol{x}_i'} \left\{ \left( -\nabla V_{\boldsymbol{\rho}}^{\nu}(\boldsymbol{x}_i^+ \boldsymbol{x}_{-i}^*, \boldsymbol{y}^\star(\boldsymbol{x}^+)) \right)^\top (\boldsymbol{x}_i' - \boldsymbol{x}_i^+) \right\} \right] + L_\nu \mathbb{E} \left[ \| \boldsymbol{x}_i^+ - \boldsymbol{x}_i^* \| \right]$$

$$+ \mathbb{E} \left[ \| \nabla V_{\boldsymbol{\rho}}^{\nu}(\boldsymbol{x}_i^+, \boldsymbol{x}_{-i}^*, \boldsymbol{y}^\star(\boldsymbol{x}^+)) - \nabla V_{\boldsymbol{\rho}}^{\nu}(\boldsymbol{x}^*, \boldsymbol{y}^*) \| \right] \cdot \mathrm{Diam}_{\mathcal{X}_i} \tag{18}$$

$$\le \mathbb{E} \left[ \max_{\boldsymbol{x}_i'} \left\{ \left( -\nabla V_{\boldsymbol{\rho}}^{\nu}(\boldsymbol{x}_i^+, \boldsymbol{x}_{-i}^*, \boldsymbol{y}^\star(\boldsymbol{x}^+)) \right)^\top (\boldsymbol{x}_i' - \boldsymbol{x}_i^+) \right\} \right] + L_\nu \mathbb{E} \left[ \| \boldsymbol{x}_i^+ - \boldsymbol{x}_i^* \| \right]$$

$$+ \ell_\nu \left( \mathbb{E} \left[ \| \boldsymbol{x}^+ - \boldsymbol{x}^* \| \right] + \mathbb{E} \left[ \| \boldsymbol{y}^\star(\boldsymbol{x}^+) - \boldsymbol{y}^* \| \right] \right) \cdot \mathrm{Diam}_{\mathcal{X}_i} \tag{19}$$

$$\le \mathbb{E} \left[ \max_{\boldsymbol{x}_i'} \left\{ \left( -\nabla V_{\boldsymbol{\rho}}^{\nu}(\boldsymbol{x}_i^+, \boldsymbol{x}_{-i}^*, \boldsymbol{y}^\star(\boldsymbol{x}^+)) \right)^\top (\boldsymbol{x}_i' - \boldsymbol{x}_i^+) \right\} \right] + L_\nu \mathbb{E} \left[ \| \boldsymbol{x}_i^+ - \boldsymbol{x}_i^* \| \right]$$

$$+ \ell_\nu \left( \mathbb{E} \left[ \| \boldsymbol{x}^+ - \boldsymbol{x}^* \| \right] + \mathbb{E} \left[ \| \boldsymbol{y}^\star(\boldsymbol{x}^+) - \boldsymbol{y}^\star(\boldsymbol{x}^*) \| \right] + \mathbb{E} \left[ \| \boldsymbol{y}^\star(\boldsymbol{x}^*) - \boldsymbol{y}^* \| \right] \right) \cdot \mathrm{Diam}_{\mathcal{X}_i}.$$

Where

- Equation (18) is due to $\| \nabla V_{\boldsymbol{\rho}}^{\nu}(\boldsymbol{x}, \boldsymbol{y}) \| \le L_\nu$;
- Equation (19) follows from the fact that $V_{\boldsymbol{\rho}}^{\nu}(\boldsymbol{x}, \boldsymbol{y})$ is $\ell_\nu$-smooth.

By choosing parameters specified above and combining Corollaries B.1 and B.2, Theorem C.4, Lemma C.12, and Claim B.1, we have the desired result. On the other hand, since

$$\mathbb{E}\left[\max_{\boldsymbol{y}'\in\mathcal{Y}} V_{\boldsymbol{\rho}}(\boldsymbol{x}^*,\boldsymbol{y}') - V_{\boldsymbol{\rho}}(\boldsymbol{x}^*,\boldsymbol{y}^*)\right] \leq \mathbb{E}\left[\max_{\boldsymbol{y}'\in\mathcal{Y}} V_{\boldsymbol{\rho}}^{\nu}(\boldsymbol{x}^*,\boldsymbol{y}') - V_{\boldsymbol{\rho}}^{\nu}(\boldsymbol{x}^*,\boldsymbol{y}^*)\right] + \frac{\nu}{(1-\gamma)^2}$$

$$= \mathbb{E}\left[V_{\boldsymbol{\rho}}^{\nu}(\boldsymbol{x}^*,\boldsymbol{y}^{\star}(\boldsymbol{x}^*)) - V_{\boldsymbol{\rho}}^{\nu}(\boldsymbol{x}^*,\boldsymbol{y}^*)\right] + \frac{\nu}{(1-\gamma)^2} \quad (20)$$

$$\leq L_{\nu}E\left[\|\boldsymbol{y}^{\star}(\boldsymbol{x}^*) - \boldsymbol{y}^*\|\right] + \frac{\nu}{(1-\gamma)^2}.$$

Where in (20) we use the fact that $\|\boldsymbol{\lambda}\|^2 \leq \frac{1}{(1-\gamma)^2}$. Combining Theorem C.4, Lemma C.12, and choosing parameters specified above gives the desired result. $\square$

## C.5 Visitation-Regularized Policy Gradient Analysis

In this section, we consider the direct parameterization for the policy of the adversary. For any policy $\boldsymbol{y} \in \mathcal{Y}$, for any state $s \in \mathcal{S}$ and any action $b \in \mathcal{B}$, we have

$$y(b|s) = y_{s,b}.$$

Where $y_{s,b}$ denotes $(s,b)^{th}$ entry of the policy vector $\boldsymbol{y}$. In this section, we mainly focus on solving the following policy optimization problem:

$$\max_{\boldsymbol{y}\in\boldsymbol{y}^{\zeta}} V_{\boldsymbol{\rho}}^{\nu}(\boldsymbol{x},\boldsymbol{y}) := \max_{\boldsymbol{y}\in\mathcal{Y}^{\zeta}}\left\{\boldsymbol{r}(\boldsymbol{x})^{\top}\boldsymbol{\lambda}(\boldsymbol{y};\boldsymbol{x}) - \frac{\nu}{2}\|\boldsymbol{\lambda}(\boldsymbol{y};\boldsymbol{x})\|^2\right\}. \quad (21)$$

Where $\boldsymbol{\lambda}(\boldsymbol{y};\boldsymbol{x})$ is the state-action visitation measure under policy $\boldsymbol{y}$ as in Definition 2.4. $\boldsymbol{r}(\boldsymbol{x})$ is the induced pay-off vector for the adversary when the team is playing according to strategy $\boldsymbol{x}$ and $\nu$ is the regularization coefficient. Then by policy gradient theorem [113], denote $F^{\nu}(\boldsymbol{\lambda}(\boldsymbol{y};\boldsymbol{x})) = V_{\boldsymbol{\rho}}^{\nu}(\boldsymbol{x},\boldsymbol{y})$ we have

$$\nabla_{\boldsymbol{y}}F^{\nu}(\boldsymbol{\lambda}(\boldsymbol{y};\boldsymbol{x})) = [\nabla_{\boldsymbol{y}}\boldsymbol{\lambda}(\boldsymbol{y};\boldsymbol{x})]^{\top}(\boldsymbol{r}(\boldsymbol{x}) - \nu\boldsymbol{\lambda}(\boldsymbol{y};\boldsymbol{x}))$$

$$= \mathbb{E}_{\boldsymbol{\rho},\boldsymbol{y}}\left[\sum_{h=0}^{\infty}\gamma^h \cdot (\boldsymbol{r}(\boldsymbol{x}) - \nu\boldsymbol{\lambda}(\boldsymbol{y};\boldsymbol{x}))_{s_h,b_h} \cdot \left(\sum_{h'=0}^{h}\nabla_{\boldsymbol{y}}\log y(b_h'|s_{h'})\right)\right].$$

Given the direct parameterization, we can show the following lemmas:

**Lemma C.10.** For any adversarial policy $\boldsymbol{y}$ and state-action pair $(s,b)$, we have $\|\nabla_{\boldsymbol{y}}\log y(b|s)\| \leq \frac{1}{\zeta}$, $\|\nabla_{\boldsymbol{y}}^2\log y(b|s)\| \leq \frac{1}{\zeta^2}$, and $\|\nabla_{\boldsymbol{y}}F^{\nu}(\boldsymbol{\lambda}(\boldsymbol{y};\boldsymbol{x}))\| \leq \frac{1}{(1-\gamma)^2\zeta} + \frac{\nu}{(1-\gamma)^3\zeta}$ for any fixed $\boldsymbol{x}$.

**Proof.** By direct parameterization $y(b|s) = y_{s,b}$, we have

$$\|\nabla_{\boldsymbol{y}}\log y(b|s)\| = \|\nabla_{\boldsymbol{y}}\log y_{s,b}\| = \left\|\frac{1}{y_{s,b}}\boldsymbol{e}_{s,b}\right\| \leq \frac{1}{\zeta}. \quad (22)$$

Where $\boldsymbol{e}_{s,b}$ denotes the standard basis for the $(s,b)^{th}$ entry. Similarly, we have

$$\|\nabla_{\boldsymbol{y}}^2\log y(b|s)\| = \left\|\text{diag}\left(\frac{1}{y_{s,b}^2}\right)\right\| \leq \frac{1}{\zeta^2}. \quad (23)$$

Where $\text{diag}(\cdot)$ denotes the standard diagonal matrix. For the policy gradient, we show that

$$\|\nabla_{\boldsymbol{y}}F^{\nu}(\boldsymbol{\lambda}(\boldsymbol{y};\boldsymbol{x}))\| = \left\|[\nabla_{\boldsymbol{y}}\boldsymbol{\lambda}(\boldsymbol{y};\boldsymbol{x})]^{\top}(\boldsymbol{r}(\boldsymbol{x}) - \nu\boldsymbol{\lambda}(\boldsymbol{y};\boldsymbol{x}))\right\|$$

$$= \left\|\mathbb{E}\left[\sum_{h=0}^{\infty}\gamma^h \cdot (r(\boldsymbol{x})_{s_h b_h} - \nu\lambda(\boldsymbol{y};\boldsymbol{x})_{s_h b_h}) \cdot \left(\sum_{h'=0}^{h}\nabla_{\boldsymbol{y}}\log y(b_{h'}|s_{h'})\right)\right]\right\|$$

$$\leq \sum_{h=0}^{\infty}\gamma^h \cdot (1 + \frac{\nu}{1-\gamma}) \cdot (h+1) \cdot \frac{1}{\zeta} \quad (24)$$

$$\leq \frac{1}{(1-\gamma)^2\zeta} + \frac{\nu}{(1-\gamma)^3\zeta}.$$

Where (24) is due to (22). $\square$

Before we proceed to show the convergence towards global optimality for (21), we first define the notion of Moreau envelope and the proximal point.

**Definition C.3** (Moreau Envelope and Proximal Point). *For any $\boldsymbol{y} \in \mathcal{Y}^\zeta$, we use $F_{1/\beta}^\nu(\boldsymbol{\lambda}(\boldsymbol{y}; \boldsymbol{x}))$ to denote the Moreau envelope of function $F^\nu(\boldsymbol{\lambda}(\boldsymbol{y}; \boldsymbol{x}))$ such that*

$$F_{1/\beta}^\nu(\boldsymbol{\lambda}(\boldsymbol{y}; \boldsymbol{x})) := \max_{\boldsymbol{z} \in \mathcal{Y}^\zeta} \left\{ F^\nu(\boldsymbol{\lambda}(\boldsymbol{z}; \boldsymbol{x})) - \frac{\beta}{2} \|\boldsymbol{\lambda}(\boldsymbol{z}; \boldsymbol{x}) - \boldsymbol{\lambda}(\boldsymbol{y}; \boldsymbol{x})\|^2 \right\}.$$

*Moreover, we define the proximal point $\hat{\boldsymbol{y}}_{1/\beta}$ of Moreau envelope as following:*

$$\hat{\boldsymbol{y}} := \operatorname*{argmax}_{\boldsymbol{z} \in \mathcal{Y}^\zeta} \left\{ F^\nu(\boldsymbol{\lambda}(\boldsymbol{y}; \boldsymbol{x})) - \frac{\beta}{2} \|\boldsymbol{\lambda}(\boldsymbol{z}; \boldsymbol{x}) - \boldsymbol{\lambda}(\boldsymbol{y}; \boldsymbol{x})\|^2 \right\}.$$

Now we proceed to show the following lemma:

**Lemma C.11.** When running Algorithm 2, for any $t \geq 0$, we have

$$\mathbb{E}\left[ \left\| \boldsymbol{y}^{(t+1)} - \hat{\boldsymbol{y}}^{(t)} \right\|^2 \Big| \boldsymbol{y}^{(t)} \right] \leq (1 - \eta_y\beta) \left\| \boldsymbol{y}^{(t)} - \hat{\boldsymbol{y}}^{(t)} \right\|^2 + 2(1 - \eta_y\beta)\eta_y(1 + \eta_y\ell_\nu) \cdot \mathcal{C}_3\gamma^H$$

$$+ \eta_y^2 \mathbb{E}\left[ \left\| \nabla_{\boldsymbol{y}} F^\nu(\boldsymbol{\lambda}(\boldsymbol{y}^{(t)}; \boldsymbol{x})) - \hat{\boldsymbol{g}}_{\boldsymbol{y}}^{(t)}) \right\|^2 \Big| \boldsymbol{y}^{(t)} \right].$$

Where $\mathcal{C}_3 = \sqrt{|\mathcal{S}||\mathcal{B}|} \frac{6H}{(1-\gamma)^3\zeta}$.

**Proof.**

$$\mathbb{E}\left[ \left\| \boldsymbol{y}^{t+1} - \hat{\boldsymbol{y}}^{(t)} \right\|^2 \big| \boldsymbol{y}^{(t)} \right]$$

$$= \mathbb{E}\left[ \left\| \operatorname{Proj}_{\mathcal{Y}}(\boldsymbol{y}^{(t)} + \eta_y\hat{\boldsymbol{g}}_{\boldsymbol{y}}^{(t)}) - \operatorname{Proj}_{\mathcal{Y}}\left((1 - \eta_y\beta)\hat{\boldsymbol{y}}^{(t)} + \eta_y\beta\boldsymbol{y}^{(t)} + \eta_y\nabla_{\boldsymbol{y}}F^\nu(\boldsymbol{\lambda}(\hat{\boldsymbol{y}}^{(t)}; \boldsymbol{x}))) \right\|^2 \big| \boldsymbol{y}^{(t)} \right] \tag{25}$$

$$\leq \mathbb{E}\left[ \left\| \boldsymbol{y}^{(t)} + \eta_y\hat{\boldsymbol{g}}_{\boldsymbol{y}}^{(t)} - \left((1 - \eta_y\beta)\hat{\boldsymbol{y}}^{(t)} + \eta_y\beta\boldsymbol{y}^{(t)} + \eta_y\nabla_{\boldsymbol{y}}F^\nu(\boldsymbol{\lambda}(\hat{\boldsymbol{y}}^{(t)}; \boldsymbol{x}))) \right\|^2 \big| \boldsymbol{y}^{(t)} \right]$$

$$= \mathbb{E}\left[ \left\| (1 - \eta_y\beta)(\boldsymbol{y}^{(t)} - \hat{\boldsymbol{y}}^{(t)}) + \eta_y\left(\nabla_{\boldsymbol{y}}F^\nu(\boldsymbol{\lambda}(\hat{\boldsymbol{y}}^{(t)}; \boldsymbol{x})) - \hat{\boldsymbol{g}}_{\boldsymbol{y}}^{(t)}\right) \right\|^2 \big| \boldsymbol{y}^{(t)} \right]$$

$$= \mathbb{E}\left[ \left\| (1 - \eta_y\beta)(\boldsymbol{y}^{(t)} - \hat{\boldsymbol{y}}^{(t)}) + \eta_y\left(\nabla_{\boldsymbol{y}}F^\nu(\boldsymbol{\lambda}(\hat{\boldsymbol{y}}^{(t)}; \boldsymbol{x})) - \nabla_{\boldsymbol{y}}F^\nu(\boldsymbol{\lambda}(\boldsymbol{y}^{(t)}; \boldsymbol{x}))\right) + \eta_y\left(\nabla_{\boldsymbol{y}}F^\nu(\boldsymbol{\lambda}(\boldsymbol{y}^{(t)}; \boldsymbol{x})) - \hat{\boldsymbol{g}}_{\boldsymbol{y}}^{(t)}\right) \right\|^2 \big| \boldsymbol{y}^{(t)} \right]$$

$$= \left\| (1 - \eta_y\beta)(\boldsymbol{y}^{(t)} - \hat{\boldsymbol{y}}^{(t)}) + \eta_y\left(\nabla_{\boldsymbol{y}}F^\nu(\boldsymbol{\lambda}(\hat{\boldsymbol{y}}^{(t)}; \boldsymbol{x})) - \nabla_{\boldsymbol{y}}F^\nu(\boldsymbol{\lambda}(\boldsymbol{y}^{(t)}; \boldsymbol{x}))\right) \right\|^2$$

$$+ 2(1 - \eta_y\beta)\eta_y\mathbb{E}\left[ \left\langle (\boldsymbol{y}^{(t)} - \hat{\boldsymbol{y}}^{(t)}) - \eta_y\left(\nabla_{\boldsymbol{y}}F^\nu(\boldsymbol{\lambda}(\hat{\boldsymbol{y}}^{(t)}; \boldsymbol{x})) - \nabla_{\boldsymbol{y}}F^\nu(\boldsymbol{\lambda}(\boldsymbol{y}^{(t)}; \boldsymbol{x}))\right), \nabla_{\boldsymbol{y}}F^\nu(\boldsymbol{\lambda}(\boldsymbol{y}^{(t)}; \boldsymbol{x})) - \hat{\boldsymbol{g}}_{\boldsymbol{y}}^{(t)}) \right\rangle \big| \boldsymbol{y}^{(t)} \right]$$

$$+ \eta_y^2\mathbb{E}\left[ \left\| \nabla_{\boldsymbol{y}}F^\nu(\boldsymbol{\lambda}(\boldsymbol{y}^{(t)}; \boldsymbol{x})) - \hat{\boldsymbol{g}}_{\boldsymbol{y}}^{(t)}) \right\|^2 \big| \boldsymbol{y}^{(t)} \right]. \tag{26}$$

Where (25) follows from Lemma 3.2 in [34]. For the first part of (26), we have

$$\left\| (1 - \eta_y\beta)(\boldsymbol{y}^{(t)} - \hat{\boldsymbol{y}}^{(t)}) + \eta_y\left(\nabla_{\boldsymbol{y}}F^\nu(\boldsymbol{\lambda}(\hat{\boldsymbol{y}}^{(t)}; \boldsymbol{x})) - \nabla_{\boldsymbol{y}}F^\nu(\boldsymbol{\lambda}(\boldsymbol{y}^{(t)}; \boldsymbol{x}))\right) \right\|^2$$

$$= (1 - \eta_y\beta)^2 \left\| \boldsymbol{y}^{(t)} - \hat{\boldsymbol{y}}^{(t)} \right\|^2 + \eta_y^2 \left\| \nabla_{\boldsymbol{y}}F^\nu(\boldsymbol{\lambda}(\hat{\boldsymbol{y}}^{(t)}; \boldsymbol{x})) - \nabla_{\boldsymbol{y}}F^\nu(\boldsymbol{\lambda}(\boldsymbol{y}^{(t)}; \boldsymbol{x})) \right\|^2$$

$$+ 2(1 - \eta_y\beta)\eta_y \left\langle \boldsymbol{y}^{(t)} - \hat{\boldsymbol{y}}^{(t)}, \nabla_{\boldsymbol{y}}F^\nu(\boldsymbol{\lambda}(\hat{\boldsymbol{y}}^{(t)}; \boldsymbol{x})) - \nabla_{\boldsymbol{y}}F^\nu(\boldsymbol{\lambda}(\boldsymbol{y}^{(t)}; \boldsymbol{x})) \right\rangle$$

$$\leq (1 - \eta_y\beta)^2 \left\| \boldsymbol{y}^{(t)} - \hat{\boldsymbol{y}}^{(t)} \right\|^2 + \eta_y^2\ell_\nu^2 \left\| \boldsymbol{y}^{(t)} - \hat{\boldsymbol{y}}^{(t)} \right\|^2 + 2(1 - \eta_y\beta)\eta_y\ell_\nu \left\| \boldsymbol{y}^{(t)} - \hat{\boldsymbol{y}}^{(t)} \right\|^2 \tag{27}$$

$$= (1 - \eta_y\beta)\left( 1 - \eta_y\beta + 2\eta_y\ell_\nu + \frac{\eta_y^2\ell_\nu^2}{1 - \eta_y\beta} \right) \left\| \boldsymbol{y}^{(t)} - \hat{\boldsymbol{y}}^{(t)} \right\|^2.$$

Where (27) follows from Lemma C.4. By setting $\eta_y, \beta$ such that $2\eta_y\ell_\nu \leq \frac{\eta_y\beta}{2}$, and $\frac{\eta_y^2\ell_\nu^2}{1 - \eta_y\beta} \leq \frac{\eta_y\beta}{2}$, we have

$$\left\| (1 - \eta_y\beta)(\boldsymbol{y}^{(t)} - \hat{\boldsymbol{y}}^{(t)}) + \eta_y(\nabla_{\boldsymbol{y}}F^\nu(\boldsymbol{\lambda}(\hat{\boldsymbol{y}}^{(t)}; \boldsymbol{x})) - \nabla_{\boldsymbol{y}}F^\nu(\boldsymbol{\lambda}(\boldsymbol{y}^{(t)}; \boldsymbol{x}))) \right\|^2$$

$$\leq (1 - \eta_y\beta) \left\| \boldsymbol{y}^{(t)} - \hat{\boldsymbol{y}}^{(t)} \right\|^2. \tag{28}$$

For the third part in (26), we have

$$2(1-\eta_y\beta)\eta_y\mathbb{E}\big[\big\langle(\boldsymbol{y}^{(t)}-\hat{\boldsymbol{y}}^{(t)})-\eta_y\big(\nabla_{\boldsymbol{y}}F^\nu(\boldsymbol{\lambda}(\hat{\boldsymbol{y}}^{(t)};\boldsymbol{x}))-\nabla_{\boldsymbol{y}}F^\nu(\boldsymbol{\lambda}(\boldsymbol{y}^{(t)};\boldsymbol{x}))\big),\nabla_{\boldsymbol{y}}F^\nu(\boldsymbol{\lambda}(\boldsymbol{y}^{(t)};\boldsymbol{x}))-\hat{\boldsymbol{g}}_{\boldsymbol{y}}^{(t)}\big\rangle\big|\boldsymbol{y}^{(t)}\big]$$

$$\leq 2(1-\eta_y\beta)\eta_y\big\|(\boldsymbol{y}^{(t)}-\hat{\boldsymbol{y}}^{(t)})-\eta_y\big(\nabla_{\boldsymbol{y}}F^\nu(\boldsymbol{\lambda}(\hat{\boldsymbol{y}}^{(t)};\boldsymbol{x}))-\nabla_{\boldsymbol{y}}F^\nu(\boldsymbol{\lambda}(\boldsymbol{y}^{(t)};\boldsymbol{x}))\big)\big\|\cdot\mathbb{E}\big[\big\|\nabla_{\boldsymbol{y}}F^\nu(\boldsymbol{\lambda}(\boldsymbol{y}^{(t)};\boldsymbol{x}))-\hat{\boldsymbol{g}}_{\boldsymbol{y}}^{(t)}\big)\big\|\big|\boldsymbol{y}^{(t)}\big]$$

$$\leq 2(1-\eta_y\beta)\eta_y(1+\eta_y\ell_\nu)\cdot\big\|\boldsymbol{y}^{(t)}-\hat{\boldsymbol{y}}^{(t)}\big\|\cdot\mathbb{E}\big[\big\|\nabla_{\boldsymbol{y}}F^\nu(\boldsymbol{\lambda}(\boldsymbol{y}^{(t)};\boldsymbol{x}))-\hat{\boldsymbol{g}}_{\boldsymbol{y}}^{(t)}\big)\big\|\big] \tag{29}$$

$$\leq 2(1-\eta_y\beta)\eta_y(1+\eta_y\ell_\nu)\cdot\mathcal{C}_3\gamma^H. \tag{30}$$

Where

- $\mathcal{C}_3 = \sqrt{|\mathcal{S}||\mathcal{B}|}\frac{6H}{(1-\gamma)^3\zeta}$;
- (29) follows from Lemma C.4;
- (30) is due to Lemma C.14.

Combine (26), (28), and (30), we have

$$\mathbb{E}\left[\left\|\boldsymbol{y}^{t+1}-\hat{\boldsymbol{y}}^{(t)}\right\|^2\Big|\boldsymbol{y}^{(t)}\right] \leq (1-\eta_y\beta)\left\|\boldsymbol{y}^{(t)}-\hat{\boldsymbol{y}}^{(t)}\right\|^2 + 2(1-\eta_y\beta)\eta_y(1+\eta_y\ell_\nu)\cdot\mathcal{C}_3\gamma^H$$

$$+ \eta_y^2\mathbb{E}\left[\left\|\nabla_{\boldsymbol{y}}F^\nu(\boldsymbol{\lambda}(\boldsymbol{y}^{(t)};\boldsymbol{x}))-\hat{\boldsymbol{g}}_{\boldsymbol{y}}^{(t)}\right\|^2\Big|\boldsymbol{y}^{(t)}\right].$$

$\square$

We then show the result for convergence to optimality for (21).

**Theorem C.4.** *By setting* $\eta_y = \frac{\nu\epsilon}{10\ell_\nu\sigma^2L_{\lambda_{\text{inv}}}^2}$ *and* $H = \frac{2\log(1/\nu\epsilon)}{1-\gamma}$. *After running Algorithm 2 for* $T = \mathcal{O}\left(\frac{\ell_\nu L_{\lambda_{\text{inv}}}^2}{\nu}\log\left(\frac{1}{\epsilon}\right) + \frac{\ell_\nu\sigma^2L_{\lambda_{\text{inv}}}^4}{\nu^2\epsilon}\log\left(\frac{1}{\epsilon}\right)\right)$ *iterations, we have*

$$\mathbb{E}\left[F^\nu(\boldsymbol{\lambda}(\boldsymbol{y}_\zeta^\star;\boldsymbol{x})) - F^\nu(\boldsymbol{\lambda}(\boldsymbol{y}^{(T)};\boldsymbol{x}))\right] \leq \epsilon.$$

*Where* $\boldsymbol{y}_\zeta^\star$ *is the unique maximizer for the optimization problem* (21).

**Proof.** From Theorem 1 in [43], by setting $\beta = 4\ell_\nu$, $\alpha \leq 2\eta_y\ell_\nu$, and $\eta_y \leq \frac{2}{9\ell_\nu}$. Then for any $\boldsymbol{z} \in \mathcal{Y}^\zeta$, we have

$$\mathbb{E}\left[F_{1/\beta}^\nu(\boldsymbol{\lambda}(\boldsymbol{y}^{(t+1)};\boldsymbol{x}))\right]$$

$$\geq \mathbb{E}\left[F^\nu(\boldsymbol{\lambda}(\boldsymbol{z};\boldsymbol{x})) - (1+s)\frac{\beta}{2}\left\|\hat{\boldsymbol{y}}^{(t)}-\boldsymbol{y}^{(t+1)}\right\|^2 - \left(1+\frac{1}{s}\right)\frac{\beta}{2}\left\|\hat{\boldsymbol{y}}^{(t)}-\boldsymbol{z}\right\|^2\right]$$

$$\geq \mathbb{E}\left[F^\nu(\boldsymbol{\lambda}(\boldsymbol{z};\boldsymbol{x})) - (1+s)(1-\eta_y\beta)\frac{\beta}{2}\left\|\boldsymbol{y}^{(t)}-\hat{\boldsymbol{y}}^{(t)}\right\|^2\right]$$

$$- \left(1+\frac{1}{s}\right)\frac{\beta}{2}\mathbb{E}[\left\|\hat{\boldsymbol{y}}^{(t)}-\boldsymbol{z}\right\|^2] - 2\eta_y(1+s)(1-\eta_y\beta)(1+\eta_y\ell_\nu)\cdot\mathcal{C}_3\gamma^H$$

$$- (1+s)\frac{\beta}{2}\eta_y^2\mathbb{E}\left[\left\|\nabla_{\boldsymbol{y}}F^\nu(\boldsymbol{\lambda}(\boldsymbol{y}^{(t)};\boldsymbol{x}))-\hat{\boldsymbol{g}}_{\boldsymbol{y}}^{(t)}\right\|^2\Big|\boldsymbol{y}^{(t)}\right]. \tag{31}$$

Where (31) is due to Lemma C.11. By setting $s = \frac{\eta_y\beta}{2}$, we have $(1+s)(1-\eta_y\beta) \leq 1 - \frac{\eta_y\beta}{2}$, $1+s \leq 2$, and $1+\frac{1}{s} \leq \frac{3}{\eta_y\beta}$. From Theorem 1 in [43], we get

$$\mathbb{E}\left[F_{1/\beta}^\nu(\boldsymbol{\lambda}(\boldsymbol{y}^{(t+1)};\boldsymbol{x}))\right]$$

$$\geq (1-\alpha)\mathbb{E}\left[F_{1/\beta}^\nu(\boldsymbol{\lambda}(\boldsymbol{y}^{(t)};\boldsymbol{x}))\right] + \alpha F^\nu(\boldsymbol{\lambda}(\boldsymbol{y}_\zeta^\star;\boldsymbol{x})) - \beta\eta_y^2\mathbb{E}\left[\left\|\nabla_{\boldsymbol{y}}F^\nu(\boldsymbol{\lambda}(\boldsymbol{y}^{(t)};\boldsymbol{x}))-\hat{\boldsymbol{g}}_{\boldsymbol{y}}^{(t)}\right\|^2\Big|\boldsymbol{y}^{(t)}\right]$$

$$- \left(\frac{3L_{\lambda_{\text{inv}}}^2\alpha^2}{2\eta_y} - \frac{(1-\alpha)\alpha\nu}{2}\right)\mathbb{E}\left[\left\|\boldsymbol{\lambda}(\hat{\boldsymbol{y}}^{(t)};\boldsymbol{x})-\boldsymbol{\lambda}(\boldsymbol{y}_\zeta^\star;\boldsymbol{x})\right\|^2\right] - 2\eta_y(1-\alpha)(1+\eta_y\ell_\nu)\cdot\mathcal{C}_3\gamma^H.$$

Define $\Lambda_t := \mathbb{E}\left[F^\nu(\boldsymbol{\lambda}(\boldsymbol{y}_\zeta^\star; \boldsymbol{x})) - F_{1/\beta}^\nu(\boldsymbol{\lambda}(\boldsymbol{y}^{(t)}; \boldsymbol{x}))\right]$, by setting $\left(\frac{3L_{\lambda_{\text{inv}}}^2 \alpha^2}{2\eta_y} - \frac{(1-\alpha)\alpha\nu}{2}\right) \leq 0$, we have

$$\Lambda_{t+1} \leq (1-\alpha)\Lambda_t + \beta\eta_y^2 \mathbb{E}\left[\left\|\nabla_{\boldsymbol{y}} F^\nu(\boldsymbol{\lambda}(\boldsymbol{y}^{(t)}; \boldsymbol{x})) - \hat{\boldsymbol{g}}_{\boldsymbol{y}}^{(t)}\right\|^2 \Big| \boldsymbol{y}^{(t)}\right] + 2\eta_y(1-\alpha)(1 + \eta_y\ell_\nu) \cdot \mathcal{C}_3 \gamma^H.$$

Summing over $T$ iterations, and denote $\mathbb{E}\left[\left\|\nabla_{\boldsymbol{y}} F^\nu(\boldsymbol{\lambda}(\boldsymbol{y}^{(t)}; \boldsymbol{x})) - \hat{\boldsymbol{g}}_{\boldsymbol{y}}^{(t)}\right\|^2 \Big| \boldsymbol{y}^{(t)}\right] = \sigma^2$, we get

$$\Lambda_T \leq (1-\alpha)^T \Lambda_0 + \frac{4\ell_\nu \eta_y^2}{\alpha}\sigma^2 + \frac{2\eta_y(1-\alpha)(1+\eta_y\ell_\nu)}{\alpha} \cdot \mathcal{C}_3 \gamma^H.$$

By setting $H = \frac{2\log(1/\nu\epsilon)}{1-\gamma}$, $\alpha \leq \min\left\{2\eta_y\ell_\nu, \frac{\nu\eta_y}{2L_{\lambda_{\text{inv}}}^2}\right\}$, and $\eta_y = \frac{2}{9\ell_\nu}$, $\frac{\nu\epsilon}{10\ell_\nu\sigma^2 L_{\lambda_{\text{inv}}}^2}$, after

$$T = \mathcal{O}\left(\frac{\ell_\nu L_{\lambda_{\text{inv}}}^2}{\nu}\log\left(\frac{1}{\epsilon}\right) + \frac{\ell_\nu \sigma^2 L_{\lambda_{\text{inv}}}^4}{\nu^2\epsilon}\log\left(\frac{1}{\epsilon}\right)\right).$$

iterations, we get $\Lambda_T \leq \epsilon$. Where $\sigma^2 = \frac{\mathcal{C}_1}{K} + \mathcal{C}_2 \cdot \gamma^{2H}$, $\mathcal{C}_1 = \frac{57}{(1-\gamma)^6\zeta^2}$, $\mathcal{C}_2 = \frac{126H^2}{(1-\gamma)^6\zeta^2}$, $\mathcal{C}_3 = \sqrt{|\mathcal{S}||\mathcal{B}|}\frac{6H}{(1-\gamma)^3\zeta}$. Since $F^\nu(\boldsymbol{\lambda}(\boldsymbol{y}; \boldsymbol{x}))$ is smooth with respect to the state-action visitation measure $\boldsymbol{\lambda}(\boldsymbol{y}; \boldsymbol{x})$. We have

$$\begin{aligned}
\Lambda_T &= \mathbb{E}\left[F^\nu(\boldsymbol{\lambda}(\boldsymbol{y}_\zeta^\star; \boldsymbol{x})) - F_{1/\beta}^\nu(\boldsymbol{\lambda}(\boldsymbol{y}^{(T)}; \boldsymbol{x}))\right] \\
&= \mathbb{E}\left[F^\nu(\boldsymbol{\lambda}(\boldsymbol{y}_\zeta^\star; \boldsymbol{x})) - \max_{\boldsymbol{z}\in\mathcal{Y}}\left\{F^\nu(\boldsymbol{\lambda}(\boldsymbol{z}; \boldsymbol{x})) - \frac{\beta}{2}\left\|\boldsymbol{\lambda}(\boldsymbol{z}; \boldsymbol{x}) - \boldsymbol{\lambda}(\boldsymbol{y}^{(T)}; \boldsymbol{x})\right\|^2\right\}\right] \\
&\geq \mathbb{E}\left[F^\nu(\boldsymbol{\lambda}(\boldsymbol{y}_\zeta^\star; \boldsymbol{x})) - F^\nu(\boldsymbol{\lambda}(\boldsymbol{y}^{(T)}; \boldsymbol{x})) + \frac{\beta}{2}\left\|\boldsymbol{\lambda}(\boldsymbol{y}^{(T)}; \boldsymbol{x}) - \boldsymbol{\lambda}(\boldsymbol{y}^{(T)}; \boldsymbol{x})\right\|^2\right] \\
&= \mathbb{E}\left[F^\nu(\boldsymbol{\lambda}(\boldsymbol{y}_\zeta^\star; \boldsymbol{x})) - F^\nu(\boldsymbol{\lambda}(\boldsymbol{y}^{(T)}; \boldsymbol{x}))\right].
\end{aligned}$$

Therefore

$$\mathbb{E}\left[F^\nu(\boldsymbol{\lambda}(\boldsymbol{y}_\zeta^\star; \boldsymbol{x})) - F^\nu(\boldsymbol{\lambda}(\boldsymbol{y}^{(T)}; \boldsymbol{x}))\right] \leq \Lambda_T \leq \epsilon.$$

$\square$

Define $\boldsymbol{y}^\star \in \mathcal{Y}$ such that $\boldsymbol{y}^\star = \operatorname{argmax}_{\boldsymbol{y}\in\mathcal{Y}}\left\{\boldsymbol{r}(\boldsymbol{x})^\top \boldsymbol{\lambda}(\boldsymbol{y}; \boldsymbol{x}) - \frac{\nu}{2}\|\boldsymbol{\lambda}(\boldsymbol{y}; \boldsymbol{x})\|^2\right\}$. We bound the distance between the the optimal $\boldsymbol{y}^\star$ and $\boldsymbol{y}^{(T)}$ from Algorithm 2.

**Lemma C.12.** For any $\boldsymbol{y} \in \mathcal{Y}^\zeta$, if $\mathbb{E}\left[F^\nu(\boldsymbol{\lambda}(\boldsymbol{y}_\zeta^\star; \boldsymbol{x})) - F^\nu(\boldsymbol{\lambda}(\boldsymbol{y}; \boldsymbol{x}))\right] \leq \epsilon$, then we have

$$\mathbb{E}\left[\|\boldsymbol{y}^\star - \boldsymbol{y}\|\right] \leq L_{\lambda_{\text{inv}}}\left(\sqrt{\frac{8L_\lambda |\mathcal{B}|\zeta}{(1-\gamma)\nu}} + \sqrt{\frac{2\epsilon}{\nu}}\right).$$

**Proof.** Since $F^\nu(\boldsymbol{\lambda}(\boldsymbol{y}; \boldsymbol{x}))$ is $\nu$-strongly concave with respect to $\boldsymbol{\lambda}(\boldsymbol{y}; \boldsymbol{x})$, we have

$$\begin{aligned}
F^\nu(\boldsymbol{\lambda}(\boldsymbol{y}_\zeta^\star; \boldsymbol{x})) &\geq F^\nu(\boldsymbol{\lambda}(\boldsymbol{y}; \boldsymbol{x})) + \left\langle \nabla_{\boldsymbol{\lambda}} F^\nu(\boldsymbol{\lambda}(\boldsymbol{y}_\zeta^\star; \boldsymbol{x})), \boldsymbol{\lambda}(\boldsymbol{y}_\zeta^\star; \boldsymbol{x}) - \boldsymbol{\lambda}(\boldsymbol{y}; \boldsymbol{x})\right\rangle \\
&\quad + \frac{\nu}{2}\left\|\boldsymbol{\lambda}(\boldsymbol{y}_\zeta^\star; \boldsymbol{x}) - \boldsymbol{\lambda}(\boldsymbol{y}; \boldsymbol{x})\right\|^2 \\
&= F^\nu(\boldsymbol{\lambda}(\boldsymbol{y}; \boldsymbol{x})) + \frac{\nu}{2}\left\|\boldsymbol{\lambda}(\boldsymbol{y}_\zeta^\star; \boldsymbol{x}) - \boldsymbol{\lambda}(\boldsymbol{y}; \boldsymbol{x})\right\|^2.
\end{aligned} \tag{32}$$

Where (32) holds because $\boldsymbol{\lambda}(\boldsymbol{y}_\zeta^\star; \boldsymbol{x})$ is the optimal solution for $F^\nu(\boldsymbol{\lambda}(\boldsymbol{y}; \boldsymbol{x}))$ for any $\boldsymbol{y} \in \mathcal{Y}$. Therefore

$$\mathbb{E}\left[\left\|\boldsymbol{\lambda}(\boldsymbol{y}_\zeta^\star; \boldsymbol{x}) - \boldsymbol{\lambda}(\boldsymbol{y}; \boldsymbol{x})\right\|\right] \leq \sqrt{\frac{2}{\nu} \cdot \mathbb{E}\left[F^\nu(\boldsymbol{\lambda}(\boldsymbol{y}_\zeta^\star; \boldsymbol{x})) - F^\nu(\boldsymbol{\lambda}(\boldsymbol{y}; \boldsymbol{x}))\right]} \leq \sqrt{\frac{2\epsilon}{\nu}}.$$

From the definition of $\boldsymbol{\lambda}(\boldsymbol{y}_\zeta^\star; \boldsymbol{x})$, it holds that for all $\boldsymbol{y}_\zeta \in \mathcal{Y}^\zeta$, we have $\left\langle -\nabla_{\boldsymbol{\lambda}} F^\nu(\boldsymbol{\lambda}(\boldsymbol{y}_\zeta^\star; \boldsymbol{x})), \boldsymbol{\lambda}(\boldsymbol{y}_\zeta^\star; \boldsymbol{x}) - \boldsymbol{\lambda}(\boldsymbol{y}_\zeta; \boldsymbol{x}) \right\rangle \leq 0$. Combine with Lemma C.6 and consider $\boldsymbol{y}^\star \in \mathcal{Y}$, we have

$$
\begin{aligned}
&\left\langle -\nabla_{\boldsymbol{\lambda}} F^\nu(\boldsymbol{\lambda}(\boldsymbol{y}_\zeta^\star; \boldsymbol{x})), \boldsymbol{\lambda}(\boldsymbol{y}_\zeta^\star; \boldsymbol{x}) - \boldsymbol{\lambda}(\boldsymbol{y}^\star; \boldsymbol{x}) \right\rangle \\
&= \left\langle -\nabla_{\boldsymbol{\lambda}} F^\nu(\boldsymbol{\lambda}(\boldsymbol{y}_\zeta^\star; \boldsymbol{x})), \boldsymbol{\lambda}(\boldsymbol{y}_\zeta^\star; \boldsymbol{x}) - \boldsymbol{\lambda}(\boldsymbol{y}_\zeta; \boldsymbol{x}) + \boldsymbol{\lambda}(\boldsymbol{y}_\zeta; \boldsymbol{x}) - \boldsymbol{\lambda}(\boldsymbol{y}^\star; \boldsymbol{x}) \right\rangle \quad (33) \\
&= \left\langle -\nabla_{\boldsymbol{\lambda}} F^\nu(\boldsymbol{\lambda}(\boldsymbol{y}_\zeta^\star; \boldsymbol{x})), \boldsymbol{\lambda}(\boldsymbol{y}_\zeta^\star; \boldsymbol{x}) - \boldsymbol{\lambda}(\boldsymbol{y}_\zeta; \boldsymbol{x}) \right\rangle + \left\langle -\nabla_{\boldsymbol{\lambda}} F^\nu(\boldsymbol{\lambda}(\boldsymbol{y}_\zeta^\star; \boldsymbol{x})), \boldsymbol{\lambda}(\boldsymbol{y}_\zeta; \boldsymbol{x}) - \boldsymbol{\lambda}(\boldsymbol{y}^\star; \boldsymbol{x}) \right\rangle \\
&\leq \left\langle -\nabla_{\boldsymbol{\lambda}} F^\nu(\boldsymbol{\lambda}(\boldsymbol{y}_\zeta^\star; \boldsymbol{x})), \boldsymbol{\lambda}(\boldsymbol{y}_\zeta; \boldsymbol{x}) - \boldsymbol{\lambda}(\boldsymbol{y}^\star; \boldsymbol{x}) \right\rangle \\
&\leq \left\| \nabla_{\boldsymbol{\lambda}} F^\nu(\boldsymbol{\lambda}(\boldsymbol{y}_\zeta^\star; \boldsymbol{x})) \right\| \cdot \left\| \boldsymbol{\lambda}(\boldsymbol{y}_\zeta; \boldsymbol{x}) - \boldsymbol{\lambda}(\boldsymbol{y}^\star; \boldsymbol{x}) \right\| \\
&\leq \frac{4L_\lambda |\mathcal{B}| \zeta}{1 - \gamma}. \quad (34)
\end{aligned}
$$

Where

- in (33) $\boldsymbol{y}_\zeta \in \mathcal{Y}^\zeta$ is chosen such that $\|\boldsymbol{y}^\star - \boldsymbol{y}_\zeta\| \leq 2\zeta|\mathcal{B}|$ according to C.6;
- (34) holds because $\|\nabla_{\boldsymbol{\lambda}} F^\nu(\boldsymbol{\lambda})\| \leq \frac{2}{1-\gamma}$ and $\boldsymbol{\lambda}(\boldsymbol{y}; \boldsymbol{x})$ is $L_\lambda$-continuous.

Since $F^\nu(\boldsymbol{\lambda})$ is $\nu$-strongly concave w.r.t $\boldsymbol{\lambda}$, we have

$$
\begin{aligned}
&\frac{\nu}{2} \left\| \boldsymbol{\lambda}(\boldsymbol{y}_\zeta^\star; \boldsymbol{x}) - \boldsymbol{\lambda}(\boldsymbol{y}^\star; \boldsymbol{x}) \right\|^2 \\
&\leq F^\nu(\boldsymbol{\lambda}(\boldsymbol{y}_\zeta^\star; \boldsymbol{x})) - F^\nu(\boldsymbol{\lambda}(\boldsymbol{y}^\star; \boldsymbol{x}) + \left\langle \nabla_{\boldsymbol{\lambda}} F^\nu((\boldsymbol{\lambda}(\boldsymbol{y}_\zeta^\star; \boldsymbol{x})), \boldsymbol{\lambda}(\boldsymbol{y}^\star; \boldsymbol{x}) - \boldsymbol{\lambda}(\boldsymbol{y}_\zeta^\star; \boldsymbol{x}) \right\rangle \\
&\leq \frac{4L_\lambda |\mathcal{B}| \zeta}{1 - \gamma}.
\end{aligned}
$$

Thus we conclude that

$$
\begin{aligned}
\mathbb{E}\left[ \|\boldsymbol{y}^\star - \boldsymbol{y}\| \right] &\leq L_{\lambda_{\mathrm{inv}}} \mathbb{E}\left[ \|\boldsymbol{\lambda}(\boldsymbol{y}^\star; \boldsymbol{x}) - \boldsymbol{\lambda}(\boldsymbol{y}; \boldsymbol{x})\| \right] \\
&\leq L_{\lambda_{\mathrm{inv}}} \left( \left\| \boldsymbol{\lambda}(\boldsymbol{y}^\star; \boldsymbol{x}) - \boldsymbol{\lambda}(\boldsymbol{y}_\zeta^\star; \boldsymbol{x}) \right\| + \mathbb{E}\left[ \left\| \boldsymbol{\lambda}(\boldsymbol{y}_\zeta^\star; \boldsymbol{x}) - \boldsymbol{\lambda}(\boldsymbol{y}; \boldsymbol{x}) \right\| \right] \right) \\
&\leq L_{\lambda_{\mathrm{inv}}} \left( \sqrt{\frac{8L_\lambda |\mathcal{B}| \zeta}{(1 - \gamma)\nu}} + \sqrt{\frac{2\epsilon}{\nu}} \right).
\end{aligned}
$$

$\square$

## C.6  Regarding the Gradient and Visitation Estimators

In this subsection we will quantify the bias and variance of the gradient and state-action visitation estimators used in Algorithms 1 and 2. In particular, REINFORCE:

- the gradient estimator for team agents is implemented by sampling a trajectory with horizon length, $H$, that is drawn from a geometric distribution for the team, and
- while, the state-action visitation estimators that the adversary uses come from sampled trajectories of a fixed horizon length $H$.

In the former case, the estimator is unbiased while in the second case the bias decays exponentially in $H$.

### C.6.1  REINFORCE for Vanilla Policy Gradient

In the present work, the team agents only need to implement a batch version of REINFORCE [104]. That is, they get estimates $\hat{\boldsymbol{g}}_k^{(t)} = \frac{1}{M} \sum_{j=1}^{M} \tilde{\boldsymbol{g}}_k^{(t)}$, where:

$$\tilde{\boldsymbol{g}}_{i,j}^{(t)} = \sum_{h_j=1}^{H_j} r_i^{(h_j)} \sum_{h=1}^{H_j} \nabla \log x_i \left( a^{(h_j)} | s^{(h_j)} \right), \qquad \text{(REINFORCE)}$$

with each $H_j$ is a random variable following a geometric distribution with parameter $(1 - \gamma)$.

Although the authors of [27] use $\zeta$-greedy parametrization in order to bound the variance of the estimator, policies drawn from the $\zeta$-truncated simplex imply the same inequality needed to bound the variance. Hence, we invoke the corresponding lemma.

**Lemma C.13** ([27, Lemma 2]). When Equation (REINFORCE) is implemented with $H$ following a geometric distribution with a parameter $1 - \gamma$, and agent $k$ selects policies from the $\zeta$-truncated simplex on each state, it is the case that the gradient estimates satisfy:

$$\mathbb{E}\left[ \hat{\boldsymbol{g}}_k^{(t)} \right] - \nabla_{\boldsymbol{x}_k} V_{\boldsymbol{\rho}}(\boldsymbol{x}^t, \boldsymbol{y}^t) = 0;$$

$$\mathbb{E}\left[ \left\| \hat{\boldsymbol{g}}_k^{(t)} - \nabla_{\boldsymbol{x}_k} V_{\boldsymbol{\rho}}(\boldsymbol{x}^t, \boldsymbol{y}^t) \right\|^2 \right] \le 24 \frac{|\mathcal{A}_k^2|}{\zeta(1 - \gamma)}.$$

### C.6.2  Gradient Estimation for Visitation-Regularized Policy Gradient

In this subsection we will describe (i) a state-action visitation estimator with bounded bias and variance and (ii) a gradient estimator of the regularized value function whose bias and variance are also bounded.

Bounding the variance of the a gradient estimator with a deterministic choice of $H$ was significantly less demanding than doing so with a randomized choice. This comes at the cost with a non-zero bias that nevertheless decays exponentially in $H$. For any policy of the adversary $\boldsymbol{y} \in \mathcal{Y}$, we introduce the $H$-horizon truncated state-action visitation measure

$$\lambda_{H,s,b}(\boldsymbol{y}; \boldsymbol{x})_{s,b} := \sum_{h=0}^{H-1} \gamma^h \, \mathbb{P}(s_h = s, b_h = b | \boldsymbol{y}, s_0 \sim \boldsymbol{\rho}). \qquad (35)$$

Where $\lambda_{H,s,b}(\boldsymbol{y}; \boldsymbol{x})$ denotes the $(s, b)^{th}$ entry of $\lambda_H(\boldsymbol{y}; \boldsymbol{x})$. For any reward vector $\boldsymbol{r}$, we have

$$[\nabla_{\boldsymbol{y}} \boldsymbol{\lambda}_H(\boldsymbol{y}; \boldsymbol{x})]^\top \boldsymbol{r} = \mathbb{E}\left[ \sum_{h=0}^{H-1} \gamma^h \cdot r(s_h, b_h) \cdot \left( \sum_{h'=0}^{h} \nabla_{\boldsymbol{y}} \log y(b_{h'} | s_{h'}) \right) \bigg| \boldsymbol{y}, s_0 \sim \boldsymbol{\rho} \right].$$

### C.6.3  Controlling the Estimation Bias and Variance

In this subsection we will present a detailed analysis regarding estimators defined in Definition 2.6, Definition 2.7, and the ones used in Algorithm 2. Particularly, in Lemma C.14 we bound the bias of aforementioned estimators. This bias is inevitable for our analysis due to the fact we are

sampling trajectories of a finite length $H$ over an infinite horizon. Proceeding to Lemma C.16 and Lemma C.17, we bound the variance of the state-action distribution measure estimator $\hat{\boldsymbol{\lambda}}^{(t)}$ and the gradient estimator $\hat{\boldsymbol{g}}_{\boldsymbol{y}}^{(t)}$ w.r.t their biased means. Finally, in Lemma C.18, we bound the distance between the gradient estimator $\hat{\boldsymbol{g}}_{\boldsymbol{y}}^{(t)}$ and the actual gradient $\nabla_{\boldsymbol{y}} F^\nu(\boldsymbol{\lambda}(\boldsymbol{y}^{(t)}; \boldsymbol{x}))$.

**Lemma C.14** (Bounded Bias of the Estimators). For any adversary's policy $\boldsymbol{y} \in \mathcal{Y}$. We let $\tau = (s_0, b_0, s_1, b_1, \cdots, s_{H-1}, b_{H-1})$ be an $H$-length trajectory sampled from $\boldsymbol{y}$, then we have $\mathbb{E}_{\tau \sim \boldsymbol{y}}\left[\tilde{\boldsymbol{\lambda}}(\tau|\boldsymbol{y})\right] = \boldsymbol{\lambda}_H(\boldsymbol{y}; \boldsymbol{x})$ and $\mathbb{E}_{\tau \sim \boldsymbol{y}}\left[\tilde{\boldsymbol{g}}(\tau|\boldsymbol{y}; \boldsymbol{r})\right] = [\nabla_{\boldsymbol{y}} \boldsymbol{\lambda}_H(\boldsymbol{y}; \boldsymbol{x})]^\top \boldsymbol{r}$. This implies that in Algorithm 2, $\mathbb{E}\left[\hat{\boldsymbol{\lambda}}^{(t)}\right] = \boldsymbol{\lambda}_H(\boldsymbol{y}^{(t)}; \boldsymbol{x})$ and $\mathbb{E}\left[\hat{\boldsymbol{g}}_{\boldsymbol{y}}^{(t)}\right] = [\nabla_{\boldsymbol{y}} \boldsymbol{\lambda}_H(\boldsymbol{y}^{(t)}; \boldsymbol{x})]^\top \boldsymbol{r}^{(t)}$. Moreover, we have:

- $\left\|\mathbb{E}\left[\hat{\boldsymbol{\lambda}}^{(t)}\right] - \boldsymbol{\lambda}(\boldsymbol{y}^{(t)}; \boldsymbol{x})\right\| \le \frac{\gamma^H}{1-\gamma}$, and

- $\left\|\mathbb{E}\left[\hat{\boldsymbol{g}}_{\boldsymbol{y}}^{(t)}\right] - \nabla_{\boldsymbol{y}} F^\nu(\boldsymbol{\lambda}(\boldsymbol{y}^{(t)}; \boldsymbol{x}))\right\| \le \left(\frac{H+1}{(1-\gamma)\zeta} + \frac{\nu H + \nu + 1}{(1-\gamma)^2 \zeta} + \frac{\nu}{(1-\gamma)^3 \zeta}\right) \cdot \gamma^H$.

**Proof.** From the definition, we have

$$\mathbb{E}_{\tau \sim \boldsymbol{y}}\left[\tilde{\boldsymbol{\lambda}}(\tau|\boldsymbol{y})\right] = \boldsymbol{\lambda}_H(\boldsymbol{y}; \boldsymbol{x}), \quad \mathbb{E}_{\tau \sim \boldsymbol{y}}\left[\tilde{\boldsymbol{g}}(\tau|\boldsymbol{y}; \boldsymbol{r})\right] = [\nabla_{\boldsymbol{y}} \boldsymbol{\lambda}_H(\boldsymbol{y}; \boldsymbol{x})]^\top \boldsymbol{r}.$$

Therefore,

$$\mathbb{E}\left[\hat{\boldsymbol{\lambda}}^{(t)}\right] = \boldsymbol{\lambda}_H(\boldsymbol{y}^{(t)}; \boldsymbol{x}), \quad \mathbb{E}\left[\hat{\boldsymbol{g}}_{\boldsymbol{y}}^{(t)}\right] = [\nabla_{\boldsymbol{y}} \boldsymbol{\lambda}_H(\boldsymbol{y}^{(t)}; \boldsymbol{x})]^\top \boldsymbol{r}^{(t)}.$$

Then it holds that

$$\left\|\mathbb{E}\left[\hat{\boldsymbol{\lambda}}^{(t)}\right] - \boldsymbol{\lambda}(\boldsymbol{y}^{(t)}; \boldsymbol{x})\right\| = \left\|\boldsymbol{\lambda}_H(\boldsymbol{y}^{(t)}; \boldsymbol{x}) - \boldsymbol{\lambda}(\boldsymbol{y}^{(t)}; \boldsymbol{x})\right\|$$
$$= \left\|\sum_{h=H}^{\infty} \gamma^h \cdot \mathbb{P}(s_h = s, b_h = b|\boldsymbol{y}^{(t)}, s_0 \sim \boldsymbol{\rho}) \cdot \boldsymbol{e}_{s_h, b_h}\right\|$$
$$\le \gamma^H \cdot \sum_{h=0}^{\infty} (\gamma^h \cdot 1)$$
$$= \frac{\gamma^H}{1 - \gamma}.$$

Similarly, we have

$$\left\|\mathbb{E}\left[\hat{\boldsymbol{g}}_{\boldsymbol{y}}^{(t)}\right] - \nabla_{\boldsymbol{y}} F^\nu(\boldsymbol{\lambda}(\boldsymbol{y}^{(t)}; \boldsymbol{x}))\right\|$$
$$= \left\|[\nabla_{\boldsymbol{y}} \boldsymbol{\lambda}_H(\boldsymbol{y}^{(t)}; \boldsymbol{x})]^\top \boldsymbol{r}^{(t)} - \nabla_{\boldsymbol{y}} F^\nu(\boldsymbol{\lambda}(\boldsymbol{y}^{(t)}; \boldsymbol{x}))\right\|$$
$$= \left\|[\nabla_{\boldsymbol{y}} \boldsymbol{\lambda}_H(\boldsymbol{y}^{(t)}; \boldsymbol{x})]^\top \nabla_{\boldsymbol{\lambda}} F^\nu(\boldsymbol{\lambda}_H(\boldsymbol{y}^{(t)}; \boldsymbol{x})) - [\nabla_{\boldsymbol{y}} \boldsymbol{\lambda}(\boldsymbol{y}^{(t)}; \boldsymbol{x})]^\top \nabla_{\boldsymbol{\lambda}} F^\nu(\boldsymbol{\lambda}(\boldsymbol{y}^{(t)}; \boldsymbol{x}))\right\|$$
$$\le \left\|[\nabla_{\boldsymbol{y}} \boldsymbol{\lambda}_H(\boldsymbol{y}^{(t)}; \boldsymbol{x})]^\top \left(\nabla_{\boldsymbol{\lambda}} F^\nu(\boldsymbol{\lambda}_H(\boldsymbol{y}^{(t)}; \boldsymbol{x})) - \nabla_{\boldsymbol{\lambda}} F(\boldsymbol{\lambda}(\boldsymbol{y}^{(t)}; \boldsymbol{x}))\right)\right\|$$
$$+ \left\|\left([\nabla_{\boldsymbol{y}} \boldsymbol{\lambda}_H(\boldsymbol{y}^{(t)}; \boldsymbol{x})]^\top - [\nabla_{\boldsymbol{y}} \boldsymbol{\lambda}(\boldsymbol{y}^{(t)}; \boldsymbol{x})]^\top\right) \nabla_{\boldsymbol{\lambda}} F^\nu(\boldsymbol{\lambda}(\boldsymbol{y}^{(t)}; \boldsymbol{x}))\right\|. \tag{36}$$

For the first part in the above inequality, we have

$$\left\|\left[\nabla_{\boldsymbol{y}}\boldsymbol{\lambda}_H(\boldsymbol{y}^{(t)};\boldsymbol{x})\right]^{\top}\left(\nabla_{\boldsymbol{\lambda}}F^{\nu}(\boldsymbol{\lambda}_H(\boldsymbol{y}^{(t)};\boldsymbol{x}))-\nabla_{\boldsymbol{\lambda}}F^{\nu}(\boldsymbol{\lambda}(\boldsymbol{y}^{(t)};\boldsymbol{x}))\right)\right\|$$

$$=\left\|\sum_{h=0}^{H-1}\gamma^h\cdot\left(\frac{\partial F^{\nu}(\boldsymbol{\lambda}_H(\boldsymbol{y}^{(t)};\boldsymbol{x}))}{\partial\lambda_{s_h,b_h}}-\frac{\partial F^{\nu}(\boldsymbol{\lambda}(\boldsymbol{y}^{(t)};\boldsymbol{x}))}{\partial\lambda_{s_h,b_h}}\right)\cdot\left(\sum_{h'=0}^{h}\nabla_{\boldsymbol{y}}\log y^{(t)}(b_{h'}|s_{h'})\right)\right\|$$

$$\leq\sum_{h=0}^{\infty}\gamma^h\cdot\left\|\nabla_{\boldsymbol{\lambda}}F^{\nu}(\boldsymbol{\lambda}_H(\boldsymbol{y}^{(t)};\boldsymbol{x}))-\nabla_{\boldsymbol{\lambda}}F^{\nu}(\boldsymbol{\lambda}(\boldsymbol{y}^{(t)};\boldsymbol{x}))\right\|_{\infty}\cdot\left\|\left(\sum_{h'=0}^{\infty}\nabla_{\boldsymbol{y}}\log y^{(t)}(b_{h'}|s_{h'})\right)\right\|$$

$$\leq\sum_{h=0}^{\infty}\gamma^h\cdot\left\|\nu\boldsymbol{\lambda}_H(\boldsymbol{y}^{(t)};\boldsymbol{x})-\nu\boldsymbol{\lambda}(\boldsymbol{y}^{(t)};\boldsymbol{x})\right\|_{\infty}\cdot\left\|\left(\sum_{h'=0}^{\infty}\nabla_{\boldsymbol{y}}\log y^{(t)}(b_{h'}|s_{h'})\right)\right\|$$

$$\leq\sum_{h=0}^{\infty}\gamma^h\cdot\nu\|\boldsymbol{\lambda}_H(\boldsymbol{y}^{(t)};\boldsymbol{x})-\boldsymbol{\lambda}(\boldsymbol{y}^{(t)};\boldsymbol{x})\|_1\cdot(h+1)\cdot\frac{1}{\zeta} \tag{37}$$

$$\leq\frac{\nu}{(1-\gamma)^2\zeta}\cdot\|\boldsymbol{\lambda}_H(\boldsymbol{y}^{(t)};\boldsymbol{x})-\boldsymbol{\lambda}(\boldsymbol{y}^{(t)};\boldsymbol{x})\|_1$$

$$\leq\frac{\nu}{(1-\gamma)^2\zeta}\cdot\left(\sum_{s,b}\sum_{h=H}^{\infty}\gamma^h\cdot\mathbb{P}(s_h=s,b_h=b|\boldsymbol{y},s_0\sim\boldsymbol{\rho})\right)$$

$$\leq\frac{\nu}{(1-\gamma)^3\zeta}\cdot\gamma^H. \tag{38}$$

For the second part in (36), we have

$$\left\|\left(\left[\nabla_{\boldsymbol{y}}\boldsymbol{\lambda}_H(\boldsymbol{y}^{(t)};\boldsymbol{x})\right]^{\top}-\left[\nabla_{\boldsymbol{y}}\boldsymbol{\lambda}(\boldsymbol{y}^{(t)};\boldsymbol{x})\right]^{\top}\right)\nabla_{\boldsymbol{\lambda}}F^{\nu}(\boldsymbol{\lambda}(\boldsymbol{y}^{(t)};\boldsymbol{x}))\right\|$$

$$=\left\|\mathbb{E}\left[\sum_{h=H}^{\infty}\gamma^h\cdot\frac{\partial F^{\nu}(\boldsymbol{\lambda}(\boldsymbol{y}^{(t)};\boldsymbol{x}))}{\partial\lambda_{s_h,b_h}}\cdot\left(\sum_{h'=0}^{h}\nabla_{\boldsymbol{y}}\log y^{(t)}(b_{h'}|s_{h'})\right)\right]\right\|$$

$$\leq\sum_{h=H}^{\infty}\gamma^h\cdot\left(1+\frac{\nu}{1-\gamma}\right)\cdot(h+1)\cdot\frac{1}{\zeta}$$

$$\leq\left(1+\frac{\nu}{1-\gamma}\right)\cdot\left(\frac{H+1}{1-\gamma}+\frac{1}{(1-\gamma^2)}\right)\cdot\frac{1}{\zeta}\cdot\gamma^H$$

$$=\left(\frac{H+1}{(1-\gamma)\zeta}+\frac{\nu H+\nu+1}{(1-\gamma)^2\zeta}+\frac{\nu}{(1-\gamma)^3\zeta}\right)\cdot\gamma^H. \tag{39}$$

Combining (36), (38), and (39), we get the result. $\qquad\square$

Before we proceed to analyze the variance of the estimators, we first show the Lipschitz continuity of the gradient estimator.

**Lemma C.15.** Let $\tau=\{s_0,b_0,s_1,b_1,\cdots,s_{H-1},b_{H-1}\}$ be an arbitrary $H$-length trajectory. The gradient estimator satisfies

- For any policy $\boldsymbol{y}$, for any reward vectors $\boldsymbol{r}_1$ and $\boldsymbol{r}_2$,

$$\|\tilde{\boldsymbol{g}}(\tau|\boldsymbol{y};\boldsymbol{r}_1)-\tilde{\boldsymbol{g}}(\tau|\boldsymbol{y};\boldsymbol{r}_2)\|\leq\frac{1}{(1-\gamma)^2\zeta}\cdot\|\boldsymbol{r}_1-\boldsymbol{r}_2\|_{\infty}.$$

- For any policies $\boldsymbol{y}_1$ and $\boldsymbol{y}_2$, for any reward vectors $\boldsymbol{r}$,

$$\|\tilde{\boldsymbol{g}}(\tau|\boldsymbol{y}_1;\boldsymbol{r})-\tilde{\boldsymbol{g}}(\tau|\boldsymbol{y}_2;\boldsymbol{r})\|\leq\left(\frac{1}{(1-\gamma)^2\zeta^2}+\frac{\nu}{(1-\gamma)^3\,\zeta^2}\right)\cdot\|\boldsymbol{y}_1-\boldsymbol{y}_2\|.$$

**Proof.**

$$\|\tilde{\boldsymbol{g}}(\tau|\boldsymbol{y};\boldsymbol{r}_1) - \tilde{\boldsymbol{g}}(\tau|\boldsymbol{y};\boldsymbol{r}_2)\| = \left\|\sum_{h=0}^{H-1}\gamma^h \cdot (r_1(s_h,b_h) - r_2(s_h,b_h)) \cdot \left(\sum_{h'=0}^{h}\nabla_{\boldsymbol{y}}\log y(b_{h'}|s_{h'})\right)\right\|$$

$$\leq \sum_{h=0}^{H-1}\gamma^h \cdot \|\boldsymbol{r}_1 - \boldsymbol{r}_2\|_\infty \cdot (h+1)\cdot\frac{1}{\zeta} \tag{40}$$

$$\leq \frac{1}{(1-\gamma)^2\zeta}\cdot\|\boldsymbol{r}_1 - \boldsymbol{r}_2\|_\infty.$$

$$\|\tilde{\boldsymbol{g}}(\tau|\boldsymbol{y}_1,\boldsymbol{r}) - \tilde{\boldsymbol{g}}(\tau|\boldsymbol{y}_2,\boldsymbol{r})\|$$

$$\leq \left\|\sum_{h=0}^{H-1}\gamma^h \cdot r(s_h,b_h) \cdot \left(\sum_{h'=0}^{h}(\nabla_{\boldsymbol{y}}\log y_1(b_{h'}|s_{h'}) - \nabla_{\boldsymbol{y}}\log y_2(b_{h'}|s_{h'}))\right)\right\|$$

$$\leq \sum_{h=0}^{H-1}\gamma^h \cdot r(s_h,b_h) \cdot (h+1)\cdot\frac{1}{\zeta^2}\cdot\|\boldsymbol{y}_1 - \boldsymbol{y}_2\| \tag{41}$$

$$\leq \frac{(1+\frac{\nu}{1-\gamma})}{(1-\gamma)^2\zeta^2}\cdot\|\boldsymbol{y}_1 - \boldsymbol{y}_2\|$$

$$= \left(\frac{1}{(1-\gamma)^2\zeta^2} + \frac{\nu}{(1-\gamma)^3\zeta^2}\right)\cdot\|\boldsymbol{y}_1 - \boldsymbol{y}_2\|.$$

Where

- (40) follows from (22);
- (41) is because of (23).

$\square$

Now we analyze the variance of the estimators in the algorithm, we start with showing the following lemma.

**Lemma C.16** (Bounded Var. of Visit. Estimator)**.** For $\hat{\boldsymbol{\lambda}}^{(t)}$ in Algorithm 2, the variance is bounded. It holds that

$$\mathbb{E}\left[\|\hat{\boldsymbol{\lambda}}^{(t)} - \boldsymbol{\lambda}_H(\boldsymbol{y}^{(t)};\boldsymbol{x})\|^2\right] \leq \frac{1}{K(1-\gamma)^2}.$$

Where $\boldsymbol{\lambda}_H(\boldsymbol{y}^{(t)};\boldsymbol{x})$ is the truncated state-action visitation measure for $\boldsymbol{\lambda}(\boldsymbol{y}^{(t)};\boldsymbol{x})$ defined in (35)

**Proof.** It holds that

$$\mathbb{E}\left[\left\|\hat{\boldsymbol{\lambda}}^{(t)} - \boldsymbol{\lambda}_H(\boldsymbol{y}^{(t)};\boldsymbol{x})\right\|^2\right] = \mathbb{E}\left[\left\|\frac{1}{K}\sum_{\tau\in\mathcal{K}_i}\tilde{\boldsymbol{\lambda}}(\tau|\boldsymbol{y}^{(t)}) - \boldsymbol{\lambda}_H(\boldsymbol{y}^{(t)};\boldsymbol{x})\right\|^2\right]$$

$$= \frac{1}{K}\cdot\mathbb{E}\left[\left\|\tilde{\boldsymbol{\lambda}}(\tau|\boldsymbol{y}^{(t)}) - \boldsymbol{\lambda}_H(\boldsymbol{y}^{(t)};\boldsymbol{x})\right\|^2\right] \tag{42}$$

$$\leq \frac{1}{K}\cdot\mathbb{E}\left[\left\|\tilde{\boldsymbol{\lambda}}(\tau|\boldsymbol{y}^{(t)})\right\|^2\right] \tag{43}$$

$$\leq \frac{1}{K(1-\gamma)^2}. \tag{44}$$

Where:

- (42) is due to $\mathbb{E}\left[\hat{\boldsymbol{\lambda}}^{(t)}\right] = \boldsymbol{\lambda}_H(\boldsymbol{y}^{(t)};\boldsymbol{x})$ and the fact that trajectories $\tau \in \mathcal{K}^{(t)}$ are independently sampled;

- (43) is because the variance is bounded by the second moment;
- (44) is because $\|\tilde{\boldsymbol{\lambda}}(\tau|\boldsymbol{y}^{(t)})\| \leq \frac{1}{1-\gamma}$.

$\square$

Now we analyze the variance of gradient estimator $\hat{\boldsymbol{g}}_{\boldsymbol{y}}^{(t)}$ by providing the following lemma:

**Lemma C.17** (Bounded Var. of Grad. Estimator). For $\hat{\boldsymbol{g}}_{\boldsymbol{y}}^{(t)}$ in Algorithm 2, we have

$$\mathbb{E}\left[\left\|\hat{\boldsymbol{g}}_{\boldsymbol{y}}^{(t)} - \left[\nabla_{\boldsymbol{y}}\boldsymbol{\lambda}_H(\boldsymbol{y}^{(t)};\boldsymbol{x})\right]^\top \boldsymbol{r}^{(t)}\right\|^2\right] \leq \frac{3}{K(1-\gamma)^4\zeta^2} + \frac{6\nu}{K(1-\gamma)^5\zeta^2} + \frac{9\nu^2}{K(1-\gamma)^6\zeta^2}.$$

**Proof.** We denote $\boldsymbol{r}^\star = \nabla_{\boldsymbol{\lambda}}F^\nu\left(\boldsymbol{\lambda}_H(\boldsymbol{y}^{(t)};\boldsymbol{x})\right) = \boldsymbol{r}(\boldsymbol{x}) - \nu\boldsymbol{\lambda}_H(\boldsymbol{y}^{(t)};\boldsymbol{x})$. Then we have

$$\mathbb{E}\left[\left\|\hat{\boldsymbol{g}}_{\boldsymbol{y}}^{(t)} - \left[\nabla_{\boldsymbol{y}}\boldsymbol{\lambda}_H(\boldsymbol{y}^{(t)};\boldsymbol{x})\right]^\top \boldsymbol{r}^{(t)}\right\|^2\right]$$

$$=\mathbb{E}\left[\left\|\frac{1}{K}\sum_{\tau\in\mathcal{K}^{(t)}}\tilde{\boldsymbol{g}}(\tau|\boldsymbol{y}^{(t)};\boldsymbol{r}^{(t)}) - \frac{1}{K}\sum_{\tau\in\mathcal{K}^{(t)}}\tilde{\boldsymbol{g}}(\tau|\boldsymbol{y}^{(t)};\boldsymbol{r}^\star) + \frac{1}{K}\sum_{\tau\in\mathcal{K}^{(t)}}\tilde{\boldsymbol{g}}(\tau|\boldsymbol{y}^{(t)};\boldsymbol{r}^\star)\right.\right.$$
$$\left.\left. - \left[\nabla_{\boldsymbol{y}}\boldsymbol{\lambda}_H(\boldsymbol{y}^{(t)};\boldsymbol{x})\right]^\top \boldsymbol{r}^\star + \left[\nabla_{\boldsymbol{y}}\boldsymbol{\lambda}_H(\boldsymbol{y}^{(t)};\boldsymbol{x})\right]^\top \boldsymbol{r}^\star - \left[\nabla_{\boldsymbol{y}}\boldsymbol{\lambda}_H(\boldsymbol{y}^{(t)};\boldsymbol{x})\right]^\top \boldsymbol{r}^{(t)}\right\|^2\right]$$

$$\leq 3\mathbb{E}\left[\left\|\frac{1}{K}\sum_{\tau\in\mathcal{K}^{(t)}}\left(\tilde{\boldsymbol{g}}(\tau|\boldsymbol{y}^{(t)};\boldsymbol{r}^{(t)}) - \tilde{\boldsymbol{g}}(\tau|\boldsymbol{y}^{(t)};\boldsymbol{r}^\star)\right)\right\|^2\right]$$

$$+ 3\mathbb{E}\left[\left\|\frac{1}{K}\sum_{\tau\in\mathcal{K}^{(t)}}\tilde{\boldsymbol{g}}(\tau|\boldsymbol{y}^{(t)};\boldsymbol{r}^\star) - \left[\nabla_{\boldsymbol{y}}\boldsymbol{\lambda}_H(\boldsymbol{y}^{(t)};\boldsymbol{x})\right]^\top \boldsymbol{r}^\star\right\|^2\right]$$

$$+ 3\mathbb{E}\left[\left\|\left[\nabla_{\boldsymbol{y}}\boldsymbol{\lambda}_H(\boldsymbol{y}^{(t)};\boldsymbol{x})\right]^\top \boldsymbol{r}^\star - \left[\nabla_{\boldsymbol{y}}\boldsymbol{\lambda}_H(\boldsymbol{y}^{(t)};\boldsymbol{x})\right]^\top \boldsymbol{r}^{(t)}\right\|^2\right]. \tag{45}$$

Where (45) is due to Cauchy-Schwarz inequality. For the first part in (45), we have

$$\mathbb{E}\left[\left\|\frac{1}{K}\sum_{\tau\in\mathcal{K}^{(t)}}\left(\tilde{\boldsymbol{g}}(\tau|\boldsymbol{y}^{(t)};\boldsymbol{r}^{(t)}) - \tilde{\boldsymbol{g}}(\tau|\boldsymbol{y}^{(t)};\boldsymbol{r}^\star)\right)\right\|^2\right]$$

$$\leq\frac{1}{K}\sum_{\tau\in\mathcal{K}^{(t)}}\mathbb{E}\left[\left\|\tilde{\boldsymbol{g}}(\tau|\boldsymbol{y}^{(t)};\boldsymbol{r}^{(t)}) - \tilde{\boldsymbol{g}}(\tau|\boldsymbol{y}^{(t)};\boldsymbol{r}^\star)\right\|^2\right] \tag{46}$$

$$\leq\frac{1}{(1-\gamma)^4\zeta^2}\cdot\mathbb{E}\left[\left\|\boldsymbol{r}^{(t)} - \boldsymbol{r}^\star\right\|_\infty^2\right] \tag{47}$$

$$\leq\frac{\nu^2}{(1-\gamma)^4\zeta^2}\cdot E\left[\left\|\hat{\boldsymbol{\lambda}}^{(t)} - \boldsymbol{\lambda}_H(\boldsymbol{y}^{(t)};\boldsymbol{x})\right\|\right] \tag{48}$$

$$\leq\frac{\nu^2}{K(1-\gamma)^6\zeta^2}. \tag{49}$$

Where:

- (46) is due to Cauchy-Schwarz inequality;
- (47) follows from Lemma C.15;
- (48) follows the same proof as in (37);

- (49) is because of Lemma C.16.

For the second part in (45), it holds that,

$$\mathbb{E}\left[\left\|\frac{1}{K}\sum_{\tau\in\mathcal{K}^{(t)}}\tilde{\boldsymbol{g}}(\tau|\boldsymbol{y}^{(t)};\boldsymbol{r}^{\star}) - \left[\nabla_{\boldsymbol{y}}\boldsymbol{\lambda}_H(\boldsymbol{y}^{(t)};\boldsymbol{x})\right]^{\top}\boldsymbol{r}^{\star}\right\|^2\right]$$

$$=\frac{1}{K}\mathbb{E}\left[\left\|\tilde{\boldsymbol{g}}(\tau|\boldsymbol{y}^{(t)};\boldsymbol{r}^{\star}) - \left[\nabla_{\boldsymbol{y}}\boldsymbol{\lambda}_H(\boldsymbol{y}^{(t)};\boldsymbol{x})\right]^{\top}\boldsymbol{r}^{\star}\right\|^2\right] \tag{50}$$

$$\leq\frac{1}{K}\mathbb{E}\left[\left\|\tilde{\boldsymbol{g}}(\tau|\boldsymbol{y}^{(t)};\boldsymbol{r}^{\star})\right\|^2\right] \tag{51}$$

$$=\frac{1}{K}\mathbb{E}\left[\left\|\sum_{h=0}^{H-1}\gamma^h\cdot r^{\star}(s_h,b_h)\cdot\left(\sum_{h'=0}^{h}\nabla_{\boldsymbol{y}}\log y^{(t)}(b_{h'}|s_{h'})\right)\right\|^2\right]$$

$$\leq\frac{1}{K}\left(\sum_{h=0}^{H-1}\gamma^h\cdot\left(1+\frac{\nu}{1-\gamma}\right)\cdot\frac{1}{\zeta}\cdot(h+1)\right)^2 \tag{52}$$

$$\leq\frac{1}{K(1-\gamma)^4\zeta^2} + \frac{2\nu}{K(1-\gamma)^5\zeta^2} + \frac{\nu^2}{K(1-\gamma)^6\zeta^2}. \tag{53}$$

Where

- (50) is due to Lemma C.14 and the fact that trajectories $\tau$ are independently sampled;
- (51) is because variance is bounded by second moment;
- (52) follows from (22).

Finally for the last part in (45), we have

$$\mathbb{E}\left[\left\|\left[\nabla_{\boldsymbol{y}}\boldsymbol{\lambda}_H(\boldsymbol{y}^{(t)};\boldsymbol{x})\right]^{\top}\boldsymbol{r}^{\star} - \left[\nabla_{\boldsymbol{y}}\boldsymbol{\lambda}_H(\boldsymbol{y}^{(t)};\boldsymbol{x})\right]^{\top}\boldsymbol{r}^{(t)}\right\|^2\right]$$

$$=\mathbb{E}\left[\left\|\left[\nabla_{\boldsymbol{y}}\boldsymbol{\lambda}_H(\boldsymbol{y}^{(t)};\boldsymbol{x})\right]^{\top}(\boldsymbol{r}^{\star}-\boldsymbol{r}^{(t)})\right\|^2\right]$$

$$\leq\mathbb{E}\left[\left\|\sum_{h=0}^{H-1}\gamma^h\cdot\|\boldsymbol{r}^{\star}-\boldsymbol{r}^{(t)}\|_{\infty}\cdot\left(\sum_{h'=0}^{h}\nabla_{\boldsymbol{y}}\log y^{(t)}(b_{h'}|s_{h'})\right)\right\|^2\right]$$

$$\leq\left(\sum_{h=0}^{H-1}\gamma^h\cdot\nu\cdot(h+1)\cdot\frac{1}{\zeta}\right)^2\cdot\mathbb{E}\left[\left\|\hat{\boldsymbol{\lambda}}^{(t)}-\boldsymbol{\lambda}_H(\boldsymbol{y}^{(t)};\boldsymbol{x})\right\|^2\right] \tag{54}$$

$$\leq\frac{\nu^2}{(1-\gamma)^4\zeta^2}\cdot\frac{1}{K(1-\gamma)^2} \tag{55}$$

$$=\frac{\nu^2}{K(1-\gamma)^6\zeta^2}. \tag{56}$$

Where

- (54) is due to (22) and (37);
- (55) is because of Lemma C.16.

Combine (45), (49), (53) and (56), we get

$$\mathbb{E}\left[\left\|\hat{\boldsymbol{g}}_{\boldsymbol{y}}^{(t)} - \left[\nabla_{\boldsymbol{y}}\boldsymbol{\lambda}_H(\boldsymbol{y}^{(t)};\boldsymbol{x})\right]^\top \boldsymbol{r}^{(t)}\right\|^2\right]$$

$$\leq \frac{3\nu^2}{K(1-\gamma)^6\zeta^2} + 3\left(\frac{1}{K(1-\gamma)^4\zeta^2} + \frac{2\nu}{K(1-\gamma)^5\zeta^2} + \frac{\nu^2}{K(1-\gamma)^6\zeta^2}\right) + \frac{3\nu^2}{K(1-\gamma)^6\zeta^2}$$

$$= \frac{3}{K(1-\gamma)^4\zeta^2} + \frac{6\nu}{K(1-\gamma)^5\zeta^2} + \frac{9\nu^2}{K(1-\gamma)^6\zeta^2}.$$

$\square$

After bounding the variance of $\hat{\boldsymbol{g}}_{\boldsymbol{y}}^{(t)}$ in Algorithm 2, we can prove the following lemma

**Lemma C.18** (Bounded Dist. with Actual Grad.). Consider $\boldsymbol{y}^{(t)}$ and $\hat{\boldsymbol{g}}_{\boldsymbol{y}}^{(t)}$ in Algorithm 2, it holds that

$$\mathbb{E}\left[\left\|\hat{\boldsymbol{g}}_{\boldsymbol{y}}^{(t)} - \nabla_{\boldsymbol{y}}F^\nu(\boldsymbol{\lambda}(\boldsymbol{y}^{(t)};\boldsymbol{x}))\right\|^2\right] \leq \frac{\mathcal{C}_1}{K} + \mathcal{C}_2 \cdot \gamma^{2H}.$$

Where

$$\mathcal{C}_1 = \frac{57}{(1-\gamma)^6\zeta^2}, \quad \mathcal{C}_2 = \frac{126H^2}{(1-\gamma)^6\zeta^2}.$$

**Proof.** Let $\boldsymbol{r}^\star = \nabla_{\boldsymbol{\lambda}}F^\nu\left(\boldsymbol{\lambda}_H(\boldsymbol{y}^{(t)};\boldsymbol{x})\right) = \boldsymbol{r}(\boldsymbol{x}) - \nu\boldsymbol{\lambda}_H(\boldsymbol{y}^{(t)};\boldsymbol{x}).$

$$\mathbb{E}\left[\left\|\hat{\boldsymbol{g}}_{\boldsymbol{y}}^{(t)} - \nabla_{\boldsymbol{y}}F^\nu(\boldsymbol{\lambda}(\boldsymbol{y}^{(t)};\boldsymbol{x}))\right\|^2\right]$$

$$= \mathbb{E}\left[\left\|\hat{\boldsymbol{g}}_{\boldsymbol{y}}^{(t)} - \left[\nabla_{\boldsymbol{y}}\boldsymbol{\lambda}_H(\boldsymbol{y}^{(t)};\boldsymbol{x})\right]^\top \boldsymbol{r}^{(t)} + \left[\nabla_{\boldsymbol{y}}\boldsymbol{\lambda}_H(\boldsymbol{y}^{(t)};\boldsymbol{x})\right]^\top \boldsymbol{r}^{(t)} - \left[\nabla_{\boldsymbol{y}}\boldsymbol{\lambda}_H(\boldsymbol{y}^{(t)};\boldsymbol{x})\right]^\top \boldsymbol{r}^\star \right.$$

$$\left. + \left[\nabla_{\boldsymbol{y}}\boldsymbol{\lambda}_H(\boldsymbol{y}^{(t)};\boldsymbol{x})\right]^\top \boldsymbol{r}^\star - \nabla_{\boldsymbol{y}}F^\nu(\boldsymbol{\lambda}(\boldsymbol{y}^{(t)};\boldsymbol{x}))\right\|^2\right]$$

$$\leq 3\mathbb{E}\left[\left\|\hat{\boldsymbol{g}}_{\boldsymbol{y}}^{(t)} - \left[\nabla_{\boldsymbol{y}}\boldsymbol{\lambda}_H(\boldsymbol{y}^{(t)};\boldsymbol{x})\right]^\top \boldsymbol{r}^{(t)}\right\|^2\right]$$

$$+ 3\mathbb{E}\left[\left\|\left[\nabla_{\boldsymbol{y}}\boldsymbol{\lambda}_H(\boldsymbol{y}^{(t)};\boldsymbol{x})\right]^\top \boldsymbol{r}^{(t)} - \left[\nabla_{\boldsymbol{y}}\boldsymbol{\lambda}_H(\boldsymbol{y}^{(t)};\boldsymbol{x})\right]^\top \boldsymbol{r}^\star\right\|^2\right]$$

$$+ 3\mathbb{E}\left[\left\|\left[\nabla_{\boldsymbol{y}}\boldsymbol{\lambda}_H(\boldsymbol{y}^{(t)};\boldsymbol{x})\right]^\top \boldsymbol{r}^\star - \nabla_{\boldsymbol{y}}F^\nu(\boldsymbol{\lambda}(\boldsymbol{y}^{(t)};\boldsymbol{x}))\right\|^2\right]. \tag{57}$$

Notice that the first part in (57) is bounded in Lemma C.17 and the second part is bounded in (56). For the last part, observe that

$$\left\| \left[ \nabla_{\boldsymbol{y}} \boldsymbol{\lambda}_H(\boldsymbol{y}^{(t)}; \boldsymbol{x}) \right]^\top \boldsymbol{r}^\star - \nabla_{\boldsymbol{y}} F^\nu(\boldsymbol{\lambda}(\boldsymbol{y}^{(t)}; \boldsymbol{x})) \right\|^2$$

$$= \left\| \left[ \nabla_{\boldsymbol{y}} \boldsymbol{\lambda}_H(\boldsymbol{y}^{(t)}; \boldsymbol{x}) \right]^\top \nabla_{\boldsymbol{\lambda}} F^\nu(\boldsymbol{\lambda}_H(\boldsymbol{y}^{(t)}; \boldsymbol{x})) - \left[ \nabla_{\boldsymbol{y}} \boldsymbol{\lambda}(\boldsymbol{y}^{(t)}; \boldsymbol{x}) \right]^\top \nabla_{\boldsymbol{\lambda}} F^\nu(\boldsymbol{\lambda}(\boldsymbol{y}^{(t)}; \boldsymbol{x})) \right\|^2$$

$$= \left\| \left[ \nabla_{\boldsymbol{y}} \boldsymbol{\lambda}_H(\boldsymbol{y}^{(t)}; \boldsymbol{x}) \right]^\top \left( \nabla_{\boldsymbol{\lambda}} F^\nu(\boldsymbol{\lambda}_H(\boldsymbol{y}^{(t)}; \boldsymbol{x})) - \nabla_{\boldsymbol{\lambda}} F^\nu(\boldsymbol{\lambda}(\boldsymbol{y}^{(t)}; \boldsymbol{x})) \right) \right.$$

$$\left. + \left( \left[ \nabla_{\boldsymbol{y}} \boldsymbol{\lambda}_H(\boldsymbol{y}^{(t)}; \boldsymbol{x}) \right]^\top - \left[ \nabla_{\boldsymbol{y}} \boldsymbol{\lambda}(\boldsymbol{y}^{(t)}; \boldsymbol{x}) \right]^\top \right) \nabla_{\boldsymbol{\lambda}} F^\nu(\boldsymbol{\lambda}(\boldsymbol{y}^{(t)}; \boldsymbol{x})) \right\|^2$$

$$\leq 2 \left\| \left[ \nabla_{\boldsymbol{y}} \boldsymbol{\lambda}_H(\boldsymbol{y}^{(t)}; \boldsymbol{x}) \right]^\top \left( \nabla_{\boldsymbol{\lambda}} F^\nu(\boldsymbol{\lambda}_H(\boldsymbol{y}^{(t)}; \boldsymbol{x})) - \nabla_{\boldsymbol{\lambda}} F^\nu(\boldsymbol{\lambda}(\boldsymbol{y}^{(t)}; \boldsymbol{x})) \right) \right\|^2$$

$$+ 2 \left\| \left( \left[ \nabla_{\boldsymbol{y}} \boldsymbol{\lambda}_H(\boldsymbol{y}^{(t)}; \boldsymbol{x}) \right]^\top - \left[ \nabla_{\boldsymbol{y}} \boldsymbol{\lambda}(\boldsymbol{y}^{(t)}; \boldsymbol{x}) \right]^\top \right) \nabla_{\boldsymbol{\lambda}} F^\nu(\boldsymbol{\lambda}(\boldsymbol{y}^{(t)}; \boldsymbol{x})) \right\|^2. \tag{58}$$

Where (58) is follows from Cauchy-Schwarz inequality. For the first part, we have

$$\left\| \left[ \nabla_{\boldsymbol{y}} \boldsymbol{\lambda}_H(\boldsymbol{y}^{(t)}; \boldsymbol{x}) \right]^\top \left( \nabla_{\boldsymbol{\lambda}} F^\nu(\boldsymbol{\lambda}_H(\boldsymbol{y}^{(t)}; \boldsymbol{x})) - \nabla_{\boldsymbol{\lambda}} F^\nu(\boldsymbol{\lambda}(\boldsymbol{y}^{(t)}; \boldsymbol{x})) \right) \right\|^2$$

$$\leq \left( \sum_{h=0}^\infty \gamma^h \cdot \nu \| \boldsymbol{\lambda}_H(\boldsymbol{y}^{(t)}; \boldsymbol{x}) - \boldsymbol{\lambda}(\boldsymbol{y}^{(t)}; \boldsymbol{x}) \|_1 \cdot (h+1) \cdot \frac{1}{\zeta} \right)^2 \tag{59}$$

$$\leq \frac{\nu^2}{(1-\gamma)^4 \zeta^2} \cdot \| \boldsymbol{\lambda}_H(\boldsymbol{y}^{(t)}; \boldsymbol{x}) - \boldsymbol{\lambda}(\boldsymbol{y}^{(t)}; \boldsymbol{x}) \|_1^2.$$

Where (59) is because of (37). Since

$$\left\| \boldsymbol{\lambda}_H(\boldsymbol{y}^{(t)}; \boldsymbol{x}) - \boldsymbol{\lambda}(\boldsymbol{y}^{(t)}; \boldsymbol{x}) \right\|_1^2 = \left( \sum_{h=H}^\infty \sum_{s,b} \gamma^t \, \mathbb{P}(s_h = s, b_h = b | \boldsymbol{y}^{(t)}, s_0 \sim \boldsymbol{\rho}) \right)^2$$

$$= \left( \gamma^H \sum_{h=0}^\infty \gamma^h \cdot 1 \right)^2$$

$$\leq \frac{\gamma^{2H}}{(1-\gamma)^2}.$$

We have

$$\left\| \left[ \nabla_{\boldsymbol{y}} \boldsymbol{\lambda}_H(\boldsymbol{y}^{(t)}; \boldsymbol{x}) \right]^\top \left( \nabla_{\boldsymbol{\lambda}} F^\nu(\boldsymbol{\lambda}_H(\boldsymbol{y}^{(t)}; \boldsymbol{x})) - \nabla_{\boldsymbol{\lambda}} F^\nu(\boldsymbol{\lambda}(\boldsymbol{y}^{(t)}; \boldsymbol{x})) \right) \right\|^2 \leq \frac{\nu^2}{(1-\gamma)^6 \zeta^2} \cdot \gamma^{2H}. \tag{60}$$

For the second part in (58), it holds that

$$\left\| \left( \left[ \nabla_{\boldsymbol{y}} \boldsymbol{\lambda}_H(\boldsymbol{y}^{(t)}; \boldsymbol{x}) \right]^\top - \left[ \nabla_{\boldsymbol{y}} \boldsymbol{\lambda}(\boldsymbol{y}^{(t)}; \boldsymbol{x}) \right]^\top \right) \nabla_{\boldsymbol{\lambda}} F^\nu(\boldsymbol{\lambda}(\boldsymbol{y}^{(t)}; \boldsymbol{x})) \right\|^2$$

$$= \left\| \mathbb{E} \left[ \sum_{h=H}^\infty \gamma^h \cdot \nabla_{\boldsymbol{\lambda}} F^\nu(\boldsymbol{\lambda}(\boldsymbol{y}^{(t)}; \boldsymbol{x}))_{s_h, b_h} \cdot \left( \sum_{h'=0}^h \nabla_{\boldsymbol{y}} \log y^{(t)}(b_{h'} | s_{h'}) \right) \right] \right\|^2$$

$$\leq \left( \sum_{h=H}^\infty \gamma^h \cdot \left( 1 + \frac{\nu}{1-\gamma} \right) \cdot (h+1) \cdot \frac{1}{\zeta} \right)^2$$

$$\leq \left( 1 + \frac{\nu}{1-\gamma} \right)^2 \cdot \frac{1}{\zeta^2} \cdot \left( \frac{(H+1)^2}{(1-\gamma)^2} + \frac{1}{(1-\gamma)^4} \right) \cdot \gamma^{2H}$$

$$= \left( \frac{(H+1)^2}{(1-\gamma)^2 \zeta^2} + \frac{2\nu(H+1)^2}{(1-\gamma)^3 \zeta^2} + \frac{\nu^2(H+1)^2 + 1}{(1-\gamma)^4 \zeta^2} + \frac{2\nu}{(1-\gamma)^5 \zeta^2} + \frac{\nu^2}{(1-\gamma)^6} \right) \cdot \gamma^{2H}. \tag{61}$$

Combine (58), (60), and (61) we get

$$
\left\| \left[ \nabla_{\boldsymbol{y}} \boldsymbol{\lambda}_H(\boldsymbol{y}^{(t)}; \boldsymbol{x}) \right]^{\top} \boldsymbol{r}^{\star} - \nabla_{\boldsymbol{y}} F^{\nu}(\boldsymbol{\lambda}(\boldsymbol{y}^{(t)}; \boldsymbol{x})) \right\|^2
$$
$$
\leq 2 \left( \frac{(H+1)^2}{(1-\gamma)^2 \zeta^2} + \frac{2\nu(H+1)^2}{(1-\gamma)^3 \zeta^2} + \frac{\nu^2(H+1)^2+1}{(1-\gamma)^4 \zeta^2} + \frac{2\nu}{(1-\gamma)^5 \zeta^2} + \frac{2\nu^2}{(1-\gamma)^6 \zeta^2} \right) \cdot \gamma^{2H}. \quad (62)
$$

Now combine Lemma C.17, (56), (57), and (62), we get

$$
\mathbb{E}\left[ \left\| \hat{\boldsymbol{g}}_{\boldsymbol{y}}^{(t)} - \nabla_{\boldsymbol{y}} F^{\nu}(\boldsymbol{\lambda}(\boldsymbol{y}^{(t)}; \boldsymbol{x})) \right\|^2 \right] \leq \frac{\mathcal{C}_1}{K} + \mathcal{C}_2 \cdot \gamma^{2H}.
$$

$\square$

# D  Nonconvex–Hidden-Strongly-Concave Optimization

In this section we generalize our results to the more general setting of any constrained min-max optimization problem of the form $\min_{\boldsymbol{x}\in\mathcal{X}}\max_{\boldsymbol{y}\in\mathcal{Y}}$ when $f$ is nonconvex–hidden-strongly-concave. In particular:

- In Theorem D.2 we prove the differentiability and Hölder continuity of the max function $\Phi(\boldsymbol{x}) = \max_{\boldsymbol{y}\in\mathcal{Y}} f(\boldsymbol{x},\boldsymbol{y})$ by utilizing the Hölder continuity of the maximizers w.r.t. to $\boldsymbol{x}$ (Theorem D.1).
- Finally, in Theorem D.3 we prove that Algorithm 3 (SGDMAX) [70, Algorithm 4] with an appropriate tuning converges to an $\epsilon$-SP for nonconvex–hidden-concave functions.

We begin by stating the assumptions we make.

**Assumption D.1.** Let $f$ be a function defined on $\mathcal{X}\times\mathcal{Y}$ where $\mathcal{X}$ and $\mathcal{Y}$ are compact convex sets. $L$-Lipschitz continuous and $\ell$-smooth.

**Assumption D.2.** Let $c$ be a "1–1" mapping between $\mathcal{Y}$ and a compact convex set $\mathcal{U}$ parameterized by $\boldsymbol{x}\in\mathcal{X}$. Further, we assume that $c$ and its inverse $c^{-1}$ are $L_c$- and $L_{c^{-1}}$-Lipschitz continuous.

**Assumption D.3.** Let $H$ be a nonconvex–strongly-concave reformulation of $f$ (as in Assumption D.1) for a mapping $c$ (as in Assumption D.2). We assume $H$ to be $L_H$-Lipschitz continuous and $\ell_H$-smooth.

Moving on, we can show that the maximizers $\boldsymbol{u}^\star(\cdot)$ are Hölder continuous w.r.t. to $\boldsymbol{x}$.

**Theorem D.1** (Continuity of the maximizers). *Let a function nonconvex–nonconcave function $f, c, H$ as in Assumptions D.1 to D.3. We define $\boldsymbol{u}^\star(\boldsymbol{x}) \coloneqq \operatorname{argmax}_{\boldsymbol{u}\in\mathcal{U}(\boldsymbol{x})} H(\boldsymbol{x},\boldsymbol{u})$, then it is the case that*

$$\|\boldsymbol{u}^\star(\boldsymbol{x}_1) - \boldsymbol{u}^\star(\boldsymbol{x}_2)\| \leq L_\star \|\boldsymbol{x}_1 - \boldsymbol{x}_2\|^{1/2}.$$

*Where $L_\star = \frac{1}{2\nu}\left(2\ell_H\sqrt{\operatorname{Diam}_{\mathcal{X}}} + 2\sqrt{\nu(1+2\ell_H)L_c\operatorname{Diam}_{\mathcal{U}} + 2\nu L_c L_H}\right)$.*

**Proof.** Consider any $\boldsymbol{x}_1, \boldsymbol{x}_2 \in \mathcal{X}$, since $\boldsymbol{u}^\star$ is the maximizer, it holds that

$$\nabla H\left(\boldsymbol{x}_1, \boldsymbol{u}^\star(\boldsymbol{x}_1)\right)^\top \left(\boldsymbol{u}_1 - \boldsymbol{u}^\star(\boldsymbol{x}_1)\right) \leq 0, \quad \forall \boldsymbol{u}_1 \in \mathcal{U}(\boldsymbol{x}_1);$$

$$\nabla H\left(\boldsymbol{x}_2, \boldsymbol{u}^\star(\boldsymbol{x}_2)\right)^\top \left(\boldsymbol{u}_2 - \boldsymbol{u}^\star(\boldsymbol{x}_2)\right) \leq 0, \quad \forall \boldsymbol{u}_2 \in \mathcal{U}(\boldsymbol{x}_2).$$

We now consider $\overline{\boldsymbol{u}}$ that belong to the set $\overline{\mathcal{U}} = \mathcal{U}(\boldsymbol{x}_1) \cup \mathcal{U}(\boldsymbol{x}_2)$. We observe that due to the Lipschitz mapping, for every $\overline{\boldsymbol{u}}\in\overline{\mathcal{U}}$, there exist a $\boldsymbol{u}_1 \in \mathcal{U}(\boldsymbol{x}_1)$ such that $\|\overline{\boldsymbol{u}} - \boldsymbol{u}_1\| \leq L_c\|\boldsymbol{x}_1 - \boldsymbol{x}_2\|$. Similar argument holds for $\boldsymbol{u}_2 \in \mathcal{U}(\boldsymbol{x}_2)$. Therefore, from previous two inequalities, we have

$$\nabla H\left(\boldsymbol{x}_1, \boldsymbol{u}^\star(\boldsymbol{x}_1)\right)^\top \left(\overline{\boldsymbol{u}} - \boldsymbol{u}^\star(\boldsymbol{x}_1)\right) \leq L_c L_H \|\boldsymbol{x}_1 - \boldsymbol{x}_2\| \quad \forall \overline{\boldsymbol{u}} \in \overline{\mathcal{U}};$$

$$\nabla H\left(\boldsymbol{x}_2, \boldsymbol{u}^\star(\boldsymbol{x}_2)\right)^\top \left(\overline{\boldsymbol{u}} - \boldsymbol{u}^\star(\boldsymbol{x}_2)\right) \leq L_c L_H \|\boldsymbol{x}_1 - \boldsymbol{x}_2\| \quad \forall \overline{\boldsymbol{u}} \in \overline{\mathcal{U}}.$$

Where in the above inequalities we used the fact that $\nabla H(\boldsymbol{x}, \boldsymbol{u}) \leq L_H$. We plug in $\overline{\boldsymbol{u}} \leftarrow \boldsymbol{u}^\star(\boldsymbol{x}_2)$ and $\overline{\boldsymbol{u}} \leftarrow \boldsymbol{u}^\star(\boldsymbol{x}_1)$ accordingly,

$$\nabla H\left(\boldsymbol{x}_1, \boldsymbol{u}^\star(\boldsymbol{x}_1)\right)^\top \left(\boldsymbol{u}^\star(\boldsymbol{x}_1) - \boldsymbol{u}^\star(\boldsymbol{x}_1)\right) \leq L_c L_H \|\boldsymbol{x}_1 - \boldsymbol{x}_2\|,$$

$$\nabla H\left(\boldsymbol{x}_2, \boldsymbol{u}^\star(\boldsymbol{x}_2)\right)^\top \left(\boldsymbol{u}^\star(\boldsymbol{x}_1) - \boldsymbol{u}^\star(\boldsymbol{x}_2)\right) \leq L_c L_H \|\boldsymbol{x}_1 - \boldsymbol{x}_2\|.$$

Adding the two inequalities results in,

$$\left(\nabla H\left(\boldsymbol{x}_1, \boldsymbol{u}^\star(\boldsymbol{x}_1)\right) - \nabla H\left(\boldsymbol{x}_2, \boldsymbol{u}^\star(\boldsymbol{x}_2)\right)\right)^\top \left(\boldsymbol{u}^\star(\boldsymbol{x}_1) - \boldsymbol{u}^\star(\boldsymbol{x}_1)\right) \leq 2L_c L_H \|\boldsymbol{x}_1 - \boldsymbol{x}_2\|. \quad (63)$$

Since $H(\boldsymbol{x}, \cdot)$ is $\nu$-strongly concave in $\boldsymbol{u}$ for all $\boldsymbol{x}$, it holds that

$$(\boldsymbol{u}_1 - \boldsymbol{u}^\star(\boldsymbol{x}_1))^\top \left(\nabla H\left(\boldsymbol{x}_1, \boldsymbol{u}_1\right) - \nabla H\left(\boldsymbol{x}_1, \boldsymbol{u}^\star(\boldsymbol{x}_1)\right)\right) + \nu\|\boldsymbol{u}_1 - \boldsymbol{u}^\star(\boldsymbol{x}_1)\|^2 \leq 0 \quad \forall \boldsymbol{u}_1 \in \mathcal{U}(\boldsymbol{x}_1).$$

We again consider feasibility set $\overline{\boldsymbol{u}} \in \overline{\mathcal{U}}$. Since for every $\overline{\boldsymbol{u}} \in \overline{\mathcal{U}}$, there exists $\boldsymbol{u}_1 \in \mathcal{U}(\boldsymbol{x}_1)$ s.t. $\|\overline{\boldsymbol{u}} - \boldsymbol{u}_2\| \leq L_c\|\boldsymbol{x}_1 - \boldsymbol{x}_2\|$. We have

$$(\boldsymbol{u}_1 + (\overline{\boldsymbol{u}} - \overline{\boldsymbol{u}}) - \boldsymbol{u}^\star(\boldsymbol{x}_1))^\top \left(\nabla H(\boldsymbol{x}_1, \boldsymbol{u}_1) + (\nabla H(\boldsymbol{x}_1, \overline{\boldsymbol{u}}) - \nabla H(\boldsymbol{x}_1, \overline{\boldsymbol{u}})) - \nabla H\left(\boldsymbol{x}_1, \boldsymbol{u}^\star(\boldsymbol{x}_1)\right)\right)$$

$$+ \nu\|\boldsymbol{u}_1 + (\overline{\boldsymbol{u}} - \overline{\boldsymbol{u}}) - \boldsymbol{u}^\star(\boldsymbol{x}_1)\|^2 \leq 0, \quad \forall \boldsymbol{u}_1 \in \mathcal{U}(\boldsymbol{x}_1).$$

We rearrange the latter display into

$$(\overline{\boldsymbol{u}} - \boldsymbol{u}^\star(\boldsymbol{x}_1))^\top \left(\nabla H(\boldsymbol{x}_1, \overline{\boldsymbol{u}}) - \nabla H(\boldsymbol{x}_1, \boldsymbol{u}^\star(\boldsymbol{x}_1))\right) + \nu\|\overline{\boldsymbol{u}} - \boldsymbol{u}^\star(\boldsymbol{x}_1)\|^2$$

$$\leq \underbrace{- (\overline{\boldsymbol{u}} - \boldsymbol{u}^\star(\boldsymbol{x}_1))^\top \left(\nabla H(\boldsymbol{x}_1, \boldsymbol{u}_1) - \nabla H(\boldsymbol{x}_1, \overline{\boldsymbol{u}})\right)}_{\Omega_1}$$

$$\underbrace{- (\boldsymbol{u}_1 - \overline{\boldsymbol{u}})^\top \left(\nabla H(\boldsymbol{x}_1, \boldsymbol{u}_1) - \nabla H(\boldsymbol{x}_1, \boldsymbol{u}^\star(\boldsymbol{x}_1))\right)}_{\Omega_2}$$

$$\underbrace{-\nu\|\boldsymbol{u}_1 - \overline{\boldsymbol{u}}\|^2 - 2\nu\langle \boldsymbol{u}_1 - \overline{\boldsymbol{u}}, \overline{\boldsymbol{u}} - \boldsymbol{u}^\star(\boldsymbol{x}_1)\rangle}_{\Omega_3}.$$

We bound $\Omega_1, \Omega_2$, and $\Omega_3$ separately.

- For $\Omega_1$, we have

$$-(\overline{\boldsymbol{u}} - \boldsymbol{u}^\star(\boldsymbol{x}_1))^\top \left(\nabla H(\boldsymbol{x}_1, \boldsymbol{u}_1) - \nabla H(\boldsymbol{x}_1, \overline{\boldsymbol{u}})\right) \leq \mathrm{Diam}_{\mathcal{U}} \cdot \ell_H \|\boldsymbol{u}_1 - \overline{\boldsymbol{u}}\|$$
$$\leq \mathrm{Diam}_{\mathcal{U}} \ell_H L_c \|\boldsymbol{x}_1 - \boldsymbol{x}_2\|.$$

- For $\Omega_2$, it holds that

$$-(\boldsymbol{u}_1 - \overline{\boldsymbol{u}})^\top \left(\nabla H(\boldsymbol{x}_1, \boldsymbol{u}_1) - \nabla H(\boldsymbol{x}_1, \boldsymbol{u}^\star(\boldsymbol{x}_1))\right) \leq L_c \|\boldsymbol{x}_1 - \boldsymbol{x}_2\| \cdot \ell_H \mathrm{Diam}_{\mathcal{U}}.$$

- For $\Omega_3$, since the first term is always non-positive, we only need to bound the second term:

$$-\langle \boldsymbol{u}_1 - \overline{\boldsymbol{u}}, \overline{\boldsymbol{u}} - \boldsymbol{u}^\star(\boldsymbol{x}_1)\rangle \leq \|\boldsymbol{u}_1 - \overline{\boldsymbol{u}}\| \|\overline{\boldsymbol{u}} - \boldsymbol{u}^\star(\boldsymbol{x}_1)\|$$
$$\leq L_c \|\boldsymbol{x}_1 - \boldsymbol{x}_2\| \cdot \mathrm{Diam}_{\mathcal{U}}.$$

Combining $\Omega_1, \Omega_2$, and $\Omega_3$, we conclude that

$$(\overline{\boldsymbol{u}} - \boldsymbol{u}^\star(\boldsymbol{x}_1))^\top \left(\nabla H(\boldsymbol{x}_1, \overline{\boldsymbol{u}}) - \nabla H(\boldsymbol{x}_1, \boldsymbol{u}^\star(\boldsymbol{x}_1))\right) + \nu\|\overline{\boldsymbol{u}} - \boldsymbol{u}^\star(\boldsymbol{x}_1)\|^2$$
$$\leq (1 + 2\ell_H)L_c \mathrm{Diam}_{\mathcal{U}} \|\boldsymbol{x}_1 - \boldsymbol{x}_2\|. \tag{64}$$

Plugging in $\overline{\boldsymbol{u}} \leftarrow \boldsymbol{u}^\star(\boldsymbol{x}_2)$ in (64) and combine it with (63), we get

$$\nu\|\boldsymbol{u}^\star(\boldsymbol{x}_2) - \boldsymbol{u}^\star(\boldsymbol{x}_1)\|^2 \leq (\boldsymbol{u}^\star(\boldsymbol{x}_2) - \boldsymbol{u}^\star(\boldsymbol{x}_1))^\top \left(\nabla H(\boldsymbol{x}_2, \boldsymbol{u}^\star(\boldsymbol{x}_2)) - \nabla H(\boldsymbol{x}_1, \boldsymbol{u}^\star(\boldsymbol{x}_2))\right)$$
$$+ L''\|\boldsymbol{x}_1 - \boldsymbol{x}_2\|$$
$$\leq \ell_H\|\boldsymbol{u}^\star(\boldsymbol{x}_2) - \boldsymbol{u}^\star(\boldsymbol{x}_1)\|\|\boldsymbol{x}_1 - \boldsymbol{x}_2\| + L''\|\boldsymbol{x}_1 - \boldsymbol{x}_2\|.$$

Where $L'' = (1 + 2\ell_H)L_c \mathrm{Diam}_{\mathcal{U}} + 2L_c L_H$.

Similarly to Lemma C.8, we can set $\lambda = \|\boldsymbol{u}^\star(\boldsymbol{x}_2) - \boldsymbol{u}^\star(\boldsymbol{x}_1)\|$ and $\chi = \|\boldsymbol{x}_1 - \boldsymbol{x}_2\|$ and consider the inequality $\nu\lambda^2 \leq \ell_H\lambda\chi + L''\chi$. We aim to find the solution of the form $\frac{\ell_H\chi + \sqrt{\chi(4\nu L'' + \ell_H^2\chi)}}{2\nu} \leq c\sqrt{\chi}$. By setting $L_\star = c$ and solve for $c$ gives

$$L_\star = \frac{1}{2\nu}\left(2\ell_H\sqrt{\mathrm{Diam}_{\mathcal{X}}} + 2\sqrt{\nu(1 + 2\ell_H)L_c\mathrm{Diam}_{\mathcal{U}} + 2\nu L_c L_H}\right).$$

$\square$

Finally, we show that $\Phi$ is differentiable and Hölder-continuous.

**Theorem D.2.** *Let function $\Phi$ be $\Phi(\boldsymbol{x}) := \max_{\boldsymbol{u}\in\mathcal{U}(\boldsymbol{x})}\{H(\boldsymbol{x}, \boldsymbol{u})\}$. Its gradient $\nabla\Phi$ is $(1/2, \ell_{1/2})$-Hölder continuous,*

$$\|\nabla\Phi(\boldsymbol{x}) - \nabla\Phi(\boldsymbol{x}')\| \leq \ell_{1/2}\|\boldsymbol{x} - \boldsymbol{x}'\|^{\frac{1}{2}},$$

*where $\ell_{1/2} := \left((1 + L_{c^{-1}})\sqrt{\mathrm{Diam}_{\mathcal{X}}} + L_{c^{-1}}L_\star\right)\ell$.*

**Proof.**

$$\|\nabla\Phi(\boldsymbol{x}) - \nabla\Phi(\boldsymbol{x}')\| = \left\|\nabla f\left(\boldsymbol{x}, c^{-1}(\boldsymbol{u}^\star(\boldsymbol{x}); \boldsymbol{x})\right) - \nabla f\left(\boldsymbol{x}', c^{-1}(\boldsymbol{u}^\star(\boldsymbol{x}'); \boldsymbol{x}')\right)\right\|$$

$$\leq \ell \|\boldsymbol{x} - \boldsymbol{x}'\| + \ell \left\|c^{-1}(\boldsymbol{u}^\star(\boldsymbol{x}); \boldsymbol{x}) - c^{-1}(\boldsymbol{u}^\star(\boldsymbol{x}'); \boldsymbol{x}')\right\|$$

$$\leq \ell \|\boldsymbol{x} - \boldsymbol{x}'\| + \ell L_{c^{-1}}\left(\|\boldsymbol{u}^\star(\boldsymbol{x}) - \boldsymbol{u}^\star(\boldsymbol{x}')\| + \|\boldsymbol{x} - \boldsymbol{x}'\|\right) \quad (65)$$

$$\leq (1 + L_{c^{-1}})\ell \|\boldsymbol{x} - \boldsymbol{x}'\| + L_{c^{-1}}L_\star\ell \|\boldsymbol{x} - \boldsymbol{x}'\|^{\frac{1}{2}} \quad (66)$$

$$\leq \left((1 + L_{c^{-1}})\sqrt{\mathrm{Diam}_{\mathcal{X}}} + L_{c^{-1}}L_\star\right)\ell \|\boldsymbol{x} - \boldsymbol{x}'\|^{\frac{1}{2}}.$$

Where

- in (65) we invoke the Lipschitz continuity of function $c^{-1}(\cdot)$;
- (66) follows from Theorem D.1.

$\square$

Following, SGDMAX is presented where we assume a stochastic gradient oracle $G = (G_x, Gy) : \mathcal{X} \times \mathcal{Y} \times \Xi \to \mathbb{R}^d$ that is unbiased and has a bounded variance:

---
**Algorithm 3** SGDMAX
---
**Input:** Initialization $\boldsymbol{x}^{(0)}$, stepsize $\eta_x$, $T_x$ iterations, batch size $M$, oracle accuracy $\zeta$.
1: **for** $t \leftarrow 1, 2, \ldots, T$ **do**
2: $\quad \boldsymbol{y}^{(t)} \leftarrow \mathsf{max\text{-}oracle}\left(f(\boldsymbol{x}^{(t)}, \cdot); \zeta\right)$
3: $\quad \hat{\boldsymbol{g}}^{(t)} \leftarrow \frac{1}{M}\sum_{j=1}^{M} G_x\left(\boldsymbol{x}^{(t-1)}, \boldsymbol{y}^{(t)}, \xi_j^{(t)}\right)$
4: $\quad \boldsymbol{x}^{(t)} \leftarrow \mathrm{Proj}_{\mathcal{X}}\left(\boldsymbol{x}_i^{(t-1)} - \eta_x\hat{\boldsymbol{g}}^{(t)}\right)$
5: **end for**
6: $\boldsymbol{y}^{(T+1)} \leftarrow \mathsf{max\text{-}oracle}\left(f(\boldsymbol{x}^{(T)}, \cdot); \zeta\right)$

---

Finally, we can state the theorem of convergence to an $\epsilon$-approximate saddle-point.

**Theorem D.3.** *Let a function $f$ as the one in Theorem D.1. For a desired accuracy $\epsilon > 0$, Algorithm 3, (SGDMAX) with a tuning of $T_x = O\left(\frac{\ell_{1/2}^2}{\epsilon^3}\right)$, $\eta_x$, a* max-oracle *accuracy $\zeta = O\left(\frac{\nu\epsilon^2}{\ell^2}\right)$, and a batch size of $M = \max\left\{1, \frac{9\sigma^2}{2\epsilon^2}\right\}$ guarantees that there exists a $t^* \in [T]$, such that,*

$$-\nabla_x f\left(\boldsymbol{x}^{(t^\star)}, \boldsymbol{y}^{(t^\star+1)}\right)^\top\left(\boldsymbol{x}' - \boldsymbol{x}^{(t^\star)}\right) \leq \epsilon, \forall \boldsymbol{x}' \in \mathcal{X};$$

$$\nabla_y f\left(\boldsymbol{x}^{(t^\star)}, \boldsymbol{y}^{(t^\star+1)}\right)^\top\left(\boldsymbol{y}' - \boldsymbol{y}^{(t^\star)}\right) \leq \epsilon, \forall \boldsymbol{y}' \in \mathcal{Y}.$$

*Further, the* max-oracle, *of accuracy $\zeta$, can be implemented by $T_y = \tilde{O}\left(\frac{L}{L_c^2\nu} + \frac{L\sigma^2}{L_c^4+\nu^2}\frac{1}{\zeta}\right)$ iterations of stochastic projected gradient ascent with a step size $\eta_y = \min\left\{\frac{2}{9L}, \frac{L_c^2\nu\zeta}{10L\sigma^2}\right\}$.*

**Proof.** The proof follows easily from the proof of projected gradient ascent in hidden-strongly-concave function found [43, Theorem 6] and Theorems B.1 and D.2. $\square$

**Remark 2.** *It has been shown that when a function $f$ enjoys a hidden-strongly-concave reformulation, it satisfies global the Proximal-PŁ condition (or equivalently, global KŁ condition) [43, 67]. While the equivalence between global KŁ condition and quadratic growth condition has been proven [10, 38] when $f$ is concave, to the authors' best knowledge, this equivalence still remains unclear when $f$ is nonconcave. This means that we cannot use [83] to prove the smoothness of the maximum function when the feasibility set of the maximizing variable is constrained.*

