# OpenReview forum: "Learning Equilibria in Adversarial Team Markov Games: A Nonconvex-Hidden-Concave Min-Max Optimization Problem"
_NeurIPS.cc/2024/Conference — NeurIPS 2024 poster_

### Official Review · Reviewer_hE5h · 2024-06-23

**Soundness:** 2
**Presentation:** 1
**Contribution:** 2
**Rating:** 4
**Confidence:** 4

**Summary:**

The paper studies policy gradient methods for computing a Nash equilibrium in adversarial team Markov games. The authors employ an occupancy measure-based regularization to deal with this non-convex minimax optimization problem. They develop a policy gradient method that allows all team members to take policy gradient steps independently, while the adversary takes regularized policy gradient steps. The authors prove that this new method finds a near Nash equilibrium in polynomial iteration/sample complexities, under some mild conditions. The authors further show the tractability of computing a near saddle point for a class of structured non-convex minimax optimization problems.

**Strengths:**

- The authors propose a policy gradient method to find a near Nash equilibrium of adversarial team Markov games, with polynomial iteration/sample complexities. This policy gradient method appear to be new for this problem.

- A new regularization technique is introduced for the adversarial player, which avoids using linear programming in the existing method.

**Weaknesses:**

- The paper has several writing issues: (i) Introduction with two paragraphs is discouraged for NeurIPS; (ii) Technical overview is verbose, and not all the content is original to this paper; (iii) The preliminary section is cumbersome and contains confusing notation for occupancy measures; (iv) The connection between stochastic gradient and team problems in Section 3 reads ambiguous; (v) The results in Section 4 are not directly related to adversarial team games.

- The motivation for using the policy gradient method to solve adversarial team Markov games is not explained from the perspective of application. It is hard to evaluate the broad impact of this work.

- The motivation for using regularization based on occupancy measures is not well presented. The authors implement regularization directly in Algorithm 1 without explaining the reasons behind it.

- It is not clear why bandit feedback is important in adversarial team Markov games since the proposed policy gradient method assumes simulated game-play, which is not a typical online learning algorithm.

- The proposed policy gradient method assumes policy coordination between team and adversary, which often is not the case in practice.

- Except for regularization for the adversarial player, other techniques are quite standard for policy gradient methods. The novelty of the proposed method is limited or the effectiveness of regularization should be emphasized.

- The polynomial iteration/sample complexities of the proposed policy gradient method are highly sub-optimal in terms of rates and problem dependence.

- The proposed policy gradient method is limited to problems with small state/action spaces, leading dimension scalability issue.

- Conditions for Theorem 3.3 to hold are not explained, hiding limitations of the proposed policy gradient method. Also, the proof of Theorem 3.3 in Appendix should be presented in a more readable way.

- There are no experimental justifications of the proposed policy gradient method.

**Questions:**

Here are other questions for improvement.

- What rewards do team take? It is not clear from Section 2.2.

- What is reformulation in line 207? a confusing inter product notation.

- unclear notation: $\Delta(\mathcal{A})$, $\boldsymbol{r}(\boldsymbol{x})$, $\hat{\boldsymbol{g}}_i$, $\mathcal{K}^{(t)}$, $\boldsymbol{x}^{(T)}$, etc.

- Why do you need $\zeta$-greedy policy? How does policy gradient projection work in Algorithms 1-2?

- wrong index: $k\in[n]$ lines 3-4 of Algorithm 1.

---

> ### Author Rebuttal · Authors · 2024-08-06
>
> We thank the reviewer for their valuable time and recommendations. We are committed to enhancing the quality of our draft and will incorporate your suggestions.
> Beginning, could the reviewer specify what they find lacking in our paper's soundness? Soundness in a theoretical paper pertains to the correctness of mathematical arguments, and we are eager to identify any flaws in our reasoning.
> We proceed to answer the reviewer’s questions and comment on the highlighted weaknesses:
>
> > **Q**: What rewards do team take?
>
> Every agent $i$ gets a reward $r_i = - \frac{1}{n} r$. And it holds that $\sum_{i}^n r_i + r = 0$.
>
> > **Q** : What is reformulation in line 207?
>
> The reformulation in line 207 is the very common reformulation of the value function from a function of the policy to a funciton of the state-action measures, see Section 2 in [1].  It is equal to $\sum_{s,b}r(s,x,b)\lambda(s,b) = r(x)^\top \lambda$. See comments for a simple explanation.
>
> > **Q** : Why do you need $\zeta$-greedy policy? How does policy gradient projection work in Algorithms 1-2?
>
> The $\zeta$-greedy policy ensures that the variance of the reinforce gradient estimator is bounded. See lines 222-226.
>
> Almost identical to project to the non-truncated simplex, for a potentially infeasible $x^+ = x - \eta \nabla f(x)$, projection is defined as the $ \arg \min_{x' \in \mathcal{X_\zeta} } \frac{1}{2} || {x^+ - x'} || ^2  $. This optimization problem is solved in polynomial time using quadratic programming.
>
>
> > **Q** :  wrong index: $k\in[n]$ lines 3-4 of Algorithm 1.
>
> Thank you for pointing this out, it is $i\in[n]$.
>
> > unclear notation: $\Delta(\mathcal{A}), \boldsymbol{r}(\boldsymbol{x}), \hat{\boldsymbol{g}}_i, \mathcal{K}^{(t)}, \boldsymbol{x}^{(T)}$ etc.
>
> We will add more definitions
>
> $\Delta(\mathcal{A})$: the very standard notation of a simplex supported on $\mathcal{A}$.
>
> $\boldsymbol{r}(\boldsymbol{x}) \in \mathbb{R}^{|\mathcal{S}| \times |\mathcal{B}|}$ is the expected reward of the adversary.
>
> $ \mathcal{K}^{(t)} $ is a batch of trajectories.
>
> $\hat{\boldsymbol{g}}_i $ is the gradient estimate at a given iteration of the algorithm.
>
> $\boldsymbol{x}^{(T)}$ is the vector of the concatenated policies of the team at iteration $T$.
>
> ---
>
> With regards to the highlighted weaknesses of our paper:
> > *"(ii) Technical overview is verbose, and not all the content is original to this paper"*
>
> We present a detailed technical overview, using various tools to address nonconvex-nonconcave minmax problem challenges. This clarity helps readers grasp the necessity and functionality of our arguments.
> While not all content is entirely original, as typical in most papers, our work builds on and advances existing research through rigorous scientific dialogue.
>
> >*”The results in Section 4 are not directly related to adversarial team games.”*
>
> We dedicated a separate section to generalize our results particularly. They are directly related from an optimization perspective.
>
> > *"The motivation for using regularization based on occupancy measures is not well presented. The authors implement regularization directly in Algorithm 1 without explaining the reasons behind it."*
>
> We believe this is inaccurate. We have actually dedicated multiple lines of our text for this purpose. Namely:
> * In the Technical Overview: Lines 107-132
> * Lines 281-285
> * Section C.3
> * Lines 982-1004
>
> > *“It is not clear why bandit feedback is important in adversarial team Markov games since [...]”*
>
> See global rebuttal (Self-Play is Inevitable).
>
> This assumption is inaccurate. Coordination only occurs during the learning process: players pause policy updates to collect trajectory samples, a standard practice in most theoretical AGT/MARL papers (e.g., [5-13]). The adversary optimizes its policy before team agents independently perform a gradient step. The first is a common standard in our field, and the second is a minor assumption.
>
> >*"The motivation for using the policy gradient method to solve adversarial team Markov games is not explained from the perspective of application"*
>
> Please see global rebuttal (Importance of Policy Gradient Methods).
>
> > *”no experimental justifications”*
>
> See global rebuttal.
>
> > *” Except for regularization for the adversarial player, other techniques are quite standard for policy gradient methods…”*
>
> We use standard techniques policy gradient techniques on purpose. The idea of this paper was to take methods that are broadly utilized in (MA)RL and prove that they can be minorly tweaked to get provable guarantees for the very challenging task of computing a NE.
>
> Please see the “Novelty in our paper” note in the global rebuttal.
>
> > *”The polynomial iteration/sample complexities of the proposed policy gradient method are highly sub-optimal in terms of rates and problem dependence.”*
>
> This work offers the first proof that an algorithm of polynomial sample complexity exists for NE in ATMGs. We cannot accept the sub-optimality claim without any formal argument.
>
> Also, see the global rebuttal.
>
> > *"The proposed policy gradient method is limited to problems with small state/action spaces, leading dimension scalability issue.
> "*
>
> RL methods typically don't scale well with large action and state spaces unless function approximation is used along certain extra assumptions on the MDP/MG. Recently, function approximation in multi-agent reinforcement learning (MARL) has gained theoretical interest. See [3] for recent results.
>
>
> > *”Conditions for Theorem 3.3 to hold are not explained, hiding limitations of the proposed policy gradient method. Also, the proof of Theorem 3.3 in Appendix should be presented in a more readable way.“*
>
> We do not hide any limitations. The conditions for theorem 3.3 to hold is merely that the Markov game follows the structure of an ATMG as defined in Section 2. This theorem is general.
>
> In lines 981-996 we offer a small summary of the steps of our arguments. We welcome more suggestions.

---

> > ### Comment · Reviewer_hE5h · 2024-08-12
> >
> > Thank you for your response. I would like to keep my initial evaluation.

---

> ### Author Response · Authors · 2024-08-06
> **Value reformulation and References**
>
> __Reformulation__
>
> Let us restate some definitions to make this reformulation clearer. We start from the value function, which is a scalar:
> $\begin{align}
>  V_{\rho}(x,y) = E  {[ \sum_{h=1}^{\infty} \gamma^{h-1} r( s^{(h)}, x,b) | s^{(0)} \sim \rho ]}
> \end{align}$.
>
> Then, we define the adversary's state-action visitation measure, which can be thought of as a vector in ${R}^{|S||B|}$:
> $$ \lambda_{s,b}( y; x) =  E [ \sum_{h=1}^{\infty} \gamma^{h-1} \mathbb{I} (s^{(h)} = s, b^{(h)} = b) | s^{(0)} \sim \rho ]   $$
>
> Finally, $(s,b)$-th entry of the vector $\boldsymbol{r}(\boldsymbol{x}) \in \mathbb{R}^{|S| |B|}$.
> $$ {r}_{s,b}(\boldsymbol{x}) =   E [r(s,a, b)] $$ with the expectations taken over $\boldsymbol{a}\sim \boldsymbol{x}$.
>
> Hence, the inner product $\boldsymbol{r}(\boldsymbol{x}) ^\top \boldsymbol{\lambda}(\boldsymbol{y}; \boldsymbol{x})$ will be equal to the summation:
> $$ \sum_{(s,b) } \Big(  {\mathbb{E}} [r(s,\boldsymbol{a}, b)] \cdot {\mathbb{E}} [ \sum_{h=1}^{\infty} \gamma^{h-1} \mathbb{I} (s^{(h)} = s, b^{(h)} = b) | s^{(0)} \sim \rho ] \Big)  .$$
>
> From the last, display we can see that the inner product is equal to the value function.
>
> We would thank the reviewer again for spending their valuable time to provide feedbacks. We would really appreciate if the author would consider raising their score.
>
> Best regards,
>
> The authors
>
> ---
>
> __References__
>
> [1] Zhang, J., Koppel, A., Bedi, A.S., Szepesvari, C. and Wang, M., 2020. Variational policy gradient method for reinforcement learning with general utilities. NeurIPS
>
> [2] Kalogiannis, F., Anagnostides, I., Panageas, I., Vlatakis-Gkaragkounis, E.V., Chatziafratis, V. and Stavroulakis, S.A., Efficiently Computing Nash Equilibria in Adversarial Team Markov Games. ICLR
>
> [3] Cui, Q., Zhang, K. and Du, S., 2023, July. Breaking the curse of multiagents in a large state space: Rl in markov games with independent linear function approximation. COLT
>
> [4] Agarwal, A., Kakade, S.M., Lee, J.D. and Mahajan, G., 2021. On the theory of policy gradient methods: Optimality, approximation, and distribution shift. JMLR
>
> [5] Daskalakis, C., Foster, D.J. and Golowich, N., 2020. Independent policy gradient methods for competitive reinforcement learning. NeurIPS
>
> [6] Zhang, R., Mei, J., Dai, B., Schuurmans, D. and Li, N., 2022. On the global convergence rates of decentralized softmax gradient play in markov potential games. NeurIPS
>
> [7] Wei, C.Y., Lee, C.W., Zhang, M. and Luo, H., 2021, July. Last-iterate convergence of decentralized optimistic gradient descent/ascent in infinite-horizon competitive markov games. COLT
>
> [8] Ding, D., Wei, C.Y., Zhang, K. and Jovanovic, M., 2022, June. Independent policy gradient for large-scale markov potential games: Sharper rates, function approximation, and game-agnostic convergence. ICML
>
> [9] Leonardos, S., Overman, W., Panageas, I. and Piliouras, G., Global Convergence of Multi-Agent Policy Gradient in Markov Potential Games. ICLR
>
> [10] Erez, L., Lancewicki, T., Sherman, U., Koren, T. and Mansour, Y., 2023, July. Regret minimization and convergence to equilibria in general-sum markov games. ICML
>
> [11] Giannou, A., Lotidis, K., Mertikopoulos, P. and Vlatakis-Gkaragkounis, E.V., 2022. On the convergence of policy gradient methods to Nash equilibria in general stochastic games. NeurIPS
>
> [12] Park, C., Zhang, K. and Ozdaglar, A., 2024. Multi-player zero-sum Markov games with networked separable interactions. NeurIPS
>
> [13] Cen, S., Chi, Y., Du, S. and Xiao, L., 2023, January. Faster last-iterate convergence of policy optimization in zero-sum Markov games. ICLR
>
> [14] Bai, Y., Jin, C. and Yu, T., 2020. Near-optimal reinforcement learning with self-play. NeurIPS

---

> ### Author Response · Authors · 2024-08-12
>
> Dear Reviewer,
>
> We thank you for your response. **We would appreciate it if you could specify how we have not fully addressed your concerns.** Understanding your perspective is important to us. We feel confident that our responses were thorough, and it is disappointing to receive limited feedback given the effort we put into our rebuttal.
>
> Please remember that *this is a theoretical paper*, and we believe it should be judged according to the relevant standards. It is vital for the benefit of the broader community that your counterarguments are shared with the authors.
>
> In our view, the discussion phase is vital for fostering scientific dialogue between authors and reviewers, which is crucial for maintaining the conference's quality. This interaction enhances the overall quality of research and papers, whether or not they are eventually published at this venue.
>
> Sincerely,
>
> The Authors

---

### Official Review · Reviewer_mZxv · 2024-07-12

**Soundness:** 2
**Presentation:** 3
**Contribution:** 3
**Rating:** 5
**Confidence:** 3

**Summary:**

This paper provides multi-agent RL policy gradient method with polynomial guarantees (iteration and sample complexity) for the adversarial team Markov games problem setting.

**Strengths:**

This paper addresses the main open question from https://openreview.net/forum?id=mjzm6btqgV, that is, developing a policy gradient learning algorithm for the adversarial team Markov games problem. The paper is well-written.

**Weaknesses:**

After the first pass of this work, I did not have many weaknesses for this work as it *made good progress* from the recent ICLR 2023 work https://openreview.net/forum?id=mjzm6btqgV.

But after the following point:

- Theorem 3.3, the main result, the polynomial dependence of both sample complexity and iterations on the **all** parameters is huge! Even with $\gamma=0.9$, sample complexity scales as $10^{93}$. The number of atoms in the current universe scales as approx $10^{84}$. *This is disregarding other factors such as states, actions, etc into sample complexity!* Whereas, most practical MARL works like Diplomacy [6], Starcraft (https://deepmind.google/discover/blog/alphastar-mastering-the-real-time-strategy-game-starcraft-ii/), etc, beat human players at a reasonable training time (approx hours/days?). Moreover, ICLR 2023 work https://openreview.net/forum?id=mjzm6btqgV sample complexity scales as $10^{32}$ (all parameters included in this)

I just can't get over what side I should choose: happy for such learning algorithm for ATMG problem or unsatisfied with the tools used with such huge *polynomial (but almost exponential with 10s of states, for example)* suboptimal guarantees.

To err on the side of this work and ML community progress, I am open to hearing any defense for this. Towards this, I am looking forward to your reply to the below two points.

- Maybe one way is to show hardness results on some finite state-action ATMGs?
- Please highlight some technical innovations in this work compared to the zero-sum MG sample complexity analysis. Are there points of improvement to be made?

Future work statement from line 329:

> variance-reduction techniques to achieve a better sample complexity

Is there any conjecture as to how much VR methods save in terms of sample complexity?

As I am in a dilemma, my current score reflects that. I'll update it after the authors-reviewers discussion period. Thanks!

**Questions:**

na

---

> ### Author Rebuttal · Authors · 2024-08-06
>
> We thank the author for their comments that motivated us to reflect on our work. Following, we try to address your concerns.
> > *"Theorem 3.3, the main result, the polynomial dependence of both sample complexity and iterations on the all parameters is huge! [..] This is disregarding other factors such as states, actions, etc into sample complexity![..] Moreover, ICLR 2023 work [..] sample complexity scales as (all parameters included in this)"*
>
> First, the ICLR'23 work does not provide sample complexity results; their algorithm assumes access to a gradient oracle. Second, we want to remind the reviewer that the bounds we provide are upper bounds for the worst case. Empirical performance may be a lot better -- compare this to the exponential worst-case complexity of the simplex algorithm for linear programming.
>
> Maybe the bounds can be tightened but they remain the first polynomial bounds in the sample complexity and iteration complexity without access to the reward and transition functions. We remind the reviewer that providing polynomial guarantees with large coefficients are not uncommon for theoretical works that aim to learn a NE in MGs. For example, a highly-cited previous work [2], has a dependence that scales up to $10^{48.5}$ for a much simpler setting.
>
> > *”most practical MARL works [..] beat human players at a reasonable training time” (approx hours/days?)*
>
> This is a very interesting point. Achieving super-human performance is easier than computing a Nash equilibrium. In order to achieve super-human performance, an AI agent needs only to be able to optimize against its opponents’ best responses. This is known as a Stackelberg equilibrium problem, which can be solved in polynomial time even for a general-sum game [1].
>
> > *”I just can't get over what side I should choose: happy for such learning algorithm for ATMG problem or unsatisfied with the tools used with such huge polynomial (but almost exponential with 10s of states, for example) suboptimal guarantees.”*
>
> We stress that polynomial complexity is a very important theoretical qualitative and has quantitative differences from an exponential one. Whether our guarantee on the upper-bound of the worst-case is suboptimal is up to discussion and it should be formally argued (see global rebuttal). The degrees of the polynomial are not significantly larger than previous work on simpler settings [2].
>
> >*”Maybe one way is to show hardness results on some finite state-action ATMGs?”*
>
> Proving some lower bounds for the complexity of computing a NE in ATMGs is a very interesting future research direction. However, this paper aims to give the first theoretical polynomial upper bounds for model-free learning of NE in ATMGs. Notwithstanding, deriving some lower bounds goes very well beyond the scope of our paper and would probably be a standalone result for a new paper.
>
> >*”Please highlight some technical innovations in this work compared to the zero-sum MG sample complexity analysis. Are there points of improvement to be made?”*
>
> We  highlight the technical challenges we faced and overcame.
>
> Namely, all of the following properties hold in a zero-sum MG:
>
> (i) the duality gap is zero as proven by Shapley; i.e., $\min_x \max_y V(x,y) = \max_y \min_x V(x,y)$
>
> (ii) the value of all Nash equilibria is the same
>
> (iii) the NEs are exchangeable, e.g. let two NEs $(x,y)$ and $(x’,y’)$. Then, policy profiles $(x’,y)$ and $(x,y’)$ will also be an NE.
>
> (iv) the optimization landscape is that of a hidden-convex–hidden-concave function.
>
> (v) any Markovian coarse-correlated equilibrium policy can be marginalized to a product policy that will be a NE.
>
> On the other hand, (i)-(v) generally fail to hold for ATMGs even when the game has only one state (i.e., in normal-form games).
> (i)-(iii) are indispensable in the design and guarantees of existing algorithms for zero-sum MGs and since they fail to hold for ATMGs, they cannot be utilized.
>
> Despite all these challenges, we manage to show that we do not need access to the reward and transition functions of the game in order to compute a NE. Additionally, this task can be achieved when the agents only collect a finite number of sample trajectories.
>
> To do so, we introduce a double-loop policy gradient algorithm. In the inner-loop the adversary optimizes a regularized function and in the outer loop the team player take independent gradient steps.
>
> The key difference between our work and [65] is that of the regularization. Our work remains the first to show that such regularization makes the outer-loop process equivalent to optimizing a nonconvex function with Holder continuous gradients (a significantly weaker property compared to the omnipresent assumption of Lipschitz continuity in ML and RL settings).  Further, the team agents have inexact gradient feedback.
> The *inexactness* is due to the gradient of the regularizer is not zero w.r.t. the team agents’ policies and the team agents have no information about the regularizer (otherwise we would allow a significant communication among agents).
>
> But:
>
> The stochasticity error (bias and variance) are bounded by standard techniques (the REINFORCE gradient estimator along $\zeta$-greedy parametrization).
>
> The inexactness error is controlled by very carefully choice of the regularizing coefficient so that it strikes a balance to not make the Holder constant too large while maintaining a inexactness error small (these two objectives are in tension with each other).
>
> We again thank the reviewer for their time and giving us the chance to elucidate our work. We hope that we have made parts of our work clearer and easier to evaluate. We would really appreciate if you would consider raising our score.
>
> Best regards,
>
> The authors
>
> [1] Conitzer, V. and Sandholm, T., 2006, June. Computing the optimal strategy to commit to. EC
>
> [2] Daskalakis, C., Foster, D.J. and Golowich, N., 2020. Independent policy gradient methods for competitive reinforcement learning. NeurIPS

---

> ### Author Response · Authors · 2024-08-06
> **Variance reduction and potential improvements on sample complexity**
>
> > *"Is there any conjecture as to how much VR methods save in terms of sample complexity?"*
>
> We believe that it might be possible through a careful design of the estimators (possibly along with a different parameterization of the policy) and bypass the $\zeta$-greedy technique to bound the variance. In this case the sample complexity can be improved by a factor of $O((1 - \gamma)^{15} \epsilon^3)$.

---

> > ### Comment · Reviewer_mZxv · 2024-08-12
> >
> > Thank you for reflecting on the reviews. I have updated my score from 4 to 5. Good luck.

---

### Official Review · Reviewer_LYwL · 2024-07-13

**Soundness:** 3
**Presentation:** 3
**Contribution:** 2
**Rating:** 4
**Confidence:** 4

**Summary:**

This papers addresses learning equilibria in adversarial team Markov games, where a team of agents sharing a common reward function competes against a single adversary. Previously, [65] addressed this problem for the model-based case with no sample complexity guarantees. On the other hand, this paper presents a learning algorithm for model-free cases with bandit feedback and provide polynomial sample complexity guarantees.

**Strengths:**

- The paper is well-organized. We can see the effort to make the complex ideas accessible for the reader.
- There is a comprehensive literature review citing more than a hundred papers.

**Weaknesses:**

- Algorithm descriptions, i.e., Algorithms 1 and 2, are confusing. For example, Algorithm 1 uses VIS-REG-PG as a function. However, VIS-REG-PG has not been presented as a function getting x as an input in Algorithm 2. The function REINFORCE has not been provided explicitly. Although REINFORCE is well-known, its explicit description in the paper could improve the paper's completeness.
- Though the learning dynamics are claimed to be uncoupled, the agents including the adversary coordinate in collecting samples by playing according to some fixed strategy for some predetermined batch sizes. This contradicts with the adversarial nature of the adversary. Collecting samples is essential for model-free and bandit-feedback scenarios, which is a major contribution of the paper compared to [65].
- The bounds on the number of gradient updates and the total sample complexity in the main result, Theorem 3.3, depend on (possibly) very large constant terms since they are proportional to $\frac{1}{(1-\gamma)^{57}\epsilon^{10}}$ and $\frac{1}{(1-\gamma)^{93}\epsilon^{16}}$, respectively. This may not be feasible for many practical applications, e.g., when the discount factor is close to $1$ and we want small approximation error $\epsilon\approx 0$. This is a limitation because another major contribution of this paper compared to [65] is the sample complexity bounds.
- No numerical example is provided. Since the bounds appear large, numerical examples can give a better idea whether the algorithm presented is feasible for practical applications or not.

**Questions:**

- What are the practical applications of adversarial team Markov games where learning equilibrium via the algorithm presented can be of interest? For example, step 3 in Algorithm 1 uses REINFORCE with the batch size of M where all agents, including the adversary, keep playing according to fix strategy $(x^{t-1},y^t)$ so that the team members can collect samples and estimate $\hat{g}_i^t$. This contradicts with the adversarial nature of the adversary. Therefore, are we considering a simulated environment where we coordinate the team members and the adversary to learn some equilibrium of the underlying game?
- Adversarial team Markov games reduce to team Markov games if the adversary has a single action, and two-agent zero-sum Markov games if the team is a singleton. Can we say something similar for the algorithm presented? In other words, if the adversary is a non-strategic player with a single action, does the algorithm presented reduce to a known learning algorithm for team Markov games (or Markov potential games)? If the team has a single player, does the algorithm reduce to a known learning algorithm for two-agent zero-sum Markov games? If the answer is no, what are the differences?
- Can the authors rewrite the algorithm descriptions? For example, what is the function VIS-REG-PG? Is it the steps from 2 to 6 or the entire for loop in Algorithm 2? This is confusing since Algorithm 1 and 2 have different epoch numbers $T_x$ and $T_y$.
- Can the authors provide numerical examples? A comparison with the algorithm of [65] would also be very helpful to observe the impact of model-free and bandit-feedback cases on the convergence.

**Limitations:**

The limitations of the paper has not been discussed separately in an explicit way. The paper (1) requires the coordination of agents including the adversary in playing the same strategy for sampling trajectories in the estimation of some parameters important for the team members though this contradicts with the adversarial nature of the adversary, and (2) provides possibly very large bounds on the number of gradient updates and the sample complexities. (1) is important for addressing the model-free and bandit-feedback cases. (2) is essentially like asymptotic guarantees from a practical perspective. Those are the major limitations since addressing model-free and bandit-feedback cases, and providing sample complexity bounds are the main contributions of this paper compared to [65].

---

> ### Author Rebuttal · Authors · 2024-08-06
>
> We thank the reviewer for their suggestions and comments and the time they took to review our work.
>
> We would appreciate it if the reviewer reconsidered adding in the strengths of our paper any of the following:
>
> (i) The proof of convergence of inexact projected gradient descent for nonconvex optimization problems where the function has merely Holder-continuous gradients (Theorem 3.1).
>
> (ii) The elaborate way we managed to utilize standard MARL to prove convergence to a NE by utilizing finite samples.
>
> (iii) The way we proved that the MARL problem at hand boils down to optimizing a nonconvex function with Holder continuous gradients while we respect the requirement of minimal communication among agents during training.
>
> (iv) The involved mathematical arguments we use throughout the paper.
>
> Before proceeding we ask the reviewer to see  Global Rebuttal:”Self-play is inevitable” for some clarifying points.
>
>
> Following we answer the weaknesses and questions the reviewer pointed out.
>
> > *”Algorithm descriptions, i.e., Algorithms 1 and 2, are confusing. For example, Algorithm 1 uses VIS-REG-PG as a function [..] The function REINFORCE has not been provided explicitly.”*
>
> Thank you for pointing this out. Algorithm 2 should take as input the MDP that results from fixing the team players’ policies and return the optimal policy. As for the REINFORCE estimator please see section C.6.1.
>
>
> >*” The bounds on the number of gradient updates and the total sample complexity in the main result, Theorem 3.3, depend on (possibly) very large constant terms [...]. This is a limitation because another major contribution of this paper compared to [65] is the sample complexity bounds.”*
>
> Indeed, the dependence on the natural parameters of the game is large. Nevertheless, it does not stray a lot further than previous work in this field [2] and most importantly remains polynomial and beats the curse of the multiagents (it is polynomial in the number of players and the sum of individual action space sizes)!
>
> We remind the author that these are upper bound for the worst case. Empirical performance need not be as bad and the theoretical guarantees might be tightened. Unquestionably though, we offer the first polynomial sample complexity guarantee and this can possibly motivate further work to tighten the analysis and design more efficient algorithms.
>
>
> > **Q**: What are the practical applications of adversarial team Markov games where learning equilibrium via the algorithm presented can be of interest? For example, step 3 in Algorithm 1 uses REINFORCE with the batch size of M where all agents, including the adversary, keep playing according to fix strategy so that the team members can collect samples and estimate . This contradicts with the adversarial nature of the adversary. Therefore, are we considering a simulated environment where we coordinate the team members and the adversary to learn some equilibrium of the underlying game?
>
> And also
>
> >*”Though the learning dynamics are claimed to be uncoupled, the agents including the adversary coordinate in collecting samples [..]. This contradicts with the adversarial nature of the adversary. Collecting samples is essential for model-free and bandit-feedback scenarios, which is a major contribution of the paper compared to [65].”*
>
> Please see global rebuttal. The adversary should not be confused with what this name signifies in online optimization; it is merely the player that wants to maximze the function.
>
> Freezing in order to collect samples is commonplace in MARL theory papers and probably inevitable, see Global Rebuttal:Self-play is inevitable as to why.
>
> > **Q**: [...] does the algorithm presented reduce to a known learning algorithm for team Markov games (or Markov potential games)? If the team has a single player, does the algorithm reduce to a known learning algorithm for two-agent zero-sum Markov games? [...]?
> When the adversary has a singleton action, our algorithm is the same as the one proposed in [2] and computes a NE in any (Markov) potential game. Moreover, it handles utility functions that are not smooth but are merely Holder-continuous (a significantly weakened notion of continuity).
>
> In the case that the team is singleton, our algorithm guarantees convergence to a NE of the underlying game. It is novel but suboptimal compared to the recent policy gradient methods that are designed for two-player zero-sum Markov games.
>
>  > **Q**: Can the authors rewrite the algorithm descriptions? For example, what is the function VIS-REG-PG? Is it the steps from 2 to 6 or the entire for loop in Algorithm 2? This is confusing since Algorithm 1 and 2 have different epoch numbers $T_x$ and $T_y$ .
>
> Thank you for pointing this out. The tuning of the parameters is precisely stated in Theorem C.3. We will revise our draft to make the pseudo-code to reflect that.
>
> > **Q**: Can the authors provide numerical examples? A comparison with the algorithm of [65] would also be very helpful to observe the impact of model-free and bandit-feedback cases on the convergence.
>
> We will provide numerical experiments. Nevertheless, we would like to point out that not all theoretical papers need experiments. Comparing with [65] will of course give worse convergence rates as we are comparing full-information feedback versus inexact and stochastic feedback for a nonconvex optimization problem.
>
> We again thank the reviewer for their comments and suggestions. We would deeply appreciate if they considered increasing their score.
>
> Best regards,
>
> The authors
>
> [1] Daskalakis, C., Foster, D.J. and Golowich, N., 2020. Independent policy gradient methods for competitive reinforcement learning. NeurIPS
>
> [2] Leonardos, S., Overman, W., Panageas, I. and Piliouras, G., Global Convergence of Multi-Agent Policy Gradient in Markov Potential Games. ICLR

---

> > ### Comment · Reviewer_LYwL · 2024-08-12
> > **Acknowledgement of reviewing the rebuttal**
> >
> > I have reviewed the rebuttal. However, my questions have not been addressed satisfactorily. I will keep my score unchanged.

---

> > > ### Author Response · Authors · 2024-08-12
> > >
> > > Dear reviewer,
> > >
> > > Thank you for your reply. **We would kindly ask the reviewer to let us know in what ways we did not adequately address their concerns.** We would like to know what is the reviewer's thoughts. We are pretty confident that we answered your concerns sufficiently and the feedback you are giving us is particularly restricted, especially when we put a significant amount of effort in replying. Please again take into consideration that this is a theoretical paper and we strongly believe it should be judged by corresponding standards.
> > >
> > > We do not believe that it is fair to the broader community to not share your counterarguments with the authors of the paper.
> > >
> > > To our experience and understanding, the discussion phase is specifically meant to allow scientific dialogue between authors and reviewers to flourish. This is crucial to the conference's quality. This peer-reviewing process helps the community improve the overall quality of papers and research; regardless if a paper gets published to that particular venue or not.
> > >
> > > Sincerely,
> > >
> > > The authors

---

### Official Review · Reviewer_LXfR · 2024-07-20

**Soundness:** 2
**Presentation:** 3
**Contribution:** 3
**Rating:** 5
**Confidence:** 2

**Summary:**

This paper investigates the identification of NE in ATMGs. The authors explore the underlying landscape, leveraging optimization theory to effectively solve this problem.

**Strengths:**

Theorem 3.1 stands out with significant contribution. Writing is very clear.

**Weaknesses:**

The other theorems, in my view, merely adapt the MARL problem to a specific instance of an optimization problem (with an observation of hidden structure). The paper does not leverage the special structure of Markov Games (or Team Markov Games). Nonetheless, I appreciate the paper's clarity in writing and its theoretical contributions.

I have read the paper thoroughly, and while it presents a significant contribution, I believe it might not be the best fit for NeurIPS. The authors address a very classical optimization problem and apply their findings to an Adversarial Team Markov Game. Although Theorem 3.1 is particularly impressive, the subsequent theorems follow from it primarily due to the compactness and finite state and action space of MARL. This paper, lacking experimental results, seems more suitable for an optimization journal. I know several MG theory papers are already accepted to NeurIPS, even without any experiment. To be clear, I do not want to discourage authors about the fit for the venue.

**Questions:**

check weaknesses

**Limitations:**

check weaknesses

---

> ### Author Rebuttal · Authors · 2024-08-06
>
> We thank the reviewer for their precious time, suggestions, and their encouraging comments.
>
> > *“The other theorems, in my view, merely adapt the MARL problem to a specific instance of an optimization problem (with an observation of hidden structure). The paper does not leverage the special structure of Markov Games (or Team Markov Games).”*
>
> Indeed, we formalize an established MARL problem as a minmax optimization problem. This is a tradition that goes back to Von Neumann study of two-player zero-sum games long before the Nash ‘51 paper.
>
> We manage to tackle a MARL/AGT problem by utilizing techniques that are virtually omnipresent in (MA)RL; namely, policy gradient methods. We tweak them just enough to guarantee convergence and provable guarantees, but they still remain genuine MARL techniques. In fact, many technicalities and novelty of this paper are embedded in the theoretical analysis of the given provable guarantees.
>
> As we stress throughout, without leveraging the structure of the ATMG, our techniques would not work. The character of the structure in most of our arguments is dominant.
> It is precisely the ATMG structure that allows the application of a nested loop policy gradient algorithm (ISPNG) with each team agents taking one gradient step simultaneously after the adversary best-responds.
>  Without the ATMG structure we would not be able to establish the nonconvex--hidden-convex structure of the minmax problem at hand.
> The structure of the problem allows us to transform it to nonconvex–hidden-*strongly*-convex problem using the regularizer on a quantity that has a natural meaning for this problem.
> We adopt the $\ell_2$ regularizer because its gradient is merely proportional to the state-action visitation measure and we can get a nearly-unbiased gradient estimator of the resulting function that also has bounded variance given the policy of the adversary lies in the $\zeta$-truncated simplex.
>
> > *”The authors address a very classical optimization problem and apply their findings to an Adversarial Team Markov Game. “*
>
> Many problems of learning NE in Markov games (in both single-agent and multi-agent settings) boil down to an optimization problem of finding a stationary point of a nonconvex-nonconcave function [1]. In fact, we started off solving this particular problem of learning a NE in ATMGs and later extended it to a optimization setting that is more general.
>
> Indeed, theorem 3.1 is a cornerstone. But in order for us to be able to use it we need to formally prove that team agents performing gradient steps on the value function after the adversary best responds is equivalent to performing gradient steps on a nonconvex function with Holder continuous gradient.
>
> Evenmore, we need to design the gradient estimators and bound their variance and inexactness errors.
>
> >*”Although Theorem 3.1 is particularly impressive, the subsequent theorems follow from it primarily due to the compactness and finite state and action space of MARL. “*
>
> Despite the significant contribution of Theorem 3.1, it only tackles the problem of stochastic constrained optimization of a function whose gradient is merely Holder-continuous, but not in an ATMG setting. Thus, a lot of extra work is needed in order to prove that the function $\Phi^\nu$ is indeed a function that is differentiable with Holder-continuous gradient and also quantifying the Holder constant and exponent $(\ell_p, p)$. Indeed, one of our main contribution is to prove the gradient of $\Phi^{\nu}$ is Holder-continuous and the techniques we use for the analysis is novel and of significant theoretical value.
>
> Moreover, we want to respect the requirement that players will only get samples of trajectories as feedback and will not share explicit information about each other's policies. This means that for the team agents they can never get the exact gradient of $\Phi^\nu$ even if they had an infinite number of sample-trajectories at hand. Please also see the global rebuttal “Novelty in our paper”.
>
> With all the difficulties mentioned above, we managed to prove that projected gradient descent with stochastic inexact gradient converges to a stationary point when the function has Holder continuous gradient. Specifically, we prove that the team agents do not need an exact estimate of the gradient as long as the error can be controlled. We put a lot of effort in bounding this error which also affects the Holder-constant of the function $\Phi^\nu$.
>
>
> > *”This paper, lacking experimental results, seems more suitable for an optimization journal. I know several MG theory papers are already accepted to NeurIPS, even without any experiment. To be clear, I do not want to discourage authors about the fit for the venue.”*
>
> We will add numerical results but we agree with you that a lot of papers that are published in ICML, ICLR, NeurIPS, COLT do not necessarily need them. We strongly believe this conference is a good fit as attested by the multitude of theoretical MARL/AGT papers that get published every year even without experiments. Also see the rebuttal for reviewer hE5h for a list of papers of the same scope that where published in these conferences.
>
>
> We thank the reviewer again for their time. Perhaps after reading our replies they can reconsider the score they gave us.
>
> Best regards,
>
> The authors
>
> [1] A. Agarwal, S. M. Kakade, J. D. Lee, and G. Mahajan. On the theory of policy gradient methods: Optimality, approximation, and distribution shift. Journal of Machine Learning Research, 22(98):1–76, 2021

---

> > ### Comment · Reviewer_LXfR · 2024-08-07
> > **Thank you**
> >
> > Thank you for your comment. I will maintain my score.

---

> > > ### Author Response · Authors · 2024-08-07
> > > **What was not satisfactory in our response?**
> > >
> > > Thank you for your response.
> > >
> > > Given that our paper is primarily theoretical, it should be assessed on its theoretical contributions. For context, please refer to our response to Reviewer hEh5, where we list similar papers published in conferences such as NeurIPS, ICML, ICLR, and COLT. These conferences consistently feature numerous theoretical papers each year. We have meticulously addressed all the concerns raised by the reviewers and made substantial efforts in designing and executing experiments to bolster our findings.
> > >
> > > Could you please clarify the reasoning behind maintaining the same score? Were our revisions insufficient in addressing your comments?

---

> ### Comment · Reviewer_LXfR · 2024-08-07
> **I will decrease the score**
>
> I feel really bad about your comment. I already mentioned to the AC that I do not want to review as I do not wish to discourage the people who work on pure theory. My score is not affected by the suitability to this conference, and I also mentioned I do not want to discourage the authors. However, I do not understand why the authors think that rebuttal should increase the score.
>
> Now I will oppose this paper being accepted to NeurIPS. I do not think that a similar paper being accepted to NeurIPS should be the reason for this paper to be accepted. For example, should a traditional VAE related paper be accepted just because it was the best paper before? I do not think so.
>
> This was what I deleted (not to discourage the authors) in my previous private comment:
> >Although Theorem 3.1 is particularly impressive, the subsequent theorems follow from it primarily due to the compactness and finite state and action space of MARL. This paper, lacking experimental results, seems more suitable for an optimization journal.
>
> >Despite my reservations about its fit for NeurIPS, I do not wish to officially discourage the authors. My thoughts might be only for me and not reflective of the broader community focused on optimization theory. Therefore, I am inclined not to submit my review.
>
> >Theorem 3.1 stands out with significant contributions, while the other theorems, in my view, merely adapt the MARL problem to a specific instance of an optimization problem. The paper does not leverage the special structure of Markov Games (or Team Markov Games). Nonetheless, I appreciate the paper's clarity in writing and its theoretical contributions.
>
> > I hope this comment helps in your decision-making process. Feel free to share it with other reviewers, but I would prefer it not be shared with the authors since my primary concern is about the paper's venue suitability. If AC thinks that paper is not in the borderline (either accept or reject), AC does not need to consider my comments.
>
> I know that it is not easy to prove Lemma C.1 and C.2 (i.e., not taken for granted), but many papers have already used that fact. The Lipschitzness of the value function and the gradient of the value function are now almost taken for granted, as far as I know. I also do RL theory and game theory.

---

> ### Author Response · Authors · 2024-08-07
> **Apologies and Clarification**
>
> We kindly apologize if we offended you. This goes well beyond our intentions! We are asking in order to be able to further improve our work -- this was our sole intention. We do not take your time for granted and we are grateful for the effort you put into the review and indeed you made yourself clear in not wanting to discourage us. We also do not take for granted that the rebuttal should increase the score.
>
>
> Having said that, we think that there is some misunderstanding:
> > I know that it is not easy to prove Lemma C.1 and C.2 (i.e., not taken for granted), but many papers have already used that fact. The Lipschitzness of the value function and the gradient of the value function are now almost taken for granted, as far as I know. I also do RL theory and game theory.
>
> **We never claimed lemma c.1 and c.2 are novel**. We claim that Theorem 3.2 (Holder continuity of the gradient of the regularized max function, $\Phi^\nu$--this is weaker than being Lipschitz continuous) is novel, which can be used along the novel Theorem 3.1 (convergence of gradient descent when the gradient is not Lipschitz in constrained nonconvex objectives) to finally end up with our main game-theoretic result (Theorem 3.3). The team agents are running policy gradient on a function that **does not** have Lipschitz gradients (as is the case for virtually all RL applicaiton) and the method still provably converges. We hope this clarifies our claims.
>
> >Now I will oppose this paper being accepted to NeurIPS. I do not think that a similar paper being accepted to NeurIPS should be the reason for this paper to be accepted. For example, should a traditional VAE related paper be accepted just because it was the best paper before? I do not think so.
>
> Our claim is not that our paper should be published; as this is for the conference contributors to decide. We would never claim it should be published based on similarity. We do claim that we solve an open problem of a previous paper that was distinguished in ICLR '23; all the while introducing new optimization results of independent interest.

---

> > ### Comment · Reviewer_LXfR · 2024-08-08
> > **I re-increased my score**
> >
> > I see. Thank you so much! I appreciate it, and got author's point.

---

### Official Review · Reviewer_YssW · 2024-09-08

**Soundness:** 3
**Presentation:** 3
**Contribution:** 3
**Rating:** 7
**Confidence:** 4

**Summary:**

This paper considers the problem of independent policy gradient learning in adversarial team markov games (ATMGs). In such games, a team of agents with identical reward functions aim to compete against a single adversary whose reward function is the negation of the team's. The paper consider the setting where the MG has unknown transitions and reward functions, and the players can only obtain information about it by sampling trajectories and observing the resulting states and rewards. The main result (Theorem 3.3) gives an efficient algorithm which produces a policy which is an epsilon-Nash equilibrium of the ATMG, in which the players act independently, and which requires a number of samples which is polynomial in the number of states, actions, horizon, 1/epsilon, and distribution mismatch coefficient.

**Strengths:**

- Previously it was only known how to compute a NE in an ATMG in the presence of known transitions and rewards ([65] in the paper). Although the algorithm in [65] is a gradient descent-style algorithm, the present paper makes a convincing argument that the existing approach is insufficient (essentially because one cannot solve a certain linear program exactly).

- As such, the paper introduces several new ideas. One key idea is the regularization of the adversary's value function by the squared l_2 norm of their *state visitation distribution* (denoted by lambda in the paper). This ensures that the adversary's value function is strongly convex as a function of their visitation distribution, which facilities Danskin theorem-type arguments.

- As a result of the above idea, the "max-function" of the team's value function (i.e., the regularized value the team receives if the adversary best-responds) turns out to only be Holder-continuous (not Lipschitz continuous, as is typically the case). The paper then proves a new convergence result for gradient descent on nonconvex Holder-continuous functions (which seems to use similar ideas to previous work [35] in the convex case, but is nice to have written out).

**Weaknesses:**

(A) One weakness is that the algorithm has an "inner loop", in which the adversary must run several steps for each step of the team. This is in contrast to, e.g., [27], which has a stronger notion of independent learning (i.e., without an inner loop).

(B) It would be nice to have some more description about why Algorithm 2 works: in particular, it's not immediately clear how to interpret the updates to r(x) (ie subtracting nu * \hat lambda) in line 4. Since r(x) does not depend on y, this doesn't seem exactly like what GD is doing?

(C) To support the paper's claim that Moreau envelope techniques do not apply, it would be nice to have some result giving an example where Phi^nu(x) is *not* weakly convex (as typically in such situations one would expect Moreau envelope techniques to work).

(D) The paper claims prominently (e.g. in the abstract) it solves "the main open question from [65]". I could not find any reference in [65] to the open question which the paper claims to solve. The paper should be updated to either say where specifically that open question is stated, or else rephrase the claim (which is misleading if [65] does not state this explicitly as an open problem).

**Questions:**

See "weaknesses" above.

**Limitations:**

The paper should be updated to address (A) + (D) in the weaknesses section above (both of which can be fixed by changing a couple of sentences).

---

### Author Rebuttal · Authors · 2024-08-06

We thank the reviewers for their valuable comments. We will integrate their suggestions and corrections in our next draft.

As a disclaimer, our focus is on the theoretical advancements in algorithmic game theory and multi-agent reinforcement learning (MARL). We explicitly do not make claims about the immediate practicality of our algorithms. While our experiments (see accompanying PDF) suggest favorable empirical performance, a full assessment of their practical utility lies beyond the scope of this paper.


__Novelty in our paper__

Our main novelty lies in the techniques we use to get guarantees of adding regularizer to the adversary’s maximization routine. Here are the challenges we addressed:

1. Independent gradient steps by both players can diverge from the NE. In [4], adversary best-responses before team agents' gradient steps lead to a function $\Phi(x) = \max_{y} V(x,y)$, converging to a strategy $x^*$. Extending to NE $x^*, y^*$ requires knowledge of reward and transition functions, making it a planning problem, not RL. Additionally, $\Phi$ is not differentiable.

2.  Our work introduces a state-action occupancy regularizer and further proves that $\Phi^\nu$ is differentiable but has a Hölder-continuous (non-Lipschitz) gradient.

3. Unlike the ubiquitous Lipschitz-gradient assumption in RL and ML, we prove that stochastic projected gradient descent can converge to a stationary point in a constrained nonconvex optimization problem with a Holder gradient.

4. Agents are only allowed to observe their own reward and state trajectory samples. This disables team agents to get an exact gradient of the regularized function $\Phi^\nu$. (This inexactness comes from the fact that estimating the gradient of the regularizer requires observing the adversary’s actions)

5. Nevertheless, the coefficient of the regularizer controls the error between the gradient the team agents estimate and the real gradient of the function.

6. In Section 4, we show that our results can be significantly generalized to the setting of any minmax optimization problem of a nonconvex--hidden-strongly-concave objective.


__Importance of Policy Gradient Methods__

Policy gradient (PG) and policy optimization (PO) methods have dominated modern RL theory and practice. The policy gradient methods of PPO and TRPO are widely used by practitioners (e..g OpenAI).

In MARL, PG/PO methods manage to *break the curse of multi-agents* (the complexity scales exponentially with the number of players) and require minimal to no communication between agents. Further, PG/PO methods can be used alongside neural networks.

We tackle the case of tabular ATMGs with policy gradient methods making significant headspace towards the provable convergence of methods that involves neural nets.

__Self-Play is Inevitable__

In [1], it was proven that there is no computationally efficient algorithm that achieves no-regret even in 2-player 0-sum MGs. This means that self-play inevitable. We follow the *independent learning protocol* that is ubiquitous in contemporary MARL theory (e.g. [2], [3]).

Bandit feedback is a term that refers to only getting feedback for the action the agent takes. It does not need to be reserved for the online learning optimization framework. All agents collect samples of trajectories and then estimate their gradients. The adversary is the name of player that maximizes the value that the team tries to minimize. The adversary of an ATMG should not be confused with what "adversary" signifies in online optimization. ($r_{adversary} = \sum_{i=1}^{n}r_{player-i}$).

__Polynomial Iteration and Sample Complexity__

We emphasize that our work contributes *the first algorithm with sample complexity bounds that are polynomial* in the number of players, $1/\epsilon$, the sum of the action-space sizes, $\gamma$, and other relevant parameters. [4] did not offer any sample complexity bounds. Previously, it was not even known whether this is possible and we cannot know how --if at all-- suboptimal this algorithm is without any formal reasoning and arguments. It is fair that the reviewers formally argue why our bound is suboptimal -- what optimization lower bound is invoked and serves as the benchmark to compare to?

In theoretical research, demonstrating that polynomial iteration/sample complexity holds is always of significant interest, regardless of the polynomial's degree. From a theoretical computer science perspective, the gap between exponential and polynomial complexity is abysmal. For example, the discovery of an algorithm for the Traveling Salesman Problem with a complexity of even $O(n^{10,000})$ would revolutionize our understanding of computer science.

For the significantly simpler setting of two-player zero-sum MG the sample complexity of the algorithm in [2] scales with $1/(1-\gamma)^{48.5}$ and $\epsilon^{-12.5}$.

__Experiments__

We present the confidence intervals of the best-iterate's NE-Gap across iterations and the NE-Gap for every iteration of ISPNG when running in randomly generated games, where $S=4, \gamma=0.9, |A_1| = |A_2| = |B| = 3$, transition and reward functions are randomly generated. We observe that in these games the algorithm converges to a NE much faster than the proposed theoretical upper bound. See the accompanying plots.

Sincerely,

The authors


---

[1] Bai, Y., Jin, C. and Yu, T., 2020. Near-optimal reinforcement learning with self-play. NeurIPS

[2] Daskalakis, C., Foster, D.J. and Golowich, N., 2020. Independent policy gradient methods for competitive reinforcement learning. NeurIPS

[3] Ding, D., Wei, C.Y., Zhang, K. and Jovanovic, M., 2022, June. Independent policy gradient for large-scale markov potential games: Sharper rates, function approximation, and game-agnostic convergence. ICML

[4] Kalogiannis, F., Anagnostides, I., Panageas, I., Vlatakis-Gkaragkounis, E.V., Chatziafratis, V. and Stavroulakis, S.A., Efficiently Computing Nash Equilibria in Adversarial Team Markov Games. ICLR

---

### Decision · Program_Chairs · 2024-09-25

**Decision:**

Accept (poster)

**Comment:**

This paper considers the problem independent learning/decentralized equilibrium computation in multi-agent reinforcement learning (MARL). The paper focuses on infinite-horizon adversarial team Markov games (ATMGs), where a collection of agents with a common reward function compete against a single opponent (the "adversary"). The main result is to provide a new decentralized learning algorithm based on independent policy gradient methods which achieves polynomial sample complexity in finite state spaces. This addresses a question left open by Kalogiannis et al (2023), who give similar guarantees under the additional assumption that rewards and dynamics are known.

This paper makes a solid and clear contribution by providing the first finite-sample guarantees for independent learning in ATMGs. The core techniques in the paper, which apply to non-convex/hidden-concave min-max optimization with non-Lipschitz gradients, are also likely to have broader interest to the optimization theory community.

The initial set of reviewers generally agreed with the significance of the main results, they raised a number of small issues, including 1) poor polynomial dependence on parameters in the sample complexity guarantees, and 2) the fact that while the algorithm is decentralized, it requires players to follow a set schedule. However, these drawbacks are somewhat superficial, and they are shared by all prior work from this line of research (e.g., Daskalakis et al. (2020)).

Since the paper received borderline scores and the reviews above had limited depth, I recruited an expert reviewer (YssW) who has worked directly on this topic during the final discussion period. They agreed that the techniques in this paper are novel and non-trivial, and championed acceptance. I agree with this conclusion. While it would certainly be interesting to see these issues raised by the other reviewers addressed in future work, I believe this work merits acceptance in its current state.

I strongly encourage the authors to address the comments raised by reviewer YssW in the final revision, as well as highlight the limitations of the work raised by other reviewers (e.g., large polynomial sample complexity) up front.